# Unveiling the impact of potential evapotranspiration method selection on trends in hydrological cycle components across Europe

Vishal Thakur[1], Yannis Markonis[1], Rohini Kumar[2], Johanna Ruth Thomson[1], Mijael Rodrigo Vargas Godoy[1,3], Martin Hanel[1], and Oldrich Rakovec[1]

[1] Faculty of Environmental Sciences, Czech University of Life Sciences Prague, Kamýcká 129, Praha – Suchdol, Czech Republic

[3] River-Coastal Science & Engineering, Tulane University, New Orleans, 70118, Louisiana, USA

[2] Department Computational Hydrosystems, UFZ-Helmholtz Centre for Environmental Research, Leipzig, Germany

**Correspondence:** Vishal Thakur (thakur@fzp.czu.cz)

**Abstract.**

Hydrological models are essential tools for assessing changes in the hydrological cycle, enabling detailed quantification of runoff (Q), total water storage (TWS), and actual evapotranspiration (AET). Precipitation (P) and potential evapotranspiration (PET) are the two major drivers in modeling these components, with the influence of P more extensively studied than PET. This study evaluates the impact of PET method selection on AET, Q, and TWS using 12 PET formulations categorized as temperature-based, radiation-based, and combinational. We applied the mesoscale Hydrological Model (mHM) to simulate 40 years of hydrological components across 553 European catchments. PET effects were analyzed through trend analysis and the Data Concurrence Index (DCI) across three catchment categories: energy-limited, mixed, and water-limited. Our results indicate that annual and seasonal trends are variably sensitive to method choice, depending on each component and catchment category. While PET shows strong agreement in trend direction, trend magnitudes vary among different PET methods. Jensen-Haise consistently produces the highest annual and seasonal PET trend magnitudes, whereas no single method consistently yields the lowest. AET trends generally align with PET but are weaker in magnitude on an annual scale. Seasonally, only energy-limited catchments show AET trends similar to PET. For Q, and TWS, most European catchments exhibit strong trend agreement across PET methods. As expected, summer is the primary contributor to annual PET trends, while for AET, its influence is most notable in energy-limited catchments. Looking at statistically significant trends, there is general agreement for PET and AET, which decline for the other hydroclimatic variables. On an annual scale, varying patterns of hydrological cycle intensification (increases in P, AET, Q, and TWS) are observed across European catchments, highlighting the influence of

PET method selection. Overall, this study highlights how the PET method selection affects the quantification of hydrological trends, emphasizing the importance of method selection for robust assessment of AET, Q, and TWS.

## 1   Introduction

Potential evapotranspiration (PET) is the potential to evaporate water from the land surface to the atmosphere without any limitation to water availability. Although the concept has been in use for centuries, Thornthwaite (1948) was the first to formally introduce the term "potential evapotranspiration" in the scientific literature. A related but distinct concept is "reference
crop evapotranspiration", which is sometimes used interchangeably with PET. However, these terms differ in their conceptual basis and applications. Reference crop evapotranspiration specifically estimates the water requirements of a standardized reference crop under ideal conditions, whereas potential evapotranspiration provides a broader representation of water and energy exchange processes over diverse landscapes and large regions (Xiang et al., 2020). PET is used in diverse research fields. In environmental studies, PET is used for aridification research and investigating extreme events, including meteorological,
agricultural, and hydrological droughts (Park et al., 2018; Zhou et al., 2023; Shi et al., 2023a). In hydrology, it is used to determine the long-term states of catchments, such as energy-limited and water-limited catchments, and it plays a key role in the Budyko framework for estimating long-term changes in hydrological components (Reaver et al., 2022). Furthermore, PET is extensively used in hydrological modeling as to define the maximum rate of possible water loss through evaporation and transpiration. It is used as one of the important input variables to simulate key hydrological components, such as actual
evapotranspiration (AET), runoff (Q), and total water storage (TWS).

Since Thornthwaite's study, more than 100 empirical PET equations have been developed, ranging from simple to complex types (Proutsos et al., 2023). They can be classified mainly into three categories based on input data: (1) Temperature-based methods, which utilize temperature as input (Shaw and Riha, 2011). Due to their simplicity and minimal data requirement, these are widely used in hydrological model (Arnold et al., 1998; Liu et al., 2008). (2) Radiation-based methods require solar
radiation (short wave or net radiation) (Xu and Singh, 2000). (3) The combinational type requires temperature, radiation, wind speed, relative humidity, vapor pressure, etc. (Vicente-Serrano et al., 2014; Allen, 1998). Out of these 100+ methods, the majority are temperature-based methods (40+), followed by radiation-based methods (30+) and combination-based methods (10+) (Proutsos et al., 2023). Many of these empirical methods were initially developed and tested for particular regional scales or climatic conditions. For instance, the Thornthwaite method is most suitable for humid climates, while the Hargreaves-
Samani method is particularly effective in arid and semi-arid regions. Similarly, the Hamon method is suitable for all climates. All methods in these three categories incorporate several assumptions (climatic conditions and data availability) resulting in significant differences in their estimates (Lu et al., 2005).

PET influences AET and consequently impacts the estimation of infiltration, Q, and TWS in hydrological models. PET can have direct as well as indirect influence on AET. In hydrological models, AET is estimated by either separately determining
water surface evaporation, soil evaporation, and vegetation transpiration and then combining these based on land use patterns or by first assessing potential evapotranspiration and subsequently adjusting it to actual evapotranspiration using the soil moisture extraction function (Zhao et al., 2013). The mesoscale Hydrological Model (mHM) explicitly represents interception, where a

portion of AET is derived from interception evaporation. This process is estimated as a fraction of PET using a power function derived from Deardorff (1978) and Liang et al. (1994). When the evaporative demand exceeds the intercepted water, the interception storage is fully depleted. Interception storage in the mHM is estimated as a function of leaf area index (LAI) that varies depending on vegetation types and season. AET in mHM is mainly contributed by canopy evaporation, soil evaporation, and open water evaporation. AET, being a key component of the water balance, affects the estimation of other water balance components (Q, and TWS). While Q remains relatively insensitive, AET and TWS are more responsive to the choice of PET method (Bai et al., 2016). Hence, uncertainty in PET estimation influences the quantification of change in water cycle components.

Many studies have investigated the sensitivity of the hydrological model output to PET. Oudin et al. (2005) evaluated 27 PET methods with four hydrological models concluding that PET is insensitive to runoff generation, with similar conclusions made by Aouissi et al. (2016) and Birhanu et al. (2018). Assessment of four PET methods with two monthly hydrological models reported that runoff is unaffected by the PET method, whereas AET and total water storage depend on the PET method (Bai et al., 2016). The study also concluded that calibration against the runoff is the main cause of PET insensitivity, and AET and total water storage compensate for it. In contrast to previous studies, Ndiaye et al. (2024) compared 21 PET methods for runoff estimation with three conceptual lumped hydrological models (GR4J, GR5J, and GR6J) in the Senegal River Basin, stating that better performance shown by combinational type methods. Similarly, Pimentel et al. (2023) compared three PET methods for their accuracy in simulating runoff and AET in the large-scale hydrological model (HYPE model). They found that Hargreaves-Samani performed best in the Amazonas, central Europe, and Oceania, and Priestley-Taylor in higher latitudes. These studies focus on the sensitivity and choice of PET methods in estimating hydrological components. While these findings reveal how PET methods can impact the magnitude of hydrological components, the impact of PET method selection on changes in these hydrological components is not often investigated. Temporal changes in these hydrological components are crucial for climate change mitigation, water availability, energy availability, and agricultural produce.

Trends in PET and its implication on hydrological components (AET) are examined by Anabalón and Sharma (2017). They compare trends in six PET and AET datasets, mainly estimated by the Penman-Monteith or Priestley-Taylor PET method. They found that PET trends were highly correlated with AET trends in energy-limited regions, while the AET trends were closely correlated with precipitation trends in water-limited regions. Additionally, they reported that PET and AET trends were inversely related in certain cases, mainly due to the prevailing influence of precipitation trends on AET trends. Similarly, Liu et al. (2022) identified a strong positive relationship between PET and AET changes in most global regions and an inverse relationship with total water storage change. The study is limited by using only the Penman-Monteith approach for PET and global datasets for AET and total water storage change. The inconsistency and lack of coherence between existing PET and AET datasets often necessitate using a single PET method compared to various AET datasets. Furthermore, previous studies have primarily focused on one-to-one trend comparisons than comprehensive analysis of all hydrological cycle components, including Q, and TWS. Thus, research is needed to explore the impact of changes in PET methods on changes in different hydrological components of hydrological models.

In this study, our objective is to assess the trends of PET using 12 different PET methods and their influence on the trend of hydrological components (AET, Q, and TWS) across 553 European catchments. We further evaluate the agreement among PET methods by applying the Data Concurrence Index (DCI) to the trends of each corresponding hydrological cycle component (AET, Q, and TWS). The mesoscale hydrological model (mHM) is used to evaluate the influence of changes in different PET methods, from simple to most advanced approaches, on hydrological components across a range of European catchments. We chose a concurrency index to assess agreement between the PET method and hydrological components at each catchment. The data concurrency index is used to compare directions between different datasets (Anabalón and Sharma, 2017). In our research, we use it to examine directional changes in PET estimates, AET, Q, and TWS across each catchment.

## 2 Methods and data

### 2.1 Study area and catchment classification

This study includes 553 European catchments ranging in size from $500\ \mathrm{km}^2$ to $252\,000\ \mathrm{km}^2$. Catchments were selected based on the following criteria: first, a minimum area of $500\ \mathrm{km}^2$; second, at least 10 years of observed discharge data from GRDC database; and third, a closed water balance condition $((P - Q)/P < 1)$. The selected catchments are divided into three categories based on the aridity index: energy-limited, mixed, and water-limited (Figure 1a). This classification is based on the aridity index (AI), estimated as the ratio of mean PET to mean precipitation, a widely used metric that quantifies the dry or wet state of the catchment (Zhang et al., 2016; Massari et al., 2022). In our approach, which involves the application of multiple PET methods, a catchment is considered energy-limited if the AI is less than one for all the PET methods. Similarly, a catchment is water-limited if all PET methods report AI greater than one. If AI values appear to be both above and below one, depending on the PET method used, then the catchment is assigned to the mixed category (Figure 1b). Three representative catchments from each category are indicated by dark black lines in (Figure 1a) and are plotted in (Figure 1b). This classification allows us to distinguish the differences in magnitudes of PET and the other key hydrological components among the catchments (Figure 1c). By employing this methodology out of 553 catchments, we find 189 catchments being energy-limited, 34 water-limited, and the rest 330 belong to the mixed category.

### 2.2 Meteorological and geomorphological data

The Ensemble Meteorological Dataset for Planet Earth (EM-Earth; Tang et al., 2022) and ERA5-Land (Muñoz-Sabater et al., 2021) were used to calculate different PET estimates and run the mesoscale Hydrological Model (mHM; Samaniego et al., 2010; Kumar et al., 2013b). The EM-Earth dataset (Tang et al., 2022) is derived from observed station data SC-Earth (Tang et al., 2021) and ERA5 data (Hersbach et al., 2020). It incorporates a novel optimal interpolation technique and considers the temporal inconsistencies between the station and reanalysis data (Tang et al., 2022). ERA5-Land dataset is a reanalysis data product created by the European Centre for Medium-Range Weather Forecasts (ECMWF) and has been widely used in numerous hydrological modeling studies (Muñoz-Sabater et al., 2021). Both datasets are available at $0.1° \times 0.1°$ spatial

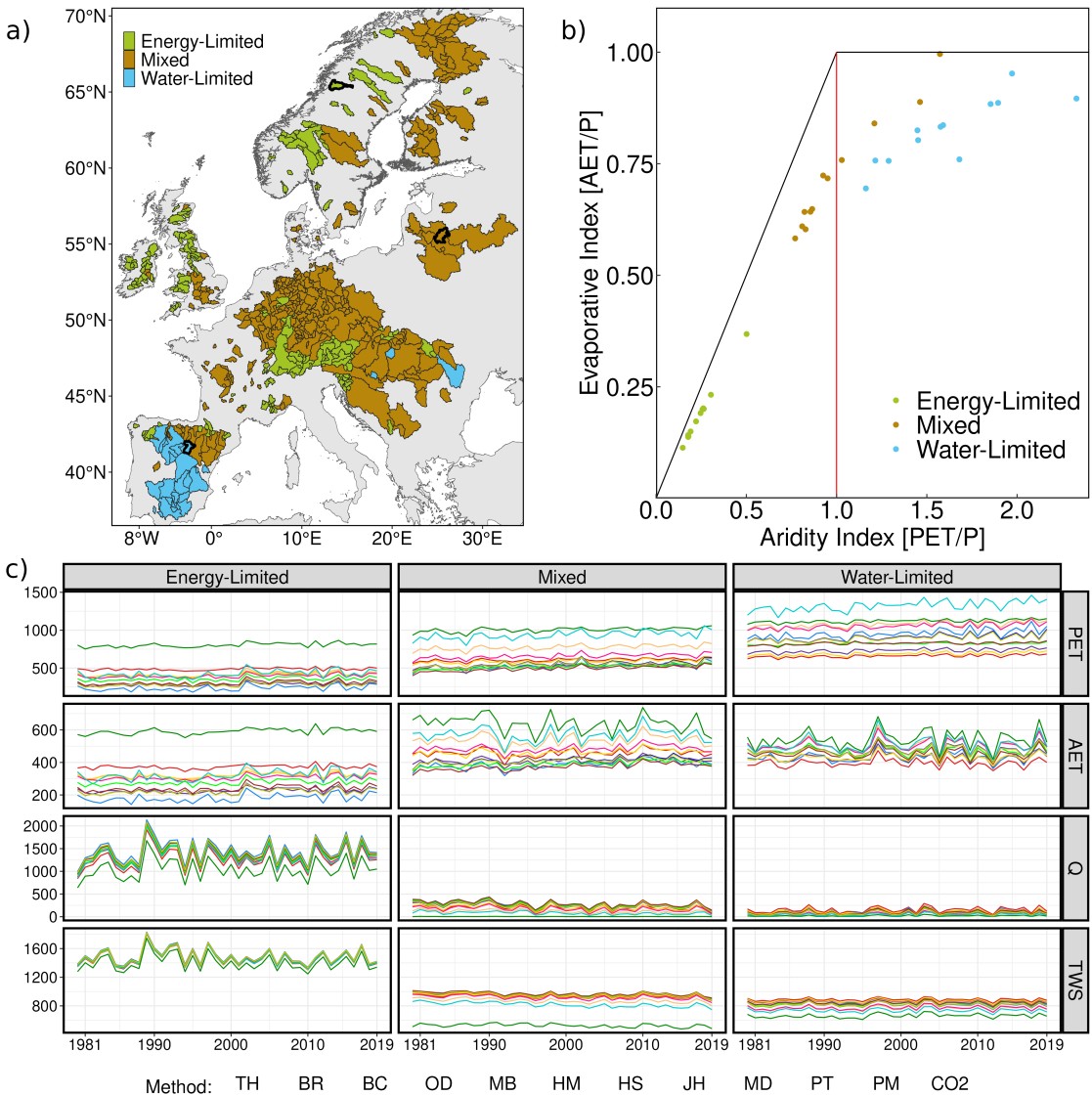

**Figure 1.** Catchment classification to energy-limited, mixed, and water-limited categories. a) Spatial location of catchments; black borders indicate a representative catchment of each category. b) Classification example within the Budyko space for the representative catchments. Colored points represent three representative catchments, with each set of 12 points per color corresponding to different PET methods of representative catchments. c) Annual time series of simulated hydrological components corresponding to each representative catchment and PET estimation method (TH: Thornthwaite, BR: Baier-Robertson, BC: Blaney-Criddle, OD: Oudin, MB: McGuinness-Bordne, HM: Hamon, HS: Hargreaves-Samani, JH: Jensen-Haise, MD: Milly-Dunne, PT: Priestley-Taylor, PM: Penman-Monteith, $CO_2$: Penman-Monteith[$CO_2$].). All units are in mm year$^{-1}$.

resolution, but EM-Earth has hourly as well as daily time step, while ERA5-Land is at hourly scale. Meteorological forcings, particularly precipitation and temperature from EM-Earth, have been found to be more accurate based on comparisons with

several datasets in data-rich regions (Tang et al., 2022).

In our analysis, we use daily temperature and precipitation from EM-Earth, and radiation (long- and short-wave), surface pressure, and wind components (U and V) from ERA5-Land for the period 1980–2019 (Table 1). The EM-Earth dataset provides high-quality precipitation and temperature data and has been shown to perform well over Europe (Tang et al., 2022). It has undergone climatology-based bias correction and accounts for precipitation undercatch. However, since EM-Earth does not include all necessary variables for PET estimation, we utilize ERA5-Land as a complementary dataset. ERA5-Land has been demonstrated to perform better than other reanalysis datasets, including ERA5 and ERA-Interim (Muñoz-Sabater et al., 2021). Nonetheless, its limitations in hydrological modeling have been acknowledged by Clerc-Schwarzenbach et al. (2024); Tarek et al. (2020). Several recent global studies follow a similar strategy, combining precipitation and temperature from EM-Earth with radiation, wind speed, and other meteorological variables from ERA5-Land (Tang et al., 2023; Yin et al., 2024; Rakovec et al., 2023). These meteorological data combinations, along with the simulated hydrological components derived from them, demonstrate lower uncertainty across Europe (Tang et al., 2023). These are homogenized to daily temporal scale and $0.125° \times 0.125°$ spatial scale to be compatible with the previous simulations run by mHM (Pohl et al., 2023; Fang et al., 2024). Homogenization using the nearest neighbor technique and necessary mathematical operations (appropriate unit conversion of datasets) are performed using the Climate Data Operator (CDO; Schulzweida, 2022).

Morphological data such as Leaf Area Index (LAI), soil properties, and terrain characteristics (such as flow direction, flow accumulation, slope, and aspect) are sourced from mHM European database (Rakovec et al., 2016). This database originally utilized data from different sources, such as soil properties from the International Soil Reference and Information Centre (ISRIC), terrain characteristics from the U.S. Geological Survey (USGS) and the National Geospatial-Intelligence Agency (NGA), LAI from Global Inventory Modeling and Mapping Studies (GIMMS) and Land cover from Global Land Cover (GlobCover) by European Space Agency (ESA). $CO_2$ concentration is sourced from Cheng et al. (2022), which is reconstructed from the Carbon Dioxide Information Analysis Center (CDIAC) data.

## 2.3 Methodology

### 2.3.1 PET methods/formulations

We incorporate 12 PET methods at a daily scale from all three categories of estimation: temperature, radiation, and combinational methods (Table 2). Temperature-based methods require temperature data, which can include average temperature, minimum temperature, or maximum temperature. Additionally, PET methods that incorporate extraterrestrial radiation are also considered under this category. Combinational-based methods require a larger number of variables compared to temperature- and radiation-based methods to estimate various physical terms, such as wind speed and surface pressure (Table 2). Most temperature-based methods use only daily average temperature (Thornthwaite, Oudin, Hamon, Jensen-Haise, McGuinness-Bordne, and Blaney-Criddle), while Baier-Robertson employs both minimum and maximum daily temperatures. Some of these methods also include an extraterrestrial radiation term in their formulation. However, since this radiation term is calculated based on latitude and follows a consistent annual cycle varying only with the calendar date, only temperature data is needed for PET calculation. We utilize only one radiation-based method, Milly-Dunne PET, which requires only net radiation

**Table 1.** Summary of meteorological and morphological data. P is Precipitation, $T_{avg}$ is average air temperature, $T_{range}$ is the temperature range, which is the difference between maximum and minimum air temperature, $T_{dew}$ is dew point temperature of air, SW is Short wave radiation, LW is Long wave radiation, U is eastward component of wind speed at 10 m, V is northward component of wind speed at 10 m, $Con_{CO_2}$ is $CO_2$ concentration

| Variable | Temporal Scale | Spatial Scale | Record length | Source | Reference |
|---|---|---|---|---|---|
| Meteorological data | | | | | |
| P | Hourly/Daily | $0.1° \times 0.1°$ | 1950–2019 | EM-Earth | Tang et al. (2022) |
| $T_{avg}$ | Hourly/Daily | $0.1° \times 0.1°$ | 1950–2019 | EM-Earth | Tang et al. (2022) |
| $T_{range}$ | Hourly/Daily | $0.1° \times 0.1°$ | 1950–2019 | EM-Earth | Tang et al. (2022) |
| $T_{dew}$ | Hourly/Daily | $0.1° \times 0.1°$ | 1950–2019 | EM-Earth | Tang et al. (2022) |
| SW | Hourly | $0.1° \times 0.1°$ | 1950-2022 | ERA5-Land | Muñoz-Sabater et al. (2021) |
| LW | Hourly | $0.1° \times 0.1°$ | 1950-2022 | ERA5-Land | Muñoz-Sabater et al. (2021) |
| U | Hourly | $0.1° \times 0.1°$ | 1950-2022 | ERA5-Land | Muñoz-Sabater et al. (2021) |
| V | Hourly | $0.1° \times 0.1°$ | 1950-2022 | ERA5-Land | Muñoz-Sabater et al. (2021) |
| Other data | | | | | |
| $Con_{CO_2}$ | Annual | $0.1° \times 0.1°$ | 1950-2022 | – | Cheng et al. (2022) |
| LAI | monthly | $1/512°$ | static | GIMMS | Tucker et al. (2005) |
| Soil properties | – | $1/512°$ | – | SoilGrids | ISRIC - World SoilInformation (2017) |
| Land cover | static | $1/512°$ | static | GlobCover | Arino et al. (2012) |
| DEM (+ derivatives) | static | $1/512°$ | static | GMTED2010 | USGS and NGA (2018) |
| Geology | static | $1/512°$ | static | GLiM | Hartmann and Moosdorf (2012) |

data to estimate PET. The combinational methods, such as Penman-Monteith and Priestley-Taylor, have a stronger physical
basis. In our analysis, all these physical terms are estimated following Allen (1998). Additionally, within the combinational
category, we employ the modified Penman-Monteith (CO2) method, which accounts for temporal variations in changing carbon
dioxide concentrations. Formulation details, including mathematical equations and associated constants for each PET method,
are provided in Table A1.

### 2.3.2   mesoscale Hydrological Model (mHM)

mHM is a hydrological model which explicitly accounts for sub-grid variability of hydrological processes (Samaniego et al.,
2010; Kumar et al., 2013b; Thober et al., 2019). mHM has been successfully applied and tested in more than 1000 Euro-
pean basins ranging in size from 4 $km^2$ to more than 100 000 $km^2$ at various spatial resolutions or grid cell size (1-100 km)
(Samaniego et al., 2010; Kumar et al., 2013b; Rakovec et al., 2016, 2019; Shrestha et al., 2024). Additionally, the model is
currently applied at the global scale with comparable and sometimes even improved model performance with respect to other
large-scale hydrological models (Samaniego et al., 2019). mHM demonstrates robust performance and applicability across
Europe (Kumar et al., 2020). It is also one of the several large-scale hydrological models, which were used by the WMO for
their annual State of Global Water Resources reports (World Meteorological Organization (WMO), 2023).

**Table 2.** List of PET methods and required input data. $T_{max}$ is maximum air temperature (°C), $T_{min}$ is minimum air temperature (°C), Pr is surface pressure (pa), $R_n$ is net radiation (J/m$^2$), $u_2$ is the wind speed at 2m from the surface (m/s), $Con_{CO_2}$ is $CO_2$ concentration (ppm)

| Type | Method name | Method abbreviation | Required input | References |
|---|---|---|---|---|
| Temperature | Hargreaves-Samani | HS | $T_{max}$, $T_{min}$, $T_{avg}$ | Hargreaves and Samani (1985) |
| | Thornthwaite | TH | $T_{avg}$ | Thornthwaite (1948) |
| | Oudin | OD | $T_{avg}$ | Oudin et al. (2005) |
| | Hamon | HM | $T_{avg}$ | Hamon (1961) |
| | Baier-Robertson | BR | $T_{max}$, $T_{min}$ | Bai et al. (2016) |
| | Jensen-Haise | JH | $T_{avg}$ | Jensen and Haise (1963) |
| | McGuinness-Bordne | MB | $T_{avg}$ | McGuinness and Bordne (1972) |
| | Blaney-Criddle | BC | $T_{avg}$ | Blaney (1952) |
| Radiation | Milly-Dunne | MD | $R_n$ | Milly and Dunne (2016) |
| Combinational | Priestley-Taylor | PT | $T_{avg}$, Pr, $R_n$ | Priestley and Taylor (1972) |
| | Penman-Monteith | PM | $T_{max}$, $T_{min}$, $T_{avg}$, $T_{dew}$, Pr, $u_2$, $R_n$ | Penman (1948) |
| | Penman-Monteith[$CO_2$] | $CO_2$ | $T_{max}$, $T_{min}$, $T_{avg}$, $T_{dew}$, Pr, $u_2$, $R_n$, $Con_{co2}$ | Yang et al. (2019) |

We run mHM(v5.12.0) over 553 European catchments, using the meteorological data from EM-Earth and the 12 different PET estimation methods. Overall, 6 636 (12 × 553) mHM simulations are performed for all the study basins. The model was set up for each catchment at a daily temporal resolution and 0.125° × 0.125° spatial scale. All meteorological forcings were kept constant with only varying PET estimates. To calculate TWS, we aggregate soil moisture at different layers, canopy interception storage, snowpack, groundwater levels, sealed area reservoirs, and unsaturated zone reservoirs at each grid cell and time step. The hydrological components (AET, Q, and TWS) and PET are averaged over the catchment area and monthly time steps. In this research default parametrization is used for model setup. The default parameterization of mHM has been shown to perform well in previous studies (Kumar et al., 2013a; Rakovec et al., 2016). Furthermore, it has been demonstrated as one of the best-performing configurations compared to other large-scale hydrological models (Samaniego et al., 2019). Additionally, our assessment of model performance is consistent with the findings of Samaniego et al. (2019). Our model evaluation against discharge shows that the median Kling–Gupta Efficiency (KGE) ranges from 0.60 to 0.75 across most PET methods (Figure S1).

### 2.3.3 Trend analysis

We use Theil-Sen's slope method to calculate the magnitude and direction of linear change in PET, AET, Q, and TWS (Sen, 1968). The Mann-Kendall trend test is used to test the statistical significance of the observed trend (Kendall, 1948). A trend is considered statistically significant at the 5% level ($p \leq 0.05$). Sen's slope is non-parametric and insensitive to outliers and

types of distribution. Due to its robust application, this method is widely used in hydrology, climate, and environmental-related studies (Anabalón and Sharma, 2017; Thackeray et al., 2022). It accounts for all possible pairs of data points from a time series and finds the median value as the slope magnitude. Eq. 1 and 2 shows the calculation steps of Sen's slope:

$$S_k = \frac{X_j - X_i}{t_j - t_i} \quad \text{where } 1 \leq i < j \leq n \tag{1}$$

$$S_{med} = \begin{cases} S_{\left[\frac{n+1}{2}\right]} & \text{if } n \text{ is odd} \\ \frac{S_{\left[\frac{n}{2}\right]} + S_{\left[\frac{n+2}{2}\right]}}{2} & \text{if } n \text{ is even,} \end{cases} \tag{2}$$

where $S_k$ is the linear slope for pair $X_i$ and $X_j$, $S_{med}$ is the median slope, $X_i$ and $X_j$ are data points from periods $t_i$ and $t_j$, $n$ is the number of data points in time series. Positive $S_{med}$ represents a positive trend, with the magnitude indicating the rate of increase. Similarly negative $S_{med}$ represents a negative trend, with the magnitude indicating the rate of decrease.

Here, we use the *trend* R package to estimate Sen's slope over a 40-year period from 1980 to 2019 at annual and seasonal (winter, spring, summer, and autumn) scales for each catchment. The units of trend at the annual scale are expressed as mm year$^{-1}$, while at the seasonal scale, they are represented as mm seas$^{-1}$ year$^{-1}$. For instance, a summer season trend of 1 mm seas$^{-1}$ year$^{-1}$ indicates that each year, an additional 1 mm is added to the summer season. The trend of each PET method were analyzed exclusively within seasons (for example winter season compared with winter season), without cross-seasonal comparisons.

### 2.3.4 Data Concurrence Index (DCI)

The Data Concurrence Index quantifies the level of concurrence between the significant trends in different datasets of the same variable (Anabalón and Sharma, 2017). DCI is applied in two forms in this study. The first is the original formulation, which includes only statistically significant trends. It is used to evaluate agreement among PET methods that yield significant changes in hydrological cycle components. The second is a modified version that considers all detected trends, regardless of significance. This captures the overall agreement across all PET methods for each catchment and each hydrological component. The detailed formulation for both cases are described in Eq. 3:

$$\text{DCI} = \begin{cases} \frac{1}{N} \sum_{i=1}^{N} \delta_i \cdot \frac{S_i}{|S_i|} & \text{For statistically significant trends} \\ \frac{1}{N} \sum_{i=1}^{N} \frac{S_i}{|S_i|} & \text{For all trends} \end{cases} \quad \text{with} \quad \delta_i = \begin{cases} 1, & \text{if } S_i \text{ is statistically significant} \\ 0, & \text{otherwise} \end{cases} \tag{3}$$

where DCI is the Data Concurrence Index, N denotes the number of datasets, and $S_i$ is the magnitude of the slope.

The positive DCI represents a higher number of positive slopes than negative slopes and vice-versa. For instance, a DCI of 1 for AET and Q implies positive change for all the PET methods. Similarly, a DCI of -1 for AET and Q implies a negative change for all the PET methods. A DCI of 0.5 indicates that nine out of 12 methods, or 75% of the methods, show a positive change, and similarly, a DCI of -0.5 indicates that nine out of 12 methods show a negative change, or 75% of the methods. A DCI of zero denotes an equal number of positive and negative slopes (six positive and six negative). Our analysis estimates DCI from PET, AET, Q, and TWS slopes at annual as well as seasonal scales.

## 3 Results

### 3.1 Trend comparison of PET methods at annual scales

By applying the Theil-Sen slope method, we observe that changes in PET depend on the choice of PET estimation formulation (Figure 2). Considerable variability is observed among the PET methods, with median slopes ranging from slightly positive to 6 mm year$^{-1}$ during the 1980–2019 period. The Jensen-Haise method shows the highest change among all methods across different catchment categories and also has the highest absolute average PET across European catchments (Figure S2). Generally, changes in PET are higher in water-limited than in energy-limited catchments. This difference arises since temperature-based methods depend on temperature changes. Conversely, combinational methods are influenced by more than one meteorological variable (temperature, wind speed, radiation, etc.). When we consider only statistically significant trends ($p < 0.05$), we observe that temperature-based PET methods consistently demonstrate statistically significant positive trends for most of the catchments (Table S2). Radiation-based and combinational methods account for fewer catchments than temperature-based PET methods (Table S2), implying that they generate weaker trends compared to the temperature-based PET methods. In addition, overall trend variability among the PET methods decreases from energy-limited to water-limited catchments, irrespective of trend significance.

AET trends generally align with PET trends in energy-limited catchments but with smaller magnitudes (Figure 2). In these catchments, all PET methods leads to a positive AET trend in terms of median values. However, a few catchments in this category reveal a slight negative change for the Blaney-Criddle, Jensen-Haise, Milly-Dunne, and Priestley-Taylor methods. For mixed catchments, the median AET trend is positive for all PET methods except Blaney-Criddle. The negative AET trends are similar to those in energy-limited catchments. Overall, the AET trend patterns (high and low trends) for energy-limited and mixed catchments are similar to the trends in PET for these catchments, regardless of trend magnitude, with a few exceptions such as Blaney-Criddle and Jensen-Haise (Figure 2). In water-limited catchments, both positive and negative trends in AET are observed. The pattern remains similar to PET trends with a few exceptions, such as Blaney-Criddle. Across all PET methods, statistically significant positive AET trends were found in 162 energy-limited, 217 mixed, and 2 water-limited catchments (Figure S3). Among these significant trends, Jensen-Haise yields the highest AET trend estimates in energy-limited and mixed catchments. These statistically significant trends closely follow the overall AET trend patterns in these catchment categories.

Q trends exhibit lower sensitivity to PET methods in energy-limited and mixed catchments compared to PET trends when considering all trends (Figure 2). Despite the positive median, a substantial fraction of catchments exhibit a negative trend in energy-limited catchments. In contrast, for mixed catchments, most PET methods produce negative Q trends, though some catchments maintain positive trends. In water-limited catchments, there is variability in PET methods; for instance, Milly-Dunne has a larger trend, whereas Blaney-Criddle shows the lowest trend. Even though PET methods are insensitive in Q, variability exists among the PET methods within each catchment category. Overall, one-third of the catchments (183) show statistically significant Q trends: 28 energy-limited, 154 mixed, and one water-limited. All energy-limited catchments show positive Q trends, except those estimated with Jensen-Haise. In mixed catchments, all statistically significant trends are negative, with the exception of Blaney-Criddle. Despite fewer catchments with statistically significant trends, the variability in Q trends across PET methods persists, particularly in energy-limited and mixed catchments.

TWS trends are sensitive to PET methods in water-limited and mixed catchments, but energy-limited catchments remain largely unaffected (Figure 2). Despite a consistent negative median trend across PET methods in energy-limited catchments, few catchments still exhibit positive trends. Mixed catchments display a similar pattern. Among PET methods, temperature-based approaches show greater variability than radiation or combinational types in mixed catchments. In water-limited catchments, trends span positive to near-zero-negative, with Blaney-Criddle, Hargreaves-Samani, and Milly-Dunne showing notable positive trends. The Blaney-Criddle method also yields higher variability in trend estimates, especially in mixed and water-limited catchments. When focusing on statistically significant trends, TWS has a similar distribution to Q (30/172/1). Most PET methods show decreasing trends in energy-limited and mixed catchments, with the exception again of Blaney-Criddle (Figure S3). Compared to the full trend set, statistically significant trends reveals stronger inter-method differences, with Jensen-Haise showing the steepest decline and Thornthwaite the weakest.

Hydrological trend estimates for AET, Q, and TWS vary across PET methods and catchment types, regardless of whether all trends or only statistically significant trends are considered. Statistically significant trends reveal greater divergence between methods in energy-limited and mixed catchments. AET trends, though weaker in magnitude than PET trends, show similar spatial patterns, particularly in energy-limited and mixed regions. No single PET method stands out as consistently dominant across all components. For the trends beyond the statistically significant threshold a stronger pattern emerges. Approximately 70% of the catchments exhibit statistically significant AET trends, compared to only 33% for Q, and 36% and TWS. Despite widespread statistical insignificance for Q, and TWS, distinct regional patterns appear. For instance, northern catchments display a mix of increasing and decreasing trends, while southern regions, especially the Iberian Peninsula, consistently demonstrate positive Q trends.

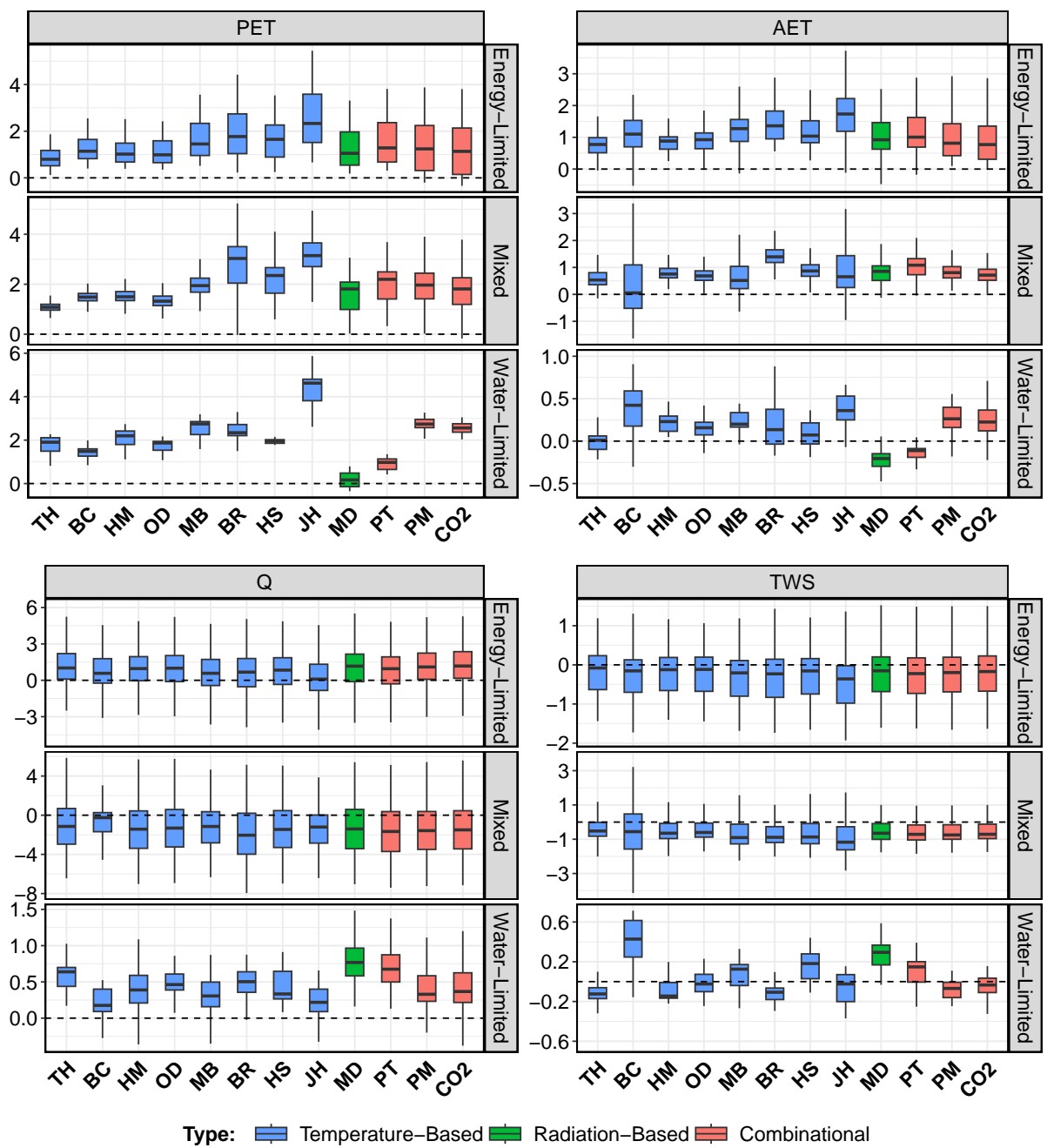

**Figure 2.** Boxplots represent the annual trends (mm year$^{-1}$) of different PET methods for PET, AET, Q, and TWS across various categories of catchments. The whiskers represent the 10th and 90th percentiles, and the box encompasses the 25th and 75th percentiles, with the median represented by middle line of the box. Abbreviations used for different PET methods are TH: Thornthwaite, BR: Baier-Robertson, BC: Blaney-Criddle, OD: Oudin, MB: McGuinness-Bordne, HM: Hamon, HS: Hargreaves-Samani, JH: Jensen-Haise, MD: Milly-Dunne, PT: Priestley-Taylor, PM: Penman-Monteith, CO$_2$: Penman-Monteith[$CO_2$].

## 3.2 Trend comparison of PET methods at seasonal scales

Hydrological cycle components exhibit considerable seasonal variability in trend magnitude. During summer (JJA), nearly all PET methods exhibit positive trends, with the exception of Milly-Dunne in water-limited catchments (Figure 3). Jensen-Haise consistently shows the highest trend and greatest variability across all catchments. Notable variability is observed among PET methods in water-limited catchments. These patterns persist in statistically significant trends, despite a smaller number of catchments per category (Figure S4). Winter records the weakest PET trends, while spring trends are comparable to those in summer (Text S2). The summer season is the primary contributor to the annual trends in PET across all catchment categories (Figure S5). The distribution of statistically significant trends aligns with these findings, with the highest catchment count in spring (460) and the lowest in winter (66).

In summer, AET exhibits an overall positive trend in energy-limited and mixed catchments for both all trends and statistically significant trends (Figure 3, S4). The Jensen-Haise method exhibits greater variability for energy-limited and mixed catchments. In water-limited catchments, despite positive PET trends, AET trends are negative across all PET methods, with Baier-Robertson showing the strongest decline, followed by Jensen-Haise (Figure 3). In spring, AET increases across methods, with the highest trends observed for Jensen-Haise in energy-limited and Baier-Robertson in mixed catchments. Combinational methods show consistent trends across both types (Figure S6). This pattern is maintained for statistically significant trends (Figure S7), with all PET methods showing positive median trends in autumn (Figure S8, S9). Summer season primarily drives annual trends in energy- and water-limited catchments, while spring and summer contribute jointly in mixed catchments depending on the PET method (Figure S5).

Q remains generally insensitive to PET method variation across all seasons and catchment types, with minor variability among methods within each category (Figure 3, S6, S8, S10). In energy-limited catchments, trends remain close to zero across PET methods. Mixed catchments show broadly negative Q trends across methods, with Blaney-Criddle exhibiting the weakest decline. In water-limited regions, Q trends are similarly insensitive, though radiation and combinational methods tend to positive median trends, unlike temperature-based methods, which exhibit mixed results. Statistically significant Q trends reveal limited seasonality in water-limited catchments, with notable exceptions like Blaney-Criddle, which contributes fewer catchments (Figures S4, S7, S9, S11). In energy-limited and mixed catchments, statistically significant Q trends demonstrate higher seasonal magnitude as weaker trends fail to surpass the significance threshold. No single PET method consistently dominates in trend magnitude, though Jensen-Haise and Blaney-Criddle frequently yield the highest number of statistically significant catchments. Spring emerges as the dominant contributor to annual Q trends in mixed and water-limited catchments, while in energy-limited areas, winter and summer are most influential, depending on the PET method (Figure S5).

TWS trends across seasons show minimal sensitivity to PET method in energy-limited and mixed catchments under all-trend conditions, though slight variability exists among PET methods (Figure 3). Water-limited catchments, however, display a mixed response. Median trend patterns are generally stable, but Blaney-Criddle consistently shows greater variability than other methods across all seasons and catchment categories. Statistically significant trends are uncommon in water-limited catchments, but trend magnitudes increase notably in energy-limited and mixed catchments during spring, summer, and au-

tumn. In winter, temperature-based methods demonstrate positive trends, with Blaney-Criddle and Jensen-Haise showing the lowest values (Figure S11). Slight variability across methods also seen in non-winter seasons (Figures S4, S7, S9). Seasonal contributions to annual trends vary by catchment category: spring contributes most in energy-limited catchments, summer in mixed catchments, and in water-limited catchments. The dominant season depends on the PET method—summer leads for several (TH, BR, HM, PM, and CO2), while spring dominates for the remaining methods.

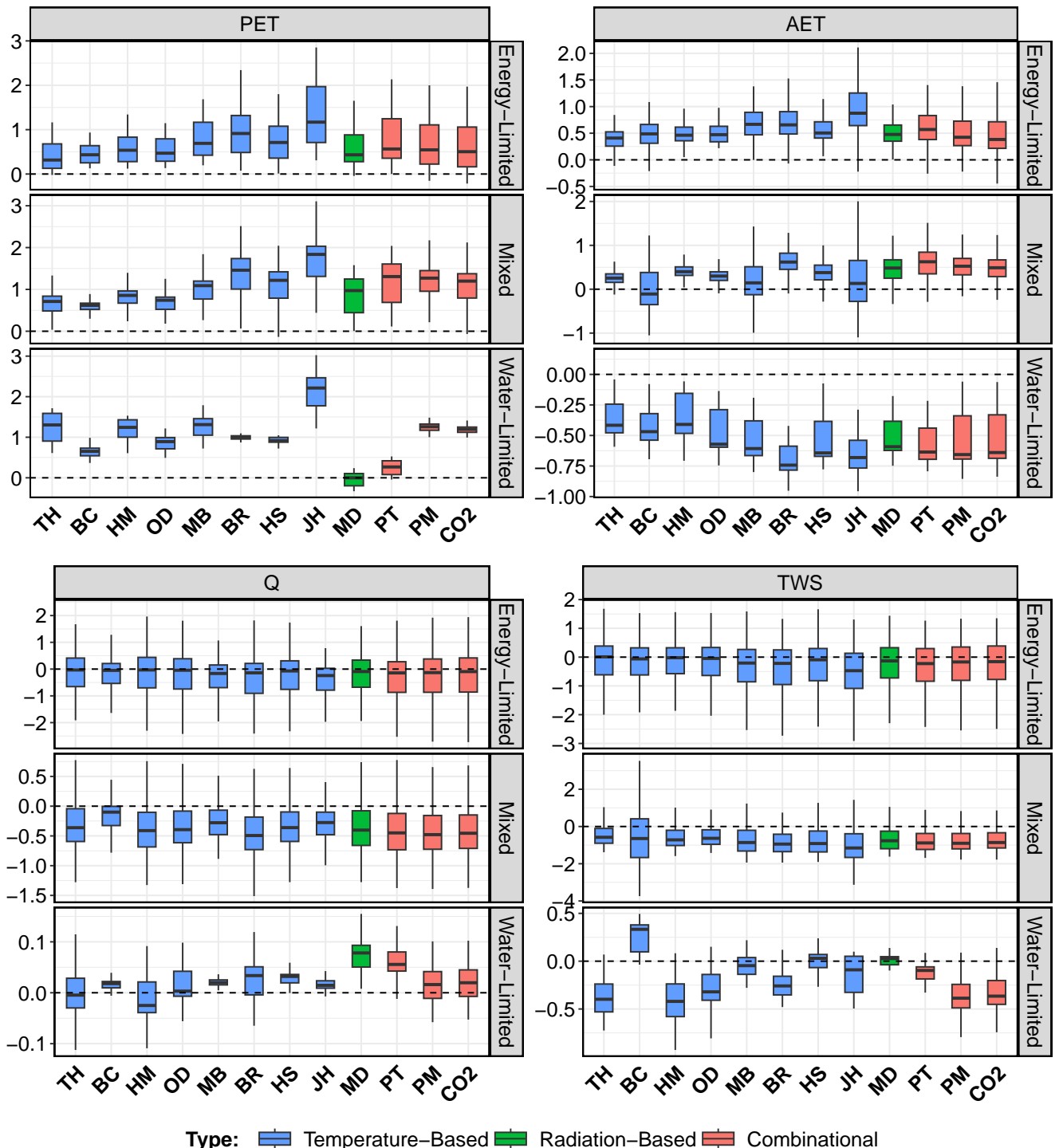

**Figure 3.** Boxplot represents summer season (JJA) trends of different PET methods for AET, and Q across three categories of catchments: energy-limited, mixed, and water-limited. The whiskers represent the 10th and 90th percentiles, and the box encompasses the 25th and 75th percentiles, with the median represented by the black line within the box. Abbreviations used for different PET methods are TH: Thornthwaite, BR: Baier-Robertson, BC: Blaney-Criddle, OD: Oudin, MB: Mcguinness-Bordne, HM: Hamon, HS: Hargreaves-Samani, JH: Jensen-Haise, MD: Milly-Dunne, PT: Priestley-Taylor, PM: Penman-Monteith, $CO_2$: Penman-Monteith$[CO_2]$. Trend units are in mm seas$^{-1}$ year$^{-1}$.

### 3.3 Catchment-wise DCI distribution across annual and seasonal scales

Even though there is a strong agreement across different PET methods in annual PET and AET trends, substantial variation exists in Q, and TWS responses (Figure 4). All catchments exhibit strong positive DCI for PET, indicating that at least 75% of methods report a positive trend. AET follows a similar pattern, with high positive DCI values across northern (Scandinavia), central, western, eastern, southeastern (Balkans), and parts of southern (Iberian Peninsula) Europe, except for a few southern catchments with low as well as negative DCI. In contrast, Q reflects strong positive agreement in southern regions but mostly negative DCI in northern and eastern Europe. Central and western European catchments are marked by both strong positive and negative DCI, with few catchments showing disagreement among PET methods. TWS shows widespread disagreement, especially in southern, western, central, southeastern, and northern European catchments. Most central, eastern, and northern catchments show strong negative agreement for TWS. Overall, PET methods show high directional consistency for PET and AET, but diverge notably for Q, and TWS.

The above findings appear consistent when the selection of PET methods becomes balanced to eight methods: four temperature-based, one radiation-based, and three combinational methods, with the latter two fixed across all 70 possible method combinations. All the 70 combination results are consistent with the previous findings. For TWS, only a few catchments in southern and western Europe show strong negative DCI, consistent with earlier areas of disagreement (Figure S12, S13). To assess the impact of weaker trends, we applied the DCI using only statistically significant trends. PET shows similar agreement patterns to the all-trend analysis, with minor disagreement in western Europe. For AET, disagreement emerges in most southern catchments and some in western, central, and eastern Europe, which previously showed agreement. Q, and TWS exhibit widespread disagreement, largely due to weak trends across PET methods. Catchments with strong agreement under statistically significant trends align well with those under all-trend analysis. PET and AET trends remain predominantly positive in both statistically significant and all-trend evaluations.

To better understand the annual changes of the level of concurrence between all PET method trends, we decompose them into sub-seasonal values. Figure 5 shows strong positive agreement for PET and AET across all seasons, while Q, and TWS exhibit predominantly negative agreement in central Europe, with evident regional variation. PET demonstrates high positive DCI in most catchments during spring, summer, and autumn, indicating consistent upward trends. In winter, agreement is highest in central, eastern, and southern Europe, but weaker elsewhere. AET shows strong seasonal agreement in central Europe, particularly in winter, spring, and summer, while disagreement emerges in northern and western regions during winter and in several southern and central catchments during autumn. AET also exhibits strong negative agreement in southern Europe during summer. Q consistently reflects a negative DCI across central Europe in all seasons, although southern catchments show positive agreement in spring and autumn. TWS trends show persistent negative DCI in central Europe across seasons, while southern Europe sees a shift from strong positive agreement in spring to strong negative in summer, with weak agreement in winter and autumn. Northern Europe shows a mix of agreement and disagreement across all seasons.

Comparing sub-seasonal concurrence based on statistically significant trends (Figure S14) highlights both consistencies and divergences compared to the all-trend results. Strong positive DCI for PET and AET is observed in most catchments during

spring, which aligns with the findings from all trends. In summer, PET shows strong agreement in central and southern regions. In contrast, Q and, TWS exhibit substantial disagreement across all seasons, with most catchments showing low concurrence among PET methods. Similar inconsistencies are observed for PET and AET in winter. Some central and eastern catchments consistently exhibit strong negative DCI for Q, and TWS in both spring and summer, reflecting similar patterns in the all-trend case. Applying statistical significance thresholds often shifts catchments from strong agreement (positive or negative) to disagreement, primarily because trends are no longer significant. Notably, no catchments switch from strong positive to strong negative DCI (or vice versa), confirming the consistency between statistically significant and all-trend results.

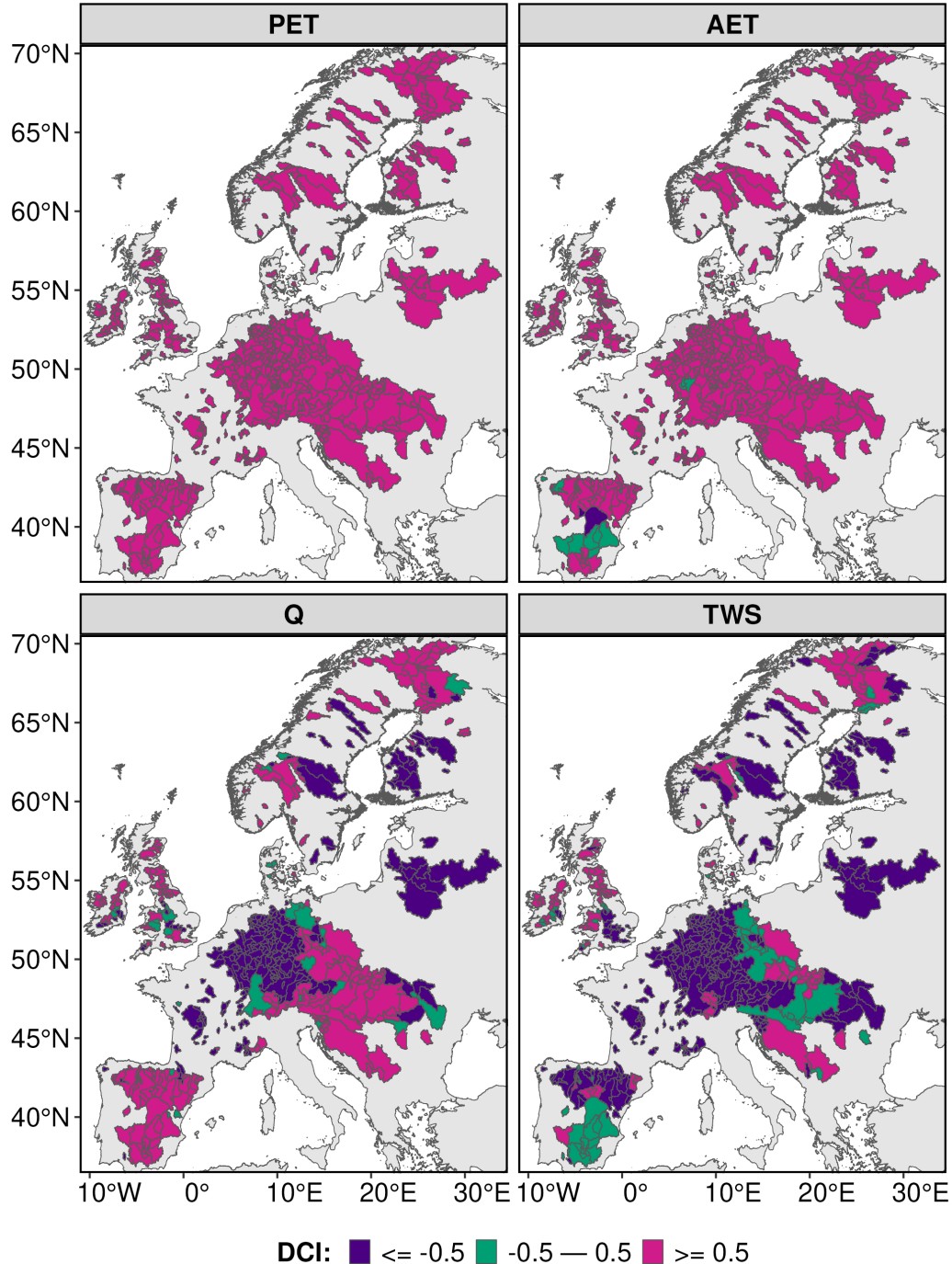

**Figure 4.** Spatial distribution of annual scale data concurrence index (DCI) for PET, AET, Q, and TWS. PET represents potential evapotranspiration, AET represents actual evapotranspiration, Q represents runoff at the outlet of the catchment and TWS represents total water storage.

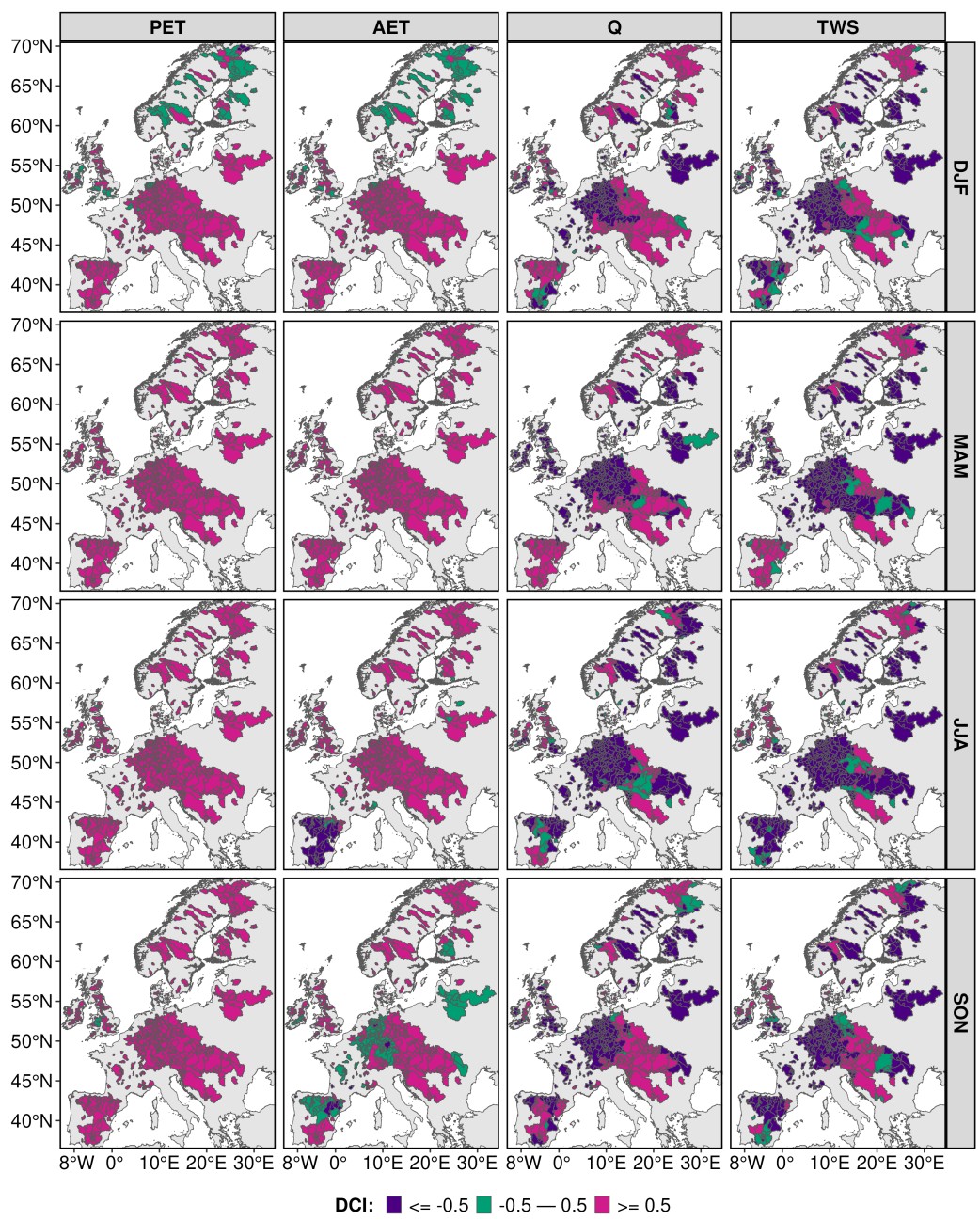

**Figure 5.** Spatial distribution of seasonal scale (winter (DJF), spring (MAM), summer (JJA), and autumn (SON)) DCI for PET, AET, Q, and TWS. Where DCI represents data concurrence index, PET represents potential evapotranspiration, AET represents actual evapotranspiration, Q represents runoff at the outlet of the catchment and TWS represents total water storage.

### 3.4 PET methods and patterns of hydrological cycle at annual and seasonal scales

In the previous section, we compared PET methods and their influence on individual hydrological components (P, AET, Q, and TWS), including agreement among PET methods. Here, we assess the influence of PET methods on patterns of key hydrological components across European catchments, identifying the most prevalent trend patterns where components (P, AET, Q, and TWS) concurrently increase or decrease. For instance, one pattern involves all components showing positive trends. Figure 6 summarizes these patterns, presenting the average and total number of catchments associated with each PET method. The analysis includes only the five most frequent patterns, covering the majority of catchments, excluding those patterns with minimal representation.

Most European catchments exhibit increasing trends across all hydrological components. The second most common pattern involves a decrease in P, Q, and TWS, with an increase in AET. Temperature-based PET methods generally account more number of catchments for these patterns than radiation and combinational methods, though exceptions exist, such as Baier-Robertson and Blaney-Criddle. Catchment distributions are spatially consistent across methods for these two patterns (Figure S15), with Blaney-Criddle showing the highest counts for patterns involving uniform increases or decreases. In case of statistically significant trends, only the all-positive pattern is prominent (Figure S16), with an average of 80 catchments. This pattern reveals notable differences between temperature-based and combinational methods. Overall, PET methods differ in the number of catchments assigned to each pattern; however, combinational methods consistently demonstrate similar catchment counts across most patterns.

Seasonal analyses reveal distinct PET method preferences across hydrological component patterns. During winter, spring, and autumn, the prevailing pattern involves decreasing P, Q, and TWS with increasing AET (Figures S17, S18, S20). In contrast, summer is characterized by increased P and AET, and decreased Q, and TWS. Substantial variation is observed among PET methods for each hydrological cycle pattern across all seasons. For instance, a pattern where all components exhibit positive trends, Baier-Robertson captures the fewest catchments, while Blaney-Criddle captures the most. The pattern reverses for the combination of decreased P, Q, and TWS with increased AET. Blaney-Criddle consistently represents the highest number of catchments in all-positive and all-negative patterns during spring, summer, and autumn. Combinational methods generally show stable catchment counts, whereas temperature-based methods exhibit greater variability. Statistically significant trends, however, are associated with very few catchments, and no single pattern dominates.

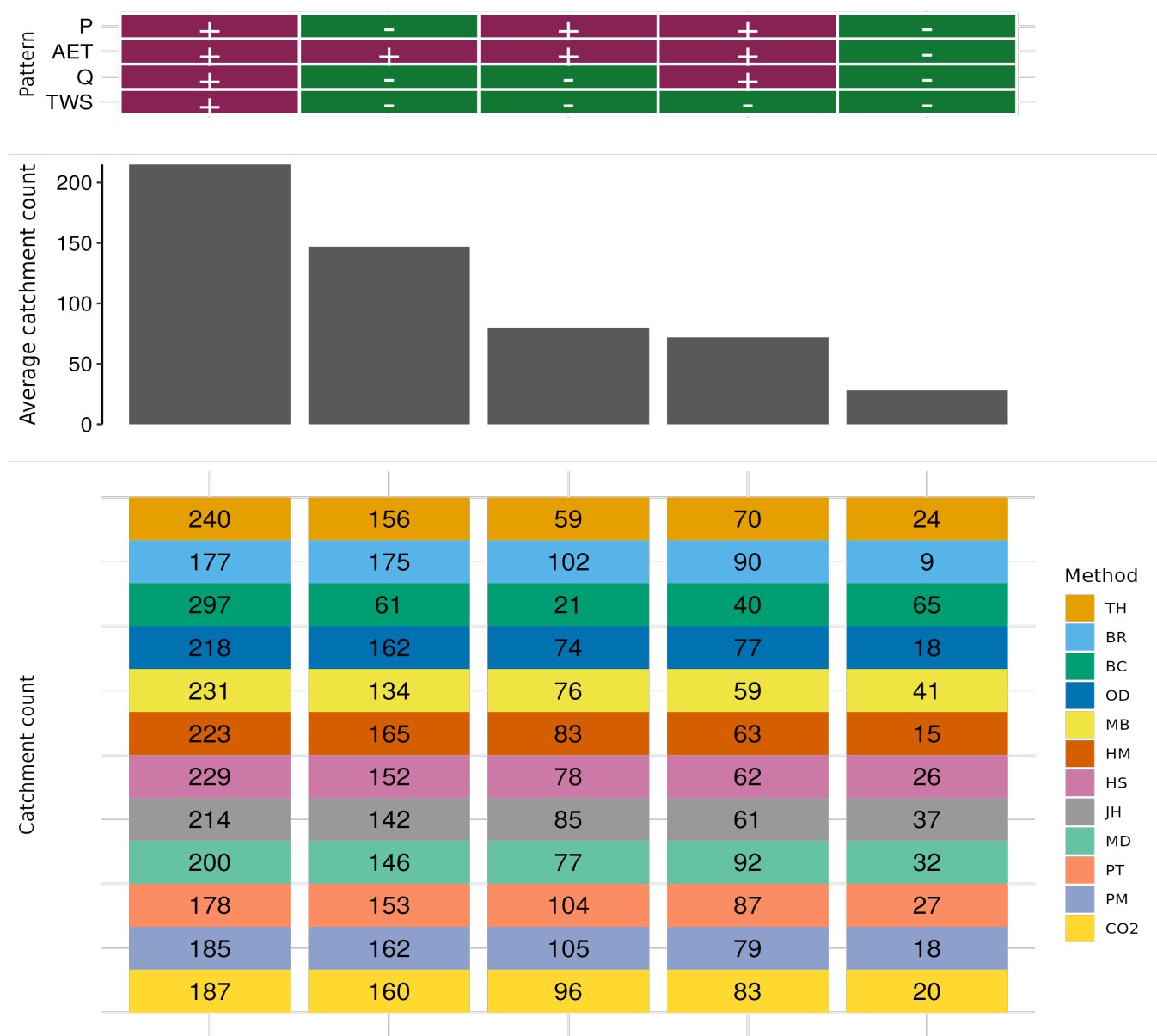

**Figure 6.** Pattern of different hydrological cycle components and the corresponding influence of PET methods on an annual scale. The first panel represents different patterns of hydrological cycle components. Each vertical column in this table corresponds to one pattern of hydrological cycle components. For example, the first column is filled with '+' signs, indicating that all hydrological components (P, AET, Q, and TWS) exhibit positive changes. The '+' and '-' signs denote positive and negative changes in the respective components. In the second panel, each bar represents the average number of catchments for each hydrological cycle pattern. The third panel shows the number of catchments associated with each PET method for the corresponding hydrological cycle patterns. The color of each cell represents a specific PET method, and each column aligns with the hydrological cycle pattern represented in the corresponding column of Panel One. Where P is precipitation, AET is actual evapotranspiration, Q is runoff, and TWS is total water storage. Abbreviations used for different PET methods are TH: Thornthwaite, BR: Baier-Robertson, BC: Blaney-Criddle, OD: Oudin, MB: McGuinness-Bordne, HM: Hamon, HS: Hargreaves-Samani, JH: Jensen-Haise, MD: Milly-Dunne, PT: Priestley-Taylor, PM: Penman-Monteith, $CO_2$: Penman-Monteith$[CO_2]$.

## 3.5 Relationship of PET and precipitation

Precipitation is an important component of the hydrological cycle. Figure 7 shows the changes in precipitation (P) without considering statistical significance. Annually, positive P trends dominate northern, western, southern, and southeastern catchments, while central Europe shows mixed positive and negative trends. In western Europe, a few catchments exhibit decreasing P trends (Figure 7A). Seasonally, southern catchments experience increased P in winter, spring, and autumn but declines in summer. Southeastern Europe shows consistent P increases across all seasons, while eastern Europe exhibits negative P trends in summer and autumn and positive trends in winter and spring (Figure 7A). P demonstrates a higher correlation with Q, and TWS across all catchment categories (Figure 7C). This suggests that P has a greater influence on Q, and TWS than PET. In energy-limited catchments, AET is mainly driven by PET. In mixed catchments, both P and PET contribute to AET. In water-limited catchments, AET is mainly influenced by P. When we consider statistically significant P trends, only 129 catchments show significant trends at the annual scale. Across seasons, the number of statistically significant catchments varies from 20 to 61 (Figure S21). Despite the limited number of statistically significant catchments, our findings regarding the influence of P and PET on AET, Q, and TWS remain consistent with all trends.

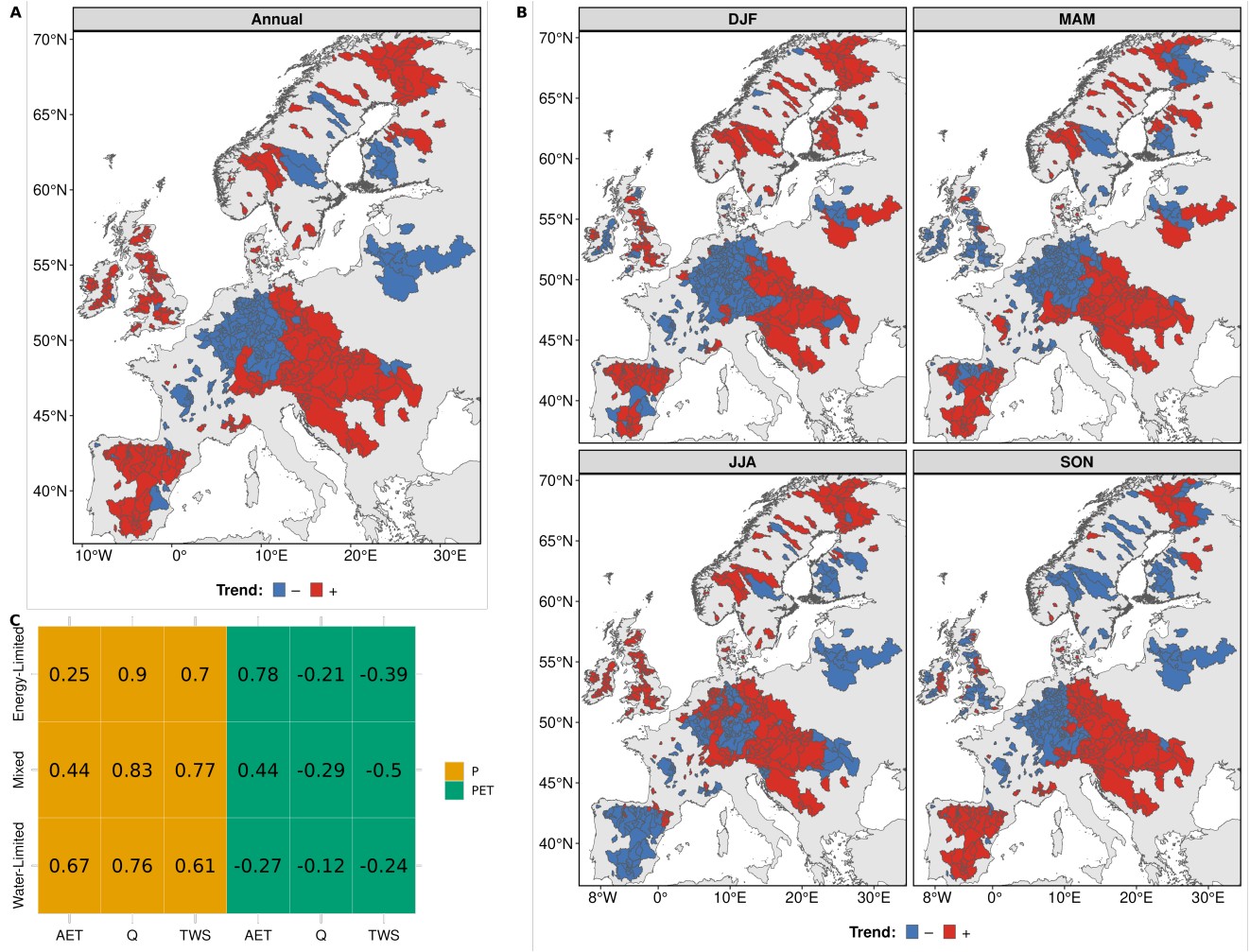

**Figure 7.** Spatial variation of precipitation (P) trends and their relationship with other hydrological components across catchment categories. Panel A shows the spatial distribution of increasing and decreasing annual P trends. Panel B illustrates the seasonal variation in increasing and decreasing P trends. Panel C represents the median correlation between P and AET, Q, and TWS, as well as between PET and AET, Q, and TWS, for each catchment category across all PET methods.

## 4 Discussion

### 4.1 Trends in P, PET, AET, Q, and TWS across Europe

PET methods consistently exhibit upward annual trends across European catchments (Figure 2), aligning with findings by
Anabalón and Sharma (2017), who reported similar increases using diverse PET datasets. Among the methods, Jensen-Haise
consistently produces the highest absolute values, a pattern observed in various regions by other studies (Kingston et al., 2009;
Hanselmann et al., 2024; Seiller and Anctil, 2016). This method relies on temperature and extraterrestrial radiation data, with
the latter remaining constant annually. Thus, the observed trends are primarily temperature-driven. Shi et al. (2023b) found that
Jensen-Haise trends align closely with Penman-Monteith, which is used as a benchmark for evaluating other PET methods. In
our study, we observe a notable distinction between the Jensen-Haise and Penman-Monteith PET methods, primarily driven by
differences in meteorological forcings. While temperature is the dominant factor in Jensen-Haise, Penman-Monteith is more
influenced by radiation, followed by temperature, vapor pressure deficit, and wind speed (Maček et al., 2018). Conversely, the
Milly-Dunne method consistently shows the lowest trends in water-limited catchments, with lower net radiation-driven trends
in southern Europe (Pfeifroth et al., 2018). Sensitivity analyses of each PET method with meteorological forcings are beyond
the scope of this research.

The AET trend is proportional to the PET trend in energy-limited and mixed catchments. In water-limited regions, there is
not enough water to evaporate, and it is mainly governed by available water (P) (Bruno and Duethmann, 2024). This results
in a notable decline in AET when compared to PET. Anabalón and Sharma (2017) similarly reported stronger correlations
between AET and PET in energy-limited regions, while AET in water-limited areas aligned more closely with precipitation.
Notably, their study did not differentiate between PET methods but relied on existing datasets. In our analysis, AET exhibits
the same directional changes as precipitation at both annual and seasonal scales in water-limited catchments (Figure 7). Despite
differences among catchment categories, PET methods demonstrate strong positive or negative agreement in AET trends.

Runoff trends vary with PET method selection in water-limited catchments but remain largely insensitive to PET methods
in energy-limited and mixed catchments. This aligns with previous findings (Bai et al., 2015; Oudin et al., 2005; Seiller and
Anctil, 2016), which reported insensitivity of runoff to PET formulations. This insensitivity is often attributed to hydrological
model calibration, where PET impacts are counterbalanced by parameterization. Notably, despite the absence of individual
PET method calibration in our study, runoff in energy-limited and mixed catchments remained insensitive to PET variation.
This is likely due to the strong correlation between precipitation and runoff trends (Figure 7c), which often outweighs the
impact of PET (Berghuijs et al., 2017; Anabalón and Sharma, 2017). The strong negative agreement among PET methods for
runoff in central European catchments is similarly related to their strong correlation with precipitation (Figure 7a).

Total water storage (TWS) appears to be insensitive in energy-limited and mixed catchments, while it exhibits variability
in the trend of different PET methods in water-limited catchments. Bai et al. (2016) observed that TWS in energy-limited
catchments is more strongly impacted by PET than in water-limited catchments, though their study focused solely on Chinese
catchments. More recently, Boeing et al. (2024) reported a decline in TWS over Germany, consistent with our findings that
TWS decreases in energy-limited and mixed catchments for all PET methods. In hydrological models, TWS compensates for

long-term changes, i.e., higher PET results in lower TWS in energy-limited catchments, whereas water-limited catchments are primarily governed by precipitation (Bai et al., 2016). This is in line of our findings on the opposite TWS trends relative to AET and a stronger agreement among PET methods in energy-limited and mixed catchments.

Precipitation generally increases across most catchments, with annual and seasonal patterns largely consistent. Exceptions occur in summer for the southern catchment and spring for eastern catchments, consistent with findings by Caloiero et al. (2018) and Markonis et al. (2019). Precipitation has a stronger influence on the hydrological cycle than AET in low and mid-latitude regions compared to higher latitudes (Zhang et al., 2019). This spatial distribution explains the stronger correlations observed in mixed and energy-limited catchments, predominantly located in mid and high latitudes. Globally, Q is more sensitive to P than PET, supporting the observed decline in Q despite rising PET across many catchments (Berghuijs et al., 2017).

## 4.2   Methodological sensitivity of PET estimation methods

PET methods vary in both absolute values and trends, even within the same category, due to structural and empirical differences in their formulations. For instance, while Jensen–Haise and Hargreaves–Samani both incorporate extraterrestrial radiation and air temperature, Hargreaves–Samani includes a diurnal temperature range term absent in Jensen–Haise. Thornthwaite, in contrast, relies solely on a heat index derived from monthly temperature. Among combinational methods, Penman–Monteith and its modified version differ by the inclusion of a $CO_2$ concentration term. Structurally similar methods like McGuinness–Bordne, Oudin, and Jensen–Haise diverge primarily due to empirically derived constants tailored to specific regions or climates (Proutsos et al., 2023). Moreover, catchment size does not significantly influence the overall findings related to PET and hydrological components (Figure S22), consistent with Tang et al. (2023), who observed that spatial averaging over larger catchments reduces uncertainty and enhances reliability.

Even though our study's experimental design varies from the global analysis of Pimentel et al. (2023), both utilize large-scale hydrological models at the basin scale. Pimentel et al. (2023) evaluated three PET methods to identify the optimal approach for estimating PET, AET, and Q across the Globe. Across Europe, they reported that Jensen-Haise in northern Europe (energy-limited), Hargreaves in central (mixed), and Priestley-Taylor alongside Hargreaves-Samani in southern Europe (water-limited) perform better to estimate PET. For AET estimation, Jensen-Haise remains the preferred method in northern Europe, with Hargreaves-Samani leading in central Europe and Priestley-Taylor in southern Europe. Priestley-Taylor was also deemed the most effective for runoff estimation. In contrast, our findings indicate a consistent distinction between Jensen-Haise and Penman-Monteith for PET across all catchment types. Similarly, for AET, Jensen-Haise consistently shows higher trends than Penman-Monteith in both energy-limited and water-limited catchments. For runoff, Priestley-Taylor and Penman-Monteith exhibit similar patterns in energy-limited and mixed catchments; however, in water-limited catchments, Priestley-Taylor yields more pronounced changes than Penman-Monteith.

## 4.3   Patterns of hydrological cycle components

When we look at the combination of changes among the hydrological cycle components (P, AET, Q, TWS) across European catchments, two dominant patterns of changes are observed: a water cycle intensification pattern of simultaneous increase

in all components and an aridification pattern of simultaneous decline in all except AET. These two patterns can be seen across over 60% of European catchments. However, when focusing only on significant trends, the pattern of intensification becomes more dominant, declining approximately to 15% (80 catchments) compared to 0.4% (2 catchments) aridification. The intensification pattern aligns with findings by Teuling et al. (2019), who reported rising P, AET, and Q in central-western Europe and declines in these components in the Mediterranean. Their analysis, based on the Penman–Monteith method, also indicated fewer catchments classified under this pattern compared to temperature-based methods, which show stronger responses. This is a vivid example of how PET method selection could amplify or dampen our estimates about hydrological cycle intensification.

On the contrary, the aridification pattern (decrease in P, Q, and TWS and increase in AET) suggests that water reserves are being depleted to sustain evapotranspiration, a mechanism particularly evident in water-limited regions. Bruno and Duethmann (2024) noted that rising atmospheric demand, without sufficient water supply, results in reduced Q, and TWS. Massari et al. (2022) similarly reported that increasing AET contributes to Q reductions in water-limited regions. Even with decreasing P and Q, continued declines in TWS appear to support increases in AET (Massari et al., 2022). For Europe, this is also very relevant to the compound warm season droughts that have been reported to increase since the beginning of the century (Markonis et al., 2021), as well as with the conditions that favor the onset and propagation of flash droughts (Shah et al., 2023). Since the evaporative demand is expected to further increase in the future (Rakovec et al., 2022), it is essential to acknowledge the uncertainties due to PET method selection.

## 4.4 Implications of PET method selection to hydroclimatic regime classification

The selection of PET methods significantly influences the hydroclimatic classification of catchments. Traditionally, catchments are categorized as water-limited or energy-limited based on the aridity index. However, our study introduces a third category, termed "mixed", which lacks a physical basis but highlights the critical role of PET method selection in defining catchment types. This is when, a PET method that consistently estimates higher values may shift a catchment from energy-limited to water-limited, whereas a method with lower estimates could reverse this shift. This underscores the importance of method selection, where variations in PET estimates can alter the hydroclimatic classification. Similarly, Zhang et al. (2016) introduced a less common classification termed "equitant", which applies a single PET method to calculate the aridity index. Such methodological differences can lead to inconsistencies in catchment classification. For instance, Kuentz et al. (2017) uses the Jensen-Haise method, Ajami et al. (2017) utilizes the Priestley-Taylor method, and Zhang et al. (2016) adopt the Penman method for aridity index estimation. In our analysis, excluding PET methods that consistently estimate higher PET values, such as Blaney-Criddle, Jensen-Haise, and McGuinness-Bordne, resulted in 42% of catchments transitioning from the mixed to the energy-limited category (Figure S23). Detailed catchment shifts based on various combinations are outlined in Table S1.

## 4.5 Limitations and future research

Our study comes with certain limitations that pave the way for future research. One key limitation is the uncertainty associated with the input data used to calculate PET methods. Previous studies have shown that temperature-based methods are sensitive to temperature, radiation-based methods to radiation, and combinational methods to multiple variables, including temperature,

radiation, relative humidity, and wind speed (Hua et al., 2020; Guo et al., 2017). In addition, we use a monthly time step, which tends to mask the influence of PET method selection on AET extremes. These extremes can behave differently from the mean state of AET, potentially leading to different implications for changes in hydrological cycle fluxes compared to those based on mean AET values (Markonis, 2025). Finally, we limited our analysis to one precipitation product to isolate the specific impact of the PET method. However, precipitation is widely recognized as the most sensitive meteorological input, with extensive studies highlighting its uncertainties (Mazzoleni et al., 2019; Markonis et al., 2024). Our findings also confirm its dominant influence over PET in certain catchment categories (Figure 7). This identifies a potential gap for exploring the combination of precipitation with PET for more accurate simulations of hydrological cycle components. Although we selected meteorological datasets with comparatively lower uncertainty, data quality, whether from observational or reanalysis sources, remains an issue for hydrological assessments. While our focus was on methodological comparisons among PET methods, future research could benefit from multi-source assessments to enhance the robustness and reliability of hydrological modeling.

Large-scale hydrological models, including mHM, typically rely on default parameterization. In mHM, these parameters were initially developed using German basins, as outlined by Samaniego et al. (2010) and Kumar et al. (2013a). Since then, mHM has been extensively tested across various basins and hydrological variables (Rakovec et al., 2016; Samaniego et al., 2019; Boeing et al., 2024). For instance, Rakovec et al. (2016) evaluated discharge simulations across 400 European catchments using 36 parameter sets, demonstrating consistent model performance regardless of parameterization, thereby reinforcing confidence in the model's reliability. Similar approaches have been applied in global water models; Beck et al. (2017) employed ensemble parameters derived from ten catchments, while Kumar et al. (2013a) tested default parameters from European basins across 80 American catchments with varying climatic conditions. While these studies demonstrate the robustness of default parameterization, investigating how PET-specific calibration affects hydrological trends could provide valuable insights for future research. Notably, the Hargreaves-Samani PET method, used in developing these parameters in mHM, demonstrated best model performance in this study but did not consistently stand out compared to other PET methods in trend analysis across hydrological components. Moreover, the study is confined to temperate European catchments, leaving a gap in assessing arid and tropical climates, where distinct patterns may emerge. While Penman-Monteith[$CO_2$], it did not exhibit substantial differences compared to the Penman-Monteith method, indicating the need for further exploration of this method. It would be interesting to assess their impact under changing climate conditions and their implications for trend assessments.

## 5   Summary and conclusions

Twelve PET methods were used to evaluate their impacts on changes in the components of the hydrological cycle using the mesoscale Hydrological Model (mHM). These methods were applied across 553 European catchments, which vary in size and include different European climate types. These catchments were classified as water-limited, energy-limited, and mixed catchments based on their aridity index. Changes in PET and hydrological components were assessed using Sen's slope for trend magnitude and the Mann–Kendall test for statistical significance. We analysed our results under two cases: first, using all trends; and second, using only statistically significant trends. To assess the agreement between different PET methods, we

used the data concurrence index for the period 1980 to 2019. The results demonstrate that the choice of PET method can substantially affect changes in AET, Q, and TWS, especially in water-limited and mixed catchments, with smaller changes and greater variability observed in water-limited catchments on an annual scale. Seasonal variations in changes and agreement between PET methods were also observed, as discussed in detail in sections 3.2 and 3.3. In general, there is agreement among the different methods that, since 1980, PET and AET are increasing over Europe, while runoff and total water storage exhibit

mixed fluctuations depending on the method used and the catchment latitude. The key findings of our study are summarized as follows:

1. PET is increasing across European catchments. The majority of the PET methods indicate a positive trend in all categories of catchments, but the increase rates differ among the methods employed.

2. At the annual scale for all trends, the Jensen-Haise PET method stands out by consistently showing the highest trends

for PET and AET across all catchment categories. The Milly-Dunne (energy-based) method is notable as the only one to exhibit a negative trend for water-limited catchments. Regarding Q, and TWS, the PET methods display different changes and variability, with no method consistently showing either the lowest or highest trends for these hydrological components. Conclusions remain consistent when considering only statistically significant trends.

3. The trend patterns between PET and AET are similar across all methods for the hydrological cycle components. However,

Q, and TWS do not exhibit the same pattern and appear to be less sensitive to choice of PET methods. Most PET methods agree on the trend direction for PET and AET, but in a few catchments, the trends for Q, and TWS show an opposite direction. The negative trends in Q, and TWS are primarily due to negative precipitation trends, which have a stronger impact on these components in all catchment categories.

4. At the seasonal scale, PET methods reveal different trends for PET, with no method consistently showing the highest or

lowest trends across all seasons. However, the Jensen-Haise method shows the highest PET trends during spring, autumn, and summer. AET trends follow a similar pattern to PET in all seasons. The PET methods show strong agreement on trend direction for central and southern European catchments, especially for PET, but there is less agreement for northern catchments in winter. Strong negative agreement is found for Q, and TWS in summer and spring, while disagreement is observed for AET in central and southern catchments during autumn.

5. The summer season contributes more to the annual PET trends than any other season across all catchment categories. Similarly, for AET, the summer season has a higher contribution to the annual AET trend in energy-limited and water-limited catchments. For runoff (Q), the spring season contributes more in mixed and water-limited catchments. For TWS, the spring season has a higher contribution in energy-limited catchments.

6. Overall, the magnitude of trends varied between PET methods for PET and the hydrological components (AET, Q, and

TWS). The use of a specific PET method in a hydrological model can notably affect studies focused on the hydrological cycle.

7. Precipitation primarily governs trends in all hydrological components and catchment types, except for AET in energy-limited catchments, which is largely influenced by the choice of PET variations.

8. The choice of PET method substantially influences hydrological patterns across European catchments on both annual and seasonal scales. Combinational methods generally account for fewer catchments than temperature-based methods in dominant hydrological patterns. This observation remains consistent for statistically significant trends as well.

9. In the case of statistically significant trends, the conclusions remain consistent with those from the all-trend analysis. The only difference is the reduced number of catchments, primarily omitted due to weaker trends.

Our research demonstrates the critical role of PET method selection and its implications for quantifying fluctuations in the hydrological cycle. Our findings reveal that two methods notably deviate from the others. Specifically, the Jensen-Haise method shows higher trend values, while the Milly-Dunne method exhibits lower trends in water-limited catchments. Consequently, we recommend exercising caution when applying these methods as they appear to be outliers. Despite these variations, the PET methods generally agree that atmospheric moisture demand is increasing across Europe, reflecting recent shifts in temperature and radiation. The observed variability in trend magnitudes emphasizes the importance of careful PET method selection to ensure robust and representative assessments of hydrological trends.

*Code and data availability.*

The necessary data required to reproduce the final analysis and figures are available in the zenodo repository (https://doi.org/10.5281/zenodo.14008649). Codes used to analyze the results are publicly available in the GitHub repository (https://github.com/imarkonis/ithaca/tree/main/projects/pet_europe).

*Author contributions.*

VT: conceptualization, PET calculation, hydrological model simulations, postprocessing, formal analysis, and writing (original draft), YM: conceptualization, supervision, writing (review and editing) RK: conceptualization, writing (review and editing), JRT: writing (review and editing), MRVG: writing (review and editing), MH: writing (review and editing), OR: setup of hydrological model, supervision, conceptualization, writing (review and editing).

*Competing interests.*

At least one of the (co)-authors is a member of the editorial board of Hydrology and Earth System Sciences.

*Acknowledgements.* This work was carried out within the project "Investigation of Terrestrial HydrologicAl Cycle Acceleration (ITHACA)" funded by the Czech Science Foundation (Grant 22-33266M). Computational resources were provided by the e-INFRA CZ project (ID:90254), supported by the Ministry of Education, Youth and Sports of the Czech Republic. VT was also funded by the Internal Grant Agency (Project no: 2023B0008), Czech University of Life Sciences. We sincerely thank editor Elena Toth, referee Franziska Clerc-Schwarzenbach, and other two referee for their valuable feedback and constructive suggestions, which significantly contributed to improving the quality of this manuscript.

## Appendix A:  Potential Evapotranspiration formulation

**Table A1.** Formulations of PET methods. Where $R_e$ is extraterrestrial radiation (MJ m$^{-2}$ d$^{-1}$), $\lambda$ is the latent heat of vaporization (MJ kg$^{-1}$), $\rho$ is water density (= 1000 kg m$^{-3}$), $d_a$ is air density (kg m$^{-3}$), $T_a$ is air temperature (°C), $T_d$ is dew point temperature (°C), $T_{\max}$ is maximum air temperature (°C), $T_{\min}$ is minimum air temperature (°C), $\Delta$ is the slope of the vapor pressure curve (kPa °C$^{-1}$), $\gamma$ is the psychrometric constant (kPa °C$^{-1}$), $e_s$ is saturation vapour pressure (kPa), $e_a$ is actual vapour pressure (kPa), u$_2$ is wind speed 2 m above the soil surface (m s$^{-1}$), $R_s$ is net short-wave radiation (MJ m$^{-2}$ d$^{-1}$), $R_n$ is net incoming solar radiation (MJ m$^{-2}$ d$^{-1}$), $G$ is soil heat flux (MJ m$^{-2}$ d$^{-1}$), $RH$ is relative humidity (%), DL is day length (h d$^{-1}$), $I$ is annual heat index, and $CO_2$ is carbon dioxide concentration (ppm). The following abbreviations are used throughout the manuscript to refer to the respective PET methods: TH (Thornthwaite), BR (Baier-Robertson), BC (Blaney-Criddle), OD (Oudin), MB (McGuinness-Bordne), HM (Hamon), HS (Hargreaves-Samani), JH (Jensen-Haise), MD (Milly-Dunne), PT (Priestley-Taylor), PM (Penman-Monteith), and CO$_2$ (Penman-Monteith[$CO_2$]).

| Method | Formulation | Reference |
|---|---|---|
| Hargreaves-Samani | $0.0023 \times \frac{R_e}{\lambda \times \rho} \times \sqrt{t_{max} - t_{min}} \times (t_{avg} + 17.8) \times 1000$ | Hargreaves and Samani (1985) |
| McGuinness-Bordne | $1000 \times \frac{R_e}{\lambda \times \rho} \times \frac{T_a + 5}{68}$ | McGuinness and Bordne (1972) |
| Hamon | $k \times 0.165 \times 216.7 \times \frac{DL}{12} \times \frac{e_s}{t_{avg} + 273.3}$ | Hamon (1961) |
| Oudin | $1000 \times \frac{R_e}{\lambda \times \rho} \times \frac{t_{avg} + 5}{100}$ | Oudin et al. (2005) |
| Baier-Robertson | $0.157 \times t_{max} + 0.158(t_{max} - t_{min}) + 0.109 \times R_e - 5.39$ | Bai et al. (2016) |
| Blaney-Criddle | $0.825 \times (0.46 \times t_{avg} + 8.13) \times \frac{100 \times DL}{365 \times 12}$ | Blaney (1952) |
| Thornthwaite | $16 \times \frac{DL}{360} \times \left(\frac{10 \times t_{avg}}{I}\right)^k$ | Thornthwaite (1948) |
| Jensen-Haise | $1000 \times \frac{R_e}{\lambda \times \rho} \times \frac{T_{avg}}{40}$ | Jensen and Haise (1963) |
| Priestley-Taylor | $\frac{1.26 \times \Delta \times (R_n - G)}{\lambda \times \rho \times (\Delta + \gamma)}$ | Priestley and Taylor (1972) |
| Milly-Dunne | $0.8 \times (R_n - G)$ | Milly and Dunne (2016) |
| Penman-Monteith | $\frac{0.408 \times \Delta \times (R_n - G) + \gamma \times \left(\frac{900}{T_{avg} + 273}\right) \times u_2 \times (e_s - e_a)}{\Delta + \gamma \times (1 + 0.34 \times u_2)}$ | Penman (1948) |
| Penman-Monteith[$CO_2$] | $\frac{0.408 \times \Delta \times (R_n - G) + \gamma \times \left(\frac{900}{T_{avg} + 273}\right) \times u_2 \times (e_s - e_a)}{\Delta + \gamma \times (1 + 0.34 \times (u_2 + 2 \times 10^{-4} \times ([CO_2] - 300)))}$ | Yang et al. (2019) |

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
