# Peer review of "Unveiling the impact of potential evapotranspiration method selection on trends in hydrological cycle components across Europe"

_Hydrology and Earth System Sciences, 2024_

## Referee Comment (RC1)

**Review of the manuscript "Unveiling the impact of potential evapotranspiration method selection on trends in hydrological cycle components across Europe", submitted to HESS by Vishal Thakur et al.**

**General comments**

Thank you for giving me the opportunity to review the manuscript "*Unveiling the impact of potential evapotranspiration method selection on trends in hydrological cycle components across Europe*" by Vishal Thakur and co-authors. I enjoyed reading this manuscript and consider the findings to be valuable and highly relevant: Even though potential evapotranspiration is a crucial component of the hydrological cycle, factors like data availability or convenience often play a role in the selection of a potential evapotranspiration formulation. There is no uniform way of dealing with potential evapotranspiration in hydrology, and often, different concepts (such as reference evapotranspiration and potential evapotranspiration) are used interchangeably. Therefore, studies like the one by Vishal Thakur et al., analysing the effects of different formulations of potential evapotranspiration are of great value to shed light on this often-neglected topic. The manuscript presents an analysis of the influence of different potential evapotranspiration formulations on different simulated components of the hydrological cycle. To do so, the authors make use of a large-scale modelling approach and analyse the modelling results with effective methods. They come to the important conclusion that the choice of a potential evapotranspiration method has an influence on the results when studying the hydrological cycle and its components.

My main concern is that the majority of the chosen formulations (or methods) are temperature-based. Since temperatures were rising in the studied period between 1980 and 2019, the potential evapotranspiration methods based on temperature do show a positive trend. So far, it is not clear if temperature-based methods are still reliable under the conditions of a warming climate (see for example studies on the so-called "pan evaporation paradox": Li et al. (2013, 10.1002/wrcr.20202); Wang et al. (2017, 10.1002/wat2.1207)). With eight out of twelve methods being temperature-based, a possible overestimation may strongly influence the results of this study.

Specific comments and suggestions that will hopefully help to improve the different parts of the manuscript are listed below.

**Specific comments**

*Introduction*

- While Thornthwaite (1948) was the first to introduce the term "evapotranspiration", according to Miralles et al. (2020, 10.1029/2020WR028055), the concept itself was already used earlier. This should be specified. Furthermore, as the different concepts regarding evaporation (including "reference crop evapotranspiration") are often used interchangeably in hydrology, I would suggest to add a statement about this problem, e.g., referring to the Miralles et al. paper mentioned above as well as the already cited paper by Xiang et al. (2020). In addition, I would argue that there should be some indication if you consider evapotranspiration to include interception or not in your study. See also the commentary by Savenije (2004, 10.1002/hyp.5563).
- In the first sentence of the paragraph starting at line 41, PET is stated to directly influence AET. In the second sentence, an alternative to this direct influence is presented. Therefore, the first sentence should be adjusted so that it becomes clear that this is just one possibility. Furthermore, depending on how interception is dealt with, this may be a necessary component to add in the list of fluxes that make up AET (in the second sentence of this paragraph).

*Methods and data*

- In the beginning of section 2.2, you give the study by Hersbach et al. (2020) as a reference for ERA5-Land. I would argue that the suitable reference there is the paper by Muñoz-Sabater et al. (2021), that you cite later in the manuscript. Furthermore, in line 97, there is a reference

missing for EM-Earth and SC-Earth, or it is not clear to me if the Tang et al. (2022) reference belongs to this. If the EM-Earth data are based on ERA5 data (and not on ERA5-Land data), I would suggest to include the reference to the Hersbach et al. (2020) paper there.

- I assume the AET, Q, and TWS data in Figure 1c to be simulated values. In my opinion, this needs to be mentioned in the figure caption as it only becomes clear after reading further.

- In line 122, you state that some of the temperature-based methods also use some extraterrestrial radiation term calculated based on latitude. Potentially, this could already be mentioned earlier, when the definition of the temperature-based category is given (i.e., it could become clearer that also radiation terms can be required and a method is still considered to be temperature-based).

- When you list the terms that are required for the Penman-Monteith and the Priestley-Taylor method, I would suggest to list all the terms (as it's only two equations), instead of giving some examples and concluding with "etc.". Alternatively, you could give one or two examples and then place a reference to the table where you list all the input data required for each method.

- For me, the description of the modelling part (paragraph starting on line 139) is hard to follow. Therefore, I ask you to give more information about this part of your study: You do one model run per catchment and PET method (553 x 12 = 6636). Is there any model calibration? If yes, please elaborate on that: What objective function(s) did you use? What was the spatial resolution? If no, where do you take the model settings from? Are these the same for all catchments? How do you make sure that the parameterization that you use matches your catchments? Did you think about equifinality and how other possible parameterizations could influence your results? How well did the model perform for the different catchments? In the very end of the discussion, the reader learns that a default parameterization was used. Please add this to the methods part and make sure that the questions listed above become clear.

- Related to the comment above, where does this default parameterization come from? Was there a method for potential evapotranspiration involved when this default parameterization was obtained? If so, may this have varying effects on the results based on this (or similar) methods (that can potentially profit from compensating effects) and the results based on different methods (that cannot profit from any compensating effects)? Potentially and if possible (may be limited due to data availability constraints) it would be interesting to compare the current results to the results of a calibrated model (i.e., calibrated for each of the different methods) – you also include a similar remark in the discussion. With additional calibrations that make use of the different methods, it could become clearer if the (assumed) use of one method for the default parameterization affects the results. If additional calibrated model runs are not possible, I would suggest you to include some text on the potential effects of the default parameterization in the discussion.

- I suggest you to clarify in lines 158 and 159 that you compare winter data to winter data, spring data to spring data, etc. so that it is immediately clear that you do not compare all four seasons with each other.

- In section 2.3.4, you describe the modification of the DCI by including also non-significant trends to be able to include all trend estimates. Are you sure that you are not including too much noise by also counting very weak positive and negative trends? I suggest to at least show that your findings do not change if only reasonably significant trends are considered.

*Results*

- In the first paragraph of section 3.1, you describe the different trends in PET for the three categories of catchments. As PET describes a potential, i.e., the maximum ET that could be achieved if there was no water-limitation, the trend in PET should not depend on if a catchment is water-limited or energy-limited. Thus, I suggest to formulate this paragraph differently, such that it is clear that the PET trend is influenced by other factors, but not by the availability of water.

- In the last paragraph of section 3.2, you suddenly write about December and not about the winter season (line 234). If you did monthly analyses, this needs to be stated clearly in the methods section. Otherwise, December may be the wrong term here.
- As already mentioned in the general comments: With the rising temperatures, all the temperature-based methods tend to show positive trends. Since most PET methods included are temperature-based, they have the largest influence on the DCI. It would be good to test if this unequal distribution of the different types influences your results.
- To make the part about the combinations of hydrological cycle component changes (section 3.4) easier to understand also before studying Figure 6, I suggest you to add an example of a "possible combination of hydrological cycle component changes" after you introduce this term on lines 269 and 270. Similarly, it may be good to inform the reader what you mean with "the first five hydrological cycle combinations" (lines 270 and 271).
- In lines 272 and 273 you state that the temperature-based methods account for more catchments with positive trends across all hydrological cycle components than combinational methods (and you put this in comparison to the Blaney-Criddle method leading to positive trends in all components for most catchments). This statement is not clear: Do you want to say that the temperature-based methods lead to the "all positive" combination in more catchments than the combinational methods? If so, this is not true for the Baier-Robertson method, if I interpret Figure 6 correctly. Please reformulate this statement and reconsider if this should be formulated as a comparison to the first part of the sentence.
- The sentence regarding the combinations with AET+ and TWS- (lines 280 and 281) should come before the statement about the last five combinations to be consistent with the order in Figure 6.
- I think that Figure 6 should be improved so that it can be grasped more quickly and easily:
  - I suggest you to put the lower part (in the following called "combinations") to the top and the upper part (in the following called "catchment count") to the bottom. The reader first wants to know what combination we are looking at, and then wants to look at the results for this combination. I see that you used the "combinations" part as an "axis label", but this is not so clear when looking at the plot in the beginning as it looks more like two different parts.
  - For the "combinations", consider displaying them differently. For example, a simple table-like graphic in which you could even work with additional colours would in my opinion be easier to interpret than the current way to display the combinations, see below. If you decide to do so, do not forget to change the caption.

    | PRE | + | - | + | + | - | - | - | + | - | - |
    |-----|---|---|---|---|---|---|---|---|---|---|
    | AET | + | + | + | + | - | - | + | + | + | - |
    | Q   | + | - | - | + | - | - | - | - | + | + |
    | TWS | + | - | - | - | - | + | + | + | + | + |

  - For the "catchment count", I think it would improve the readability of the figure if not all bars were the same height, i.e., if the height of the bar would decrease from left to right, proportionally to the higher number of catchments that are contained in the bars on the left than in the ones on the right. This way, it would become clearer which combinations occur how often. For that, you could consider using horizontal instead of vertical bars to gain more space (this would also allow you to use the "combinations" part as "axis labels", but then of the vertical axis).
  - Please double-check the catchment count for each method. To my understanding, the number of catchments per method contained in this plot should always sum up to 553. However, for example for the Thornthwaite method, the number of catchments only sums up to 240+156+59+70+24+2=551.
  - As the Thornthwaite method and the Blaney-Criddle method are displayed in neighbouring fields for the seventh combination, please consider not using red and green for

these two methods as they are hard to distinguish for colour-blind people. Alternatively, please use patterns in addition, to make it possible to distinguish the two colours.

*Discussion*

- In line 302, you compare your results to the ones in the study by Hanselmann et al. (2024). Please note that while the authors of this study are affiliated in Poland, the research has been conducted for Spitsbergen.
- In the paragraph starting on line 336, you discuss your results as well as the results of the study by Teuling et al. (2019). Based on the Teuling et al. paper as well as based on the PET methods that you considered in your study, I assume that you mean the Penman-Monteith method in this paragraph when you write about the Penman method. Please double-check and correct. The same issue occurs again in line 384.
- Later in the same paragraph, where you discuss a possible drying hydrological cycle, I think that a discussion of the study by Milly & Dunne (2016) that you refer to elsewhere in your manuscript is lacking as they studied this topic.
- In line 368, you write about methods that consistently overestimate PET. It is unclear to me how you define overestimation here: Do you just consider the methods leading to the highest estimates to be overestimating PET? If so, I would not consider it to be surprising that the catchments shift to the energy-limited category (as all the higher estimates are excluded). If all the low estimates would be excluded, the catchments would probably shift to the water-limited category. Please elaborate more on what you mean there.
- When you discuss the limitations of your study, you state that on the one hand, the PET methods were found to be sensitive to the input data, and on the other hand, that hydrological models are sensitive to the precipitation input. Based on this, I think it would be good to write something about the data quality of the input data that you used (as you basically state that your results are highly sensitive to these data).

*Summary and conclusions*

- In the very last part of the summary, you recommend an ensemble of PET formulations instead of one single formulation. While agreeing with this statement, I think that it should be supported by the study and not only occur in the end without being mentioned before. Thus, I suggest you to use the ensemble of the different methods as a thirteenth option of the PET calculation in your manuscript.

*Appendix*

- Please add the references from where you obtained the formulations for the different methods to Table A1.

**Technical corrections**

- The words of the title should not be capitalized.
- In the abstract, the third category is called "combination type", later it is called "combinational type". Please improve for consistency.
- Please note that "ERA5-Land" is written with a capital L (not "ERA5-land"). This is not consistently correct throughout the manuscript.
- In line 75, you specify two of the components again after the abbreviations have often been used in the preceding introduction. Either, leave this away, i.e., just give the abbreviations, or give the full names of the components plus the abbreviations for all three components consistently. Furthermore, I think the sentence starting at the end of this line is redundant (stating the same as the preceding sentence in other words), please double-check and correct.
- On line 117 there is a full stop after "estimation", but then the sentence seems to continue. Please double-check and correct.

- For the McGuiness-Bordne method, first occurring on line 119, you use different ways of spelling. To my knowledge, "McGuiness-Bordne" is the correct spelling, please double-check and adjust accordingly (do not forget figures and captions). Similarly, you don't spell "Baier-Robertson" and "Milly-Dunne" consistently (you sometimes write "Bair-Robertson", and in Table A1 "Milley-Dunne"). Please make sure that you use the correct spelling throughout your manuscript, including also the supplementary material (occurrence of "Milley and Dunne" in text S2). For all methods consisting of two last names, decide if you want to use a hyphen, a space, or an "and" to connect them.
- Double-check the references given in the tables: For example, in Table 2, the reference for the Hargreaves-Samani method is in a different style than the other references.
- It is mathematically problematic to use several letters for the same variable in an equation, e.g., "ND" could be interpreted as "N times D". Please consider reformulating.
- The first sentence of the results (line 175) is a repetition of the methods, please consider deleting.
- In the paragraph starting on line 184, it is not fully clear that the trends that are described are AET trends. I suggest you to formulate this clearer, for example by rewriting the second sentence to "…all PET methods lead to a positive AET trend in terms of median values." Similarly, in the following paragraphs describing the trends in Q and TWS (and in the results section in general), make sure that it is always clear that it isn't the PET methods that have trends but the PET methods that induce trends on the different components (if I understand your statements correctly).
- In line 206, you state: "…but the overall pattern of PET methods matches well with PET in all categories." This statement is unclear, did you mean a good match of PET with AET?
- In the caption of figure 3, you use mm seas$^{-1}$ year$^{-1}$ as a trend unit. I assume that 1 mm seas$^{-1}$ year$^{-1}$ in AET means that each year, 1 mm more goes to AET during the season of interest. As this seems to be an unusual unit (at least for me), I suggest you to explain this in the methods section of your manuscript already.
- In line 2019, the sentence starting with "AET in the summer season" is unclearly formulated.
- The references in lines 309, 310, 313 are not formatted correctly.
- In the supplementary material (Text S1), make sure that each excerpt starts on a new line and that there is always a space between the colon after the study name and the excerpt itself. Furthermore, look for typos in the copied excerpts (e.g., in the second excerpt from the Anabalón & Sharma study, you are missing a lot of spaces). For the Shi et al. (2023) paper, it would be good to indicate which paper you mean (a or b).
- Please include the methods' abbreviations in the caption of Table S1.
- I assume that Figure S1 and Figure S4 show the winter season, please double-check and correct the captions.
- In Figure S4, some of the numbers in the plot are not readable. Please see suggestions to improve Figure 6 of the manuscript (that also apply for all the other figures of this type). This should solve the problem. However, make sure that the different numbers are not written on top of each other.
- For Figures S10 and S11, the correct axis labels would be "Trend in … […]" as the data are showing the trend and not the value of a certain component.

**Individual typos**

- Line 9: comma missing between "water-limited" and "and"
- Line 13: space between closing parenthesis and comma needs to be deleted
- Line 32: word missing before the comma, probably "components"
- Line 37: should probably be "in hydrological models", rather than "in the hydrological model"
- Line 42: comma missing between "runoff" and "and"
- Caption of Table 1: "Short wave" and "longwave" radiation should be spelled consistently
- Line 133: space missing between "size" and the opening parenthesis

- Table 2: hyphen between "Penman" and "Monteith" is missing for the last method listed
- Line 141: "the" in the beginning of the line needs to be deleted
- Line 124: comma missing after "canopy interception"
- Line 160: parentheses around reference missing
- Line 162: comma missing between "significance" and "i.e."
- Line 179: full stop missing after "Penman-Monteith[$CO_2$]"
- Line 228: "and" instead of "are"
- Caption of Figure 2: should be "categories of catchments" (not "categories of catchment"); the same typo occurs in the captions of the other plots of this type
- Line 242: "a strong agreement" (not "an strong agreement")
- Line 252: comma missing between "Q" and "and"
- Caption of Figure 6: comma missing between "runoff" and "and"
- Line 279: "components" missing after "cycle"
- Line 419: "The" should not be capitalized
- Line 431: "of" missing between "use" and "an"
- Caption of Figure S12: should be "PET, AET, Q, and TWS" instead of "PET, AET, Q and, TWS". Unit should be mm seas$^{-1}$ year$^{-1}$ instead of mm/seas/year.

---

## Author Comment (AC1)

Community #1 Miyuru Gunathilake

Referee's comments are in black text

Authors's response are in blue text

1) General Comment: The manuscript by Thakur et al. 2024 is well written. The methodology is clear and robust. The authors used the mesoscale Hydrological Model (mHM) to simulate water balance components of 550+ catchments across Europe under diverse climatic conditions. The outputs offer valuable insights to the scientific community.

We would like to express our deep thank to Community #1 Miyuru Gunathilake for constructive feedback, which help in improving the quality of our manuscript. We sincerely appreciate the time and effort invested in providing such a thorough review.

2) Minor Comments: There are some minor comments which the authors could incorporate to further enhance the readability.

 a) To carry out statistical tests (Mann-Kendall etc.) the data distribution should follow certain criteria(s). (For instance, normality etc.). Have you checked for this?

Yes, we agree with you that certain tests require a specific type of distribution. However, in this case, we estimate the magnitude of the slope using the Sen's slope method. Due to its non-parametric nature, it does not require normality.

 b) The description under "2.3.2 mesoscale Hydrological Model (mHM)" could be moved to the Appendix.

We also received insightful feedback from the other three Referees on this section. We prefer to keep it in the main text, as it is an important part of our research.

 c) In the Abstract it is mentioned that "The findings reveal that the Jensen-Haise method produces the highest trends for PET on both annual and seasonal scales (summer, spring, and autumn)".

What did you mean by "highest"? "Magnitude" wise or in terms of the "Significance" of the trend? Please be clear.

We acknowledge your comment. We intended to say "highest trend magnitude". We will revise this sentence for better clarity.

d) Please check the manuscript for spacing. In some instance you have double spaces after the full stop.

Thankyou very much for your comment. We will correct it in revised manuscript.

---

## Author Comment (AC2)

Referee #3

Referee's comments are in black text

Authors's response are in blue text

1) Overall evaluation

The authors presented a study on the impact of 12 PET formulations on the trend of a set of components of the hydrological cycle in 553 catchments across Europe. They used a large-scale rainfall-runoff model to simulate actual evapotranspiration (AET), total water storage (TWS) and runoff (Q) multiple times by varying the PET forcing according to the 12 selected methods. Then, they analysed the annual and seasonal trend of PET, AET, TWS and Q obtained thought the different PET methods. They concluded that the choice of PET formulation influences the components of the hydrological cycle.

The work has a strong potential and the issue is of great interest in the field of catchment hydrology. In addition, this experiment could help fill a gap in the literature, which currently lacks a clear understanding of the effects of different PET formulations on rainfall-runoff modelling. However, I have few major concerns, especially about the methodological approach, which I think should be addressed in order to enhance the reliability of the results, facilitate and improve their interpretation, and meet the standards required for publication in HESS.

We sincerely thank Referee #3 for constructive feedback, which helps to improve the quality of our manuscript. We sincerely appreciate the time and effort invested in providing such a thorough and insightful review.

Most of my concerns were already highlighted in detail by the other two referees. Therefore, I would focus exclusively on the most critical issues, which need significant improvements.

2) General comments

   a) Modelling framework and model accuracy

A more detailed description of the modelling framework is certainly needed in order to better understand the experiment and its results. Please provide information about model spatial and temporal resolution, model calibration (or previously calibrated model settings) including objective function(s), calibration/validation period, input data used, etc. If a default parameterisation is used, as stated in the very last part of the manuscript, I believe the authors should elaborate about it and its impact on the outcomes of the analysis (i.e. can it be reliable?). In general, I suggest providing a brief overview about model performances against observed streamflow (which I suppose were used somehow for model parameterisation and/or to evaluate the default parameterisation) across the study catchments. I am aware that's definitely not the focus of the study but, since the entire analysis is based on a set of model outputs (streamflow included), I believe it is important to verify (and show) model accuracy in order to consolidate the interpretation of the results and draw solid conclusions. In fact, even if on one hand good model accuracy in reproducing streamflow does not guarantee a faithful reproduction of other hydrological components, on the other hand I would tend not to rely on the state variables of a poorly performing model. Maybe you can mention about model performance in the text and report the details in the Supplement.

Finally, I agree with referee Franziska Clerc-Schwarzenbach that, if a method for potential evapotranspiration was involved in the model parameterisation, authors should provide details about it and comment about the potential effect it could have on the outcomes of the experiment.

We thank reviewer for providing constructive comments. The detailed model setup and and its performance is discussed Referee #1 comment, in Section 3) Methods and Data, in subsections (e) and (f). The discussion is as follows: For each basin, we performed 12 model runs, with each run corresponding to one PET method. Therefore, for the 553 catchments, the total number of model runs is 6 636 (553 × 12). The mHM model was run at a daily time step with a spatial resolution of $0.125° \times 0.125°$ grid resolution. We did not perform any model calibration in our study. We used

the default model's parameterization because we wanted to mimic how large-scale/global hydrologic models performed, as if they would be employed across continents or global scale. The basin-wise setup used here, enabled us to estimate corresponding river discharge, and quantify all components of the water balance equation. The default parameterization of mHM has been shown to perform well in previous studies (Kumar et al., 2013; Rakovec et al., 2016). Furthermore, it has been demonstrated as one of the best-performing configurations compared to other large-scale hydrological models (Samaniego et al., 2019). For instance, Samaniego et al. (2019) compared the performance of mHM with other hydrological models across 357 catchments. Their results showed that the median Kling–Gupta efficiency (KGE) for mHM was approximately 0.6 across these catchments. To address reviewer's concern regarding the model's performance, we conducted an evaluation of its performance against discharge across the basins, as presented in Figure 1. Overall, the model performed well, with median KGE values ranging from 0.6 to 0.75 for most PET methods. However, the Blaney-Criddle method showed a median KGE slightly higher than 0.3, which was lower compared to other methods.

The default parameterization came from the model developers, and it was originally established over a diverse set of German basins, in the pioneering work of Samaniego et al. (2010). Since then, mHM has become a well-established model that has been extensively evaluated across various basins and hydrological variables (Rakovec et al., 2016; Samaniego et al., 2019; Boeing et al., 2024). For example, Rakovec et al. (2016) analyzed the model's performance across 400 European catchments. Their evaluation compared mHM's discharge simulations using 36 different parameter sets and found that the model's performance was consistent regardless of parameterization. Introducing new model setups or performing additional calibration and comparative analyses is beyond the scope of this study. We agree that the calibration aspect is important and offers interesting insights. However, we prefer to explore it in our future research.

Additionally, we will discuss the potential effects of model calibration in the discussion section to provide further context on this limitation in our manuscript.

[Figure]

**Figure 1.** Evaluation of the hydrological model (mHM) performance using simulated monthly streamflow across 553 catchments, forced with EM-Earth meteorological data. The figure presents the cumulative frequency distributions of the Kling–Gupta efficiency (KGE) and its three components: correlation (r), variability ratio (alpha), and bias ratio (beta), providing insights into the model's performance across different PET methods.

b) Trend analysis

First of all, I am sorry to say that the trend analysis is lacking. In particular, authors computed and took into account exclusively the non-parametric Sen's slope test, which estimates the magnitude of the trend of a time series but does not ensure its statistical significance. To affirm that a signal has a trend, it must be statistically significant. Therefore, I ask to the authors to complete the trend

analysis by associating a significance test (e.g. Mann-Kendall) to each trend magnitude (Sen's slope) and, consequently, change all the results and their interpretation accordingly.

In addition, I suggest excluding (maybe adopting a threshold) very week positive/negative trends when computing DCI, which may include a lot of noise and mask some aspects of your results.

These are reasonable concerns, also raised by other two Reviewers, as the inclusion of weak trends can indeed introduce noise to the analysis. considering all trends allows for an assessment of the spatial consistency of directional changes, which is a key observation in our study. If these weaker trends were purely random noise, their distribution would be approximately symmetric, with equal numbers exhibiting positive and negative changes. In contrast, the fact that the weaker trends predominantly align in the same direction suggests that, while they may not meet conventional significance thresholds (e.g., p = 0.05), they are not statistically irrelevant. The overwhelming consistency in their direction suggests a potential underlying signal rather mere stochastic variability. Overreliance on statistical significance can lead to rejecting meaningful patterns simply because they do not meet an arbitrary threshold, due to low variability rather than the absence of a real effect.

We also respectfully disagree that in order to talk about a trend it needs to be significant. This is a common misconception in time series analysis, which has risen a lot of criticism in many scientific disciplines. We think the best example is the milestone Editorial in the American Statistican by Ronald L. Wasserstein and Lazar (2019). Among many suggestions about the correct use of statistical significance they state that "no single index should substitute for scientific reasoning" and caution against the rigid use of p-values as an absolute determinant of scientific conclusions.

Still, we recognize the importance of showing how trend filtering, i.e., statistical significance testing, affects our results. Therefore, we will maintain our original analysis, which includes all trends, while complementing it with the analysis of significant trends in the supplementary

material (Figure 2 & Figure 3 shows DCI considering significant trend with p value 0.05). Additionally, we will articulate more clearly the reasons for our decision and describe the impact of trend significance in the revised manuscript.

[Figure]

**Figure 2.** Spatial distribution of annual scale data concurrence index (DCI) for PET, AET, Q, and TWS by considering only significant trend at 95% significance level. PET represents potential evapotranspiration, AET represents actual evapotranspiration, Q represents runoff at the outlet of the catchment and TWS represents total water storage.

[Figure]

**Figure 3.** Spatial distribution of seasonal scale data concurrence index (DCI) for PET, AET, Q, and TWS by considering only significant trend at 95 % significance level. PET represents potential evapotranspiration, AET represents actual evapotranspiration, Q represents runoff at the outlet of the catchment and TWS represents total water storage.

c) Results and discussion

I personally find some parts of the results section very hard to follow. In particular, please consider reviewing the text on seasonal trends (Section 3.2) and on combination of hydrological cycle components (Section 3.4).

We will revise it with better clarity in our manuscript.

In addition, when commenting DCI outcomes in Figure 4 and 5, authors refer to Northern/central/Southern Europe to develop the description. It would be useful to be more specific, because sometimes the text is misleading. For instance, at line 256 you state "...Q shows a strong decreasing trend for all PET methods in most central European catchments" but if I look at figure 5, central-Eastern DCI for Q are mostly negative. Is eastern Europe not included in "central"? If so, comment also about Eastern Europe. Again, Great Britain is considered Northern or central Europe.

We agree with you. we will clarify it further in our revised manuscript.

Figure 6 is not intuitive and difficult to interpret (and must be revised since some of the PET methods don't sum 553?). I strongly agree with the suggestions of Franziska Clerc-Schwarzenbach and Anonmymous referee #2. Also, the figure format and meaning should be explained in detail in the text before commenting it. Moreover, I suggest adding maps of the catchments coloured accordingly to the obtained combinations (or at least some of them), in order to be able to locate basins in space.

We will update the figure as we described in Referee #1 comments section 4) Results and subsection g).

The trends of AET, TWS and Q are strongly influenced not only by the PET method but also by PRE trends. Even if it is obvious, I would report PRE trends (and their significance) in the results (and not only in the Supplement) and use it to justify the trend direction of the other components.

Thank you very much for your comment, we will add it in the result section. We will update it in revised manuscript

Finally, the discussion about the obtained combinations of hydrological cycle components is poor. I believe it should be extended.

Thank you for your comment. A similar point was also raised by Referee #2. We will further elaborate on the discussion in this paragraph in our revised manuscript.

3) Additional minor comments

a) Figure 1b: I would specify in the text (not only in the caption) that the example refers to the catchments with bolder black contours in panel a. In addition, I would avoid interpolating the points: please use just dots of different colour.

Thank you very much for your comment. The interpolation lines in Figure 1b have been removed and are now represented as colored points for each catchment category (Figure 4). We will revise the text as suggested by the Referee in the revised manuscript.

[Figure]

**Figure 4.** Catchment classification to energy-limited, mixed, and water-limited categories. a) Catchment locations; black borders indicate a representative catchment of each category. b) Classification example within the Budyko space for the representative catchments. c) Annual time series of simulated hydrological components from the mesoscale hydrological model for each representative catchment and PET estimation method (TH: Thornthwaite, BR: Bair-Robertson, BC: Blaney-Criddle, OD: Oudin, MB: McGuinness-Borden, HM: Hamon, HS: Hargreaves-Samani, JH: Jensen-Haise, MD: Milly-Dunne, PT: Priestley-Taylor, PM: Penman-Monteith, $CO_2$: Modified Penman-Monteith accounts $CO_2$.). All units are in mm year$^{-1}$.

b) line 95: Please give some information about time coverage of the datasets, which I guess can justify your following choice regarding the simulation period.

Thank you for your comment. We have included the record length along with the temporal and spatial information for each dataset in Table 1 of the manuscript.

c) lines 103-104: I perfectly understand this choice, since ERA5-Land precipitation and temperature are known to be often not accurate, leading to a degradation of model performances. However, since one may wonder why not all variables from ERA5-Land are used, I would refer to recent studies highlighting such issues (e.g Clerc-Schwarzenbach et al. 2024, Tarek et al. 2020)

We agree with the Referee's comment, which has also been highlighted by the other two Referees. PET estimation and hydrological modeling are highly dependent on input data quality. The EM-Earth dataset provides high-quality precipitation and temperature data and has been shown to perform well over Europe (Tang et al., 2022). It has undergone climatology-based bias correction and accounts for precipitation undercatch. However, since EM-Earth does not include all necessary variables for PET estimation, we utilize ERA5-Land as a complementary dataset. ERA5-Land has been demonstrated to perform better than other reanalysis datasets, including ERA5 and ERA-Interim (Muñoz-Sabater et al., 2021). Several recent global studies follow a similar strategy, combining precipitation and temperature from EM-Earth with radiation, wind speed, and other meteorological variables from ERA5-Land (Tang et al., 2023; Yin et al., 2024).

**References**

Boeing, F., Wagener, T., Marx, A., Rakovec, O., Kumar, R., Samaniego, L., and Attinger, S.: Increasing influence of evapotranspiration on prolonged water storage recovery in Germany, Environmental Research Letters, 19, 024 047, https://doi.org/10.1088/1748-9326/ad24ce, 2024.

Kumar, R., Livneh, B., and Samaniego, L.: Toward computationally efficient large-scale hydrologic predictions with a multiscale regionalization scheme, Water Resources Research, 49, 5700–5714, 2013.

Muñoz-Sabater, J., Dutra, E., Agustí-Panareda, A., Albergel, C., Arduini, G., Balsamo, G., Boussetta, S., Choulga, M., Harrigan, S., Hersbach, H., Martens, B., Miralles, D. G., Piles, M., Rodríguez-Fernández, N. J., Zsoter, E., Buontempo, C., and Thépaut, J.-N.: ERA5-Land: a state-of-the-art global reanalysis dataset for land applications, Earth System Science Data, 13, 4349–4383, https://doi.org/10.5194/essd-13-4349-2021, 2021.

Rakovec, O., Kumar, R., Mai, J., Cuntz, M., Thober, S., Zink, M., Attinger, S., Schäfer, D., Schrön, M., and Samaniego, L.: Multiscale and Multivariate Evaluation of Water Fluxes and States over European River Basins, Journal of Hydrometeorology, 17, 287–307, https://doi.org/10.1175/JHM-D-15-0054.1, 2016.

Ronald L. Wasserstein, A. L. S. and Lazar, N. A.: Moving to a World Beyond "p < 0.05", The American Statistician, 73, 1–19, https://doi.org/10.1080/00031305.2019.1583913, 2019.

Samaniego, L., Kumar, R., and Attinger, S.: Multiscale parameter regionalization of a grid-based hydrologic model at the mesoscale, Water Resources Research, 46, https://doi.org/10.1029/2008WR007327, 2010.

Samaniego, L., Thober, S., Wanders, N., Pan, M., Rakovec, O., Sheffield, J., Wood, E. F., Prudhomme, C., Rees, G., Houghton-Carr, H., Fry, M., Smith, K., Watts, G., Hisdal, H., Estrela, T., Buontempo, C., Marx, A., and Kumar, R.: Hydrological Forecasts and Projections for Improved Decision-Making in the Water Sector in Europe, Bulletin of the American Meteorological Society, 100, 2451–2472, https://doi.org/10.1175/BAMS-D-17-0274.1, 2019.

Tang, G., Clark, M. P., and Papalexiou, S. M.: EM-Earth: The Ensemble Meteorological Dataset for Planet Earth, Bulletin of the American Meteorological Society, 103, E996–E1018, https://doi.org/10.1175/BAMS-D-21-0106.1, 2022.

Tang, G., Clark, M. P., Knoben, W. J. M., Liu, H., Gharari, S., Arnal, L., Beck, H. E., Wood, A. W., Newman, A. J., and Papalexiou, S. M.: The Impact of Meteorological Forcing Uncertainty on Hydrological Modeling: A Global Analysis of Cryosphere Basins, Water Resources Research, 59, e2022WR033 767, https://doi.org/10.1029/2022WR033767, 2023.

Yin, Z., Lin, P., Riggs, R., Allen, G. H., Lei, X., Zheng, Z., and Cai, S.: A synthesis of Global Streamflow Characteristics, Hydrometeorology, and Catchment Attributes (GSHA) for large sample river-centric studies, Earth System Science Data, 16, 1559–1587, https://doi.org/10.5194/essd-16-1559-2024, 2024.

---

## Author Comment (AC3)

**Reply to Referee's comments**

Referee #2

Referee's comments are in black text

Authors's response are in blue text

1) General Comments: In this work, "Unveiling the Impact of Potential Evapotranspiration Method Selection on Trends in Hydrological Cycle Components Across Europe," the authors assess 12 potential evapotranspiration (PET) formulations across the European continent using regional hydrological modelling to quantify their impact on PET trends and their implications for the main hydrological cycle components: actual evapotranspiration (AET), total water storage (TWS), and runoff (Q). They conclude that the PET model selection conditions the simulated trends and influences the analyzed hydrological component. The paper reads well; I enjoyed it while reading it. Moreover, I think the study is relevant for the catchment hydrological community; the impact of PET formulations has usually been overcome in calibration frameworks, and assessing its actual impact at a continental scale is a valuable result.

We deeply thank to Referee #2 for constructive feedback on this manuscript. The valuable provided comments will contribute significantly to enhancing the manuscript's quality. We are grateful for your time and effort invested in improving our study.

However, I have some concerns, especially regarding the methodological approach. On the one hand, I missed information about the model framework, for instance, what baseline calibration you used and how the main analyzed hydrological cycle components are linked to the model. This is key to understanding the impact of your analysis. On the other hand, the authors talked about trends, but no proper statistical trend analysis has been performed.

Details regarding the model setup and its performance are thoroughly addressed in Referee #1 comments under Section 3) Methods and data, specifically in subsections (e) and (f). Additionally, we

conducted an analysis considering significant trends, as discussed in Referee #1 comment on subsection (h) of Section 3) Methods and data.

Specific comments

2) Introduction

    a) In paragraph one (lines 24-33), it would be nice to briefly mention that other concepts like reference evapotranspiration are widely used when computing AET.

    We agree with your comment. We have addressed the discussion on reference evapotranspiration in Referee #1's comment under subsection (a) of the Introduction section. The text that we will include in the revised manuscript is as follows: *Potential evapotranspiration (PET) is the potential to evaporate water from the land surface to the atmosphere without Any limitation to water availability. Although the concept has been in use for centuries, Thornthwaite (1948) was the first to formally introduce the term "potential evapotranspiration" in the scientific literature. A related but distinct concept is "reference crop evapotranspiration", which is sometimes used interchangeably with PET. However, these terms differ in their conceptual basis and applications. Reference crop evapotranspiration specifically estimates the water requirements of a standardized reference crop under ideal conditions, whereas potential evapotranspiration provides a broader representation of water and energy exchange processes over diverse landscapes and large regions (Xiang et al., 2020).*

    b) In paragraph two (lines 24-33), I would also include some sentences explaining why there have been more than 50 models for computing PET. Are they physically based formulations? Are they empirical and therefore linked to where they were initially formulated?

    We would like to clarify that, based on recent literature, more than 100 empirical PET formulations have been developed. We will correct this information in the revised manuscript. The majority of these are temperature-based methods (40+), followed by radiation-based methods (30+), while combination-based methods are the least common (10+). Many of these empirical

methods were initially developed and tested for particular regional scales or climatic conditions. For instance, the Thornthwaite method is most suitable for humid climates, while the Hargreaves-Samani method is particularly effective in arid and semi-arid regions. Similarly, the Hamon method is suitable for all climates. We will correct and reframe lines 24-33 in the revised manuscript.

c) In the third paragraph (lines 41-47). Since your study assesses the impact of PET selection in other water cycle components, I would include more context and references about the connection between different components.

We agree with your comment. We will further extend this paragraph in revised manuscript.

3) Methods and data

a) Could you elaborate a bit about the quality of your data? Later in the discussion, you mentioned that their uncertainties were important to your results.

We agree with the Referee's comment, which has also been highlighted by the other two Referees. PET estimation and hydrological modeling are highly dependent on input data quality. The EM-Earth dataset provides high-quality precipitation and temperature data and has been shown to perform well over Europe (Tang et al., 2022). It has undergone climatology-based bias correction and accounts for precipitation undercatch. However, since EM-Earth does not include all necessary variables for PET estimation, we utilize ERA5-Land as a complementary dataset. ERA5-Land has been demonstrated to perform better than other reanalysis datasets, including ERA5 and ERA-Interim (Muñoz-Sabater et al., 2021). Several recent global studies follow a similar strategy, combining precipitation and temperature from EM-Earth with radiation, wind speed, and other meteorological variables from ERA5-Land (Tang et al., 2023; Yin et al., 2024).

b) How were the criteria used for choosing the 533 catchments? You mentioned that you try to cover all European climates, but is that the only reason? Why is there no catchment in Italy or Greece? Why are there these big differences between catchment sizes?

We acknowledge your comments. To select the catchments, different filters were applied. First, the catchment area should be greater than 500 km$^2$ to ensure that only sufficiently large basins are considered, given the spatial resolution of the meteorological dataset. Second, the observed discharge data for each catchment should be available for more than 10 years. Finally, the catchment should maintain a closed water balance based on Budyko space. It is assessed using the Evaporative Index ((P-Q)/P); we consider the catchment which have Evaporative Index of less than one. Catchment fulfill all these three conditions, then we consider that catchment. We will add this to the revised manuscript.

c) Why do the authors select these 12 specific PET models? Please add in Appendix A1 reference to each one of the chosen formulations.

We selected widely used PET methods from different categories to ensure broad applicability across hydrology, agriculture, and climate science. Additionally, we included a PET formulation that accounts for stomatal responses, which has gained increasing attention due to its ability to capture the impact of stomatal regulation on evapotranspiration. We will add the references in the Appendix A1 in the revised manuscript.

d) The authors mention that "the basins were not calibrated for each PET method to access their response in hydrological cycle components." I understand that this is a hypothesis of your study, but in any case, I assume they must be a baseline calibration. How does the model perform in this baseline calibration? Which PET is considered in this baseline calibration? Which parameter set was used in this reference calibration? Which the target variable that the model was calibrated for? I think that, in general, a deeper description of the model might help the reader to understand the implications of selecting one or other PET. Especially how the parameterization of PET-AET-soil water balance interaction is solved.

Thank you for raising this question. We have discussed the modeling details, including the model setup and its performance, in Referee #1 comments, Section 3) Methods and Data, specifically in subsections (e) and (f). The discussion is as follows:

For each basin, we performed 12 model runs, with each run corresponding to one PET method. Therefore, for the 553 catchments, the total number of model runs is $6\,636$ ($553 \times 12$). The mHM model was run at a daily time step with a spatial resolution of $0.125° \times 0.125°$ grid resolution. We did not perform any model calibration in our study. We used the default model's parameterization because we wanted to mimic how large-scale/global hydrologic models performed, as if they would be employed across continents or global scale. The basin-wise setup used here, enabled us to estimate corresponding river discharge, and quantify all components of the water balance equation. The default parameterization of mHM has been shown to perform well in previous studies (Kumar et al., 2013; Rakovec et al., 2016). Furthermore, it has been demonstrated as one of the best-performing configurations compared to other large-scale hydrological models (Samaniego et al., 2019). For instance, Samaniego et al. (2019) compared the performance of mHM with other hydrological models across 357 catchments. Their results showed that the median Kling–Gupta efficiency (KGE) for mHM was approximately 0.6 across these catchments. To address reviewer's concern regarding the model's performance, we conducted an evaluation of its performance against discharge across the basins, as presented in Figure 1. Overall, the model performed well, with median KGE values ranging from 0.6 to 0.75 for most PET methods. However, the Blaney-Criddle method showed a median KGE slightly higher than 0.3, which was lower compared to other methods.

The default parameterization came from the model developers, and it was originally established over a diverse set of German basins, in the pioneering work of Samaniego et al. (2010). Since then, mHM has become a well-established model that has been extensively evaluated across various basins and hydrological variables (Rakovec et al., 2016; Samaniego et al., 2019; Boeing

et al., 2024). For example, Rakovec et al. (2016) analyzed the model's performance across 400 European catchments. Their evaluation compared mHM's discharge simulations using 36 different parameter sets and found that the model's performance was consistent regardless of parameterization. Introducing new model setups or performing additional calibration and comparative analyses is beyond the scope of this study. We agree that the calibration aspect is important and offers interesting insights. However, we prefer to explore it in our future research. Additionally, we will discuss the potential effects of model calibration in the discussion section to provide further context on this limitation in our manuscript.

[Figure]

**Figure 1.** Evaluation of the hydrological model (mHM) performance using simulated monthly streamflow across 553 catchments, forced with EM-Earth meteorological data. The figure presents the cumulative frequency distributions of the Kling–Gupta efficiency (KGE) and its three components: correlation (r), variability ratio (alpha), and bias ratio (beta), providing insights into the model's performance across different PET methods.

e) Regarding the trend analysis, to my knowledge, Sen's slope method is a non-parametric test to compute the magnitude and direction of linear change on a time series, as the authors state in lines 146-150. To be able to talk about a trend, it needs to be significant. To assess that, other statistical tests should be performed. The Mann-Kendall test and its variations are the most commonly used for these purposes. For instance, one paper you cited, Anabalón and Sharma, 2017, used a test of the Kendall family (Seasonal Kendall trend test) to determine the trends and the Sen's slope test to determine their magnitudes. If only Sen's slope is computed, the authors might talk about the evolution of the slope but not about trends.

These are reasonable concerns, also raised by other two Reviewers, as the inclusion of weak trends can indeed introduce noise to the analysis. considering all trends allows for an assessment of the spatial consistency of directional changes, which is a key observation in our study. If these weaker trends were purely random noise, their distribution would be approximately symmetric, with equal numbers exhibiting positive and negative changes. In contrast, the fact that the weaker trends predominantly align in the same direction suggests that, while they may not meet conventional significance thresholds (e.g., $p = 0.05$), they are not statistically irrelevant. The overwhelming consistency in their direction suggests a potential underlying signal rather mere stochastic variability. Overreliance on statistical significance can lead to rejecting meaningful patterns simply because they do not meet an arbitrary threshold, due to low variability rather than the absence of a real effect.

We also respectfully disagree that in order to talk about a trend it needs to be significant. This is a common misconception in time series analysis, which has risen a lot of criticism in many scientific disciplines. We think the best example is the milestone Editorial in the American Statistican by Ronald L. Wasserstein and Lazar (2019). Among many suggestions about the correct use of statistical significance they state that "no single index should substitute for

scientific reasoning" and caution against the rigid use of p-values as an absolute determinant of scientific conclusions.

Still, we recognize the importance of showing how trend filtering, i.e., statistical significance testing, affects our results. Therefore, we will maintain our original analysis, which includes all trends, while complementing it with the analysis of significant trends in the supplementary material (Figure 2 & Figure 3 shows DCI considering significant trend with p value 0.05). Additionally, we will articulate more clearly the reasons for our decision and describe the impact of trend significance in the revised manuscript.

[Figure]

**Figure 2.** Spatial distribution of annual scale data concurrence index (DCI) for PET, AET, Q, and TWS by considering only significant trend at 95% significance level. PET represents potential evapotranspiration, AET represents actual evapotranspiration, Q represents runoff at the outlet of the catchment and TWS represents total water storage.

[Figure]

**Figure 3.** Spatial distribution of seasonal scale data concurrence index (DCI) for PET, AET, Q, and TWS by considering only significant trend at 95 % significance level. PET represents potential evapotranspiration, AET represents actual evapotranspiration, Q represents runoff at the outlet of the catchment and TWS represents total water storage.

4) Results

a) As I stated before, the authors should not talk about trends but rather the evolution of the slope. So, sections 3.1, 3.2, and 3.3 might be rewritten in this sense or carried out the trend analysis and talk about trends.

In our previous response, we acknowledged that the obtained results are meaningful even in the absence of statistical significance. Thus, we will modify these sections to preserve the original findings and integrate additional results corresponding to statistically significant trends.

b) Besides talking about trends, I think it would be nice for the reader to have some information about the actual values, at least for PET. I would include 12 maps, one per PET model selected with actual PET values. It would help the reader to spot differences.

We agree with your comments. We will add the figure in the supplementary material.

5) Discussion

a) There are topics I would highlight in the discussion that it did not. For instance, are the different sizes of the catchments conditioning your results? Another issue I would appreciate including is a discussion about why specific models produce different results when they are within the same category. That is, why are there differences between temperature models? Is it linked to their specific formulation?

Thank you for your constructive comments. Differences in PET models within the same category arise due to variations in their mathematical formulations and underlying assumptions. For instance, in temperature-based methods, Thornthwaite incorporates only the average temperature and was originally developed for humid climates, whereas the Hargreaves-Samani method considers the diurnal temperature range along with the average temperature and was initially formulated for arid and semi-arid climates. Similarly, among combination-type methods, the Penman-Monteith model explicitly accounts for wind speed and vapor pressure deficit, while the Priestley-Taylor method simplifies this by using an empirical coefficient.

Furthermore, we will include a discussion on catchment size conditioning in our revised manuscript.

6) Summary and conclusions

    a) Conclusion 7 is obvious. I would remove it.

    We agree with your comment; however, Referee #3 recommended incorporating a discussion on precipitation in the main text of the manuscript. Thus, we consider it appropriate to retain this point in the conclusion with necessary modifications, even though it appears obvious.

7) Technical corrections

    a) Please homogenize "combination type" vs. "combinational type."

    We will update it in the revised manuscript.

    b) Figure 1b: it is unclear what each dot represents. I assume they are catchments, but it seems to me less than 533.

    We agree with your comments. It seems there may have been some misunderstanding regarding Figure 1b. To clarify, Figure 1b represents only the test catchments, which are specifically highlighted with thick black borders in Figure 1a. To improve its clarity, the figure (Figure 4) has been further revised based on the comments provided by Referee#3.

[Figure]

**Figure 4.** Catchment classification to energy-limited, mixed, and water-limited categories. a) Catchment locations; black borders indicate a representative catchment of each category. b) Classification example within the Budyko space for the representative catchments. c) Annual time series of simulated hydrological components from the mesoscale hydrological model for each representative catchment and PET estimation method (TH: Thornthwaite, BR: Bair-Robertson, BC: Blaney-Criddle, OD: Oudin, MB: McGuinness-Borden, HM: Hamon, HS: Hargreaves-Samani, JH: Jensen-Haise, MD: Milly-Dunne, PT: Priestley-Taylor, PM: Penman-Monteith, CO$_2$: Modified Penman-Monteith accounts CO$_2$.). All units are in mm year$^{-1}$.

c) Line 141: "For the each" should be "for each."

We will correct it in the revised manuscript.

d) In Figures 2 and 3, I recommend using the same range on the y-axis for each variable to see the differences. In addition, I would add the number of catchments in each category, that is, 189, 330, and 34 for energy-limited, mixed, and water-limited, respectively.

Thank you for highlighting this aspect. A uniform y-axis range across catchment categories would indeed facilitate direct inter-catchment category comparisons. However, our primary goal is to analyze and compare the changes in PET methods within each catchment category individually. Adjusting the y-axis range for each category ensures that trends for PET methods remain distinguishable. Using the same y-axis range would cause trends in energy-limited catchments, which are generally higher, to dominate the visual scale. This would obscure the subtle variations among PET methods in water-limited catchments. Consequently, the comparison within water-limited categories would become less clear and less informative.

e) Figure 6, not all categories sum 553. Please revise. In addition, I think the bar representation with different numbers of catchments and models in each is not intuitive. Maybe the same structure but in a table format for the upper part, in which models are rows, would be more intuitive.

We are agree with your comments. We have addressed the issues related to catchment counts and improvements in figure interaction in Referee #1's comment under Section 4) Results, subsection (g). The discussion is as follows:

In Figure 6, we presented only ten possible hydrological cycle combinations, whereas there are actually 14 possible combinations, as shown in Figure 5. The excluded combinations were not considered because they were represented by only a very small number of catchments. The remaining two catchments from the Thornthwaite method belonged to the (PRE-, AET+, Q+, TWS-) combination. Including these catchments brings the total count to 553 (240 + 156 + 59 + 70 + 24 + 2 + 2). This ensures that all catchments are accounted for in the analysis.

[Figure]

**Figure 5.** Fourteen distinct combinations of hydrological cycle components and their respective influence of PET methods on an annual scale. PRE+, AET+, Q+, and TWS+ represent an increasing trend for PRE, AET, Q, and TWS respectively. Similarly, PRE-, AET-, Q- and TWS- represent a decreasing trend. Where PRE is precipitation, AET is actual evapotranspiration, Q is runoff and TWS is total water storage. Abbreviations used for different PET methods are TH: Thornthwaite, BR: Bair-Robertson, BC: Blaney-Criddle, OD: Oudin, MB: McGuinness-Borden, HM: Hamon, HS: Hargreaves-Samani, JH: Jensen-Haise, MD: Milly-Dunne, PT: Priestley-Taylor, PM: Penman-Monteith, CO$_2$: Modified Penman-Monteith accounts CO$_2$.

f) Please check the name of the PET model throughout the manuscript; there are some inconsistencies.

We will remove these inconsistencies in the revised manuscript.

**References**

Boeing, F., Wagener, T., Marx, A., Rakovec, O., Kumar, R., Samaniego, L., and Attinger, S.: Increasing influence of evapotranspiration on prolonged water storage recovery in Germany, Environmental Research Letters, 19, 024 047, https://doi.org/10.1088/1748-9326/ad24ce, 2024.

Kumar, R., Livneh, B., and Samaniego, L.: Toward computationally efficient large-scale hydrologic predictions with a multiscale regionalization scheme, Water Resources Research, 49, 5700–5714, 2013.

Muñoz-Sabater, J., Dutra, E., Agustí-Panareda, A., Albergel, C., Arduini, G., Balsamo, G., Boussetta, S., Choulga, M., Harrigan, S., Hersbach, H., Martens, B., Miralles, D. G., Piles, M., Rodríguez-Fernández, N. J., Zsoter, E., Buontempo, C., and Thépaut, J.-N.: ERA5-Land: a state-of-the-art global reanalysis dataset for land applications, Earth System Science Data, 13, 4349–4383, https://doi.org/10.5194/essd-13-4349-2021, 2021.

Rakovec, O., Kumar, R., Mai, J., Cuntz, M., Thober, S., Zink, M., Attinger, S., Schäfer, D., Schrön, M., and Samaniego, L.: Multiscale and Multivariate Evaluation of Water Fluxes and States over European River Basins, Journal of Hydrometeorology, 17, 287–307, https://doi.org/10.1175/JHM-D-15-0054.1, 2016.

Ronald L. Wasserstein, A. L. S. and Lazar, N. A.: Moving to a World Beyond "p < 0.05", The American Statistician, 73, 1–19, https://doi.org/10.1080/00031305.2019.1583913, 2019.

Samaniego, L., Kumar, R., and Attinger, S.: Multiscale parameter regionalization of a grid-based hydrologic model at the mesoscale, Water Resources Research, 46, https://doi.org/10.1029/2008WR007327, 2010.

Samaniego, L., Thober, S., Wanders, N., Pan, M., Rakovec, O., Sheffield, J., Wood, E. F., Prudhomme, C., Rees, G., Houghton-Carr, H., Fry, M., Smith, K., Watts, G., Hisdal, H., Estrela, T., Buontempo, C., Marx, A., and Kumar, R.: Hydrological Forecasts and Projections for Improved Decision-Making in the Water Sector in Europe, Bulletin of the American Meteorological Society, 100, 2451–2472, https://doi.org/10.1175/BAMS-D-17-0274.1, 2019.

Tang, G., Clark, M. P., and Papalexiou, S. M.: EM-Earth: The Ensemble Meteorological Dataset for Planet Earth, Bulletin of the American Meteorological Society, 103, E996–E1018, https://doi.org/10.1175/BAMS-D-21-0106.1, 2022.

Tang, G., Clark, M. P., Knoben, W. J. M., Liu, H., Gharari, S., Arnal, L., Beck, H. E., Wood, A. W., Newman, A. J., and Papalexiou, S. M.: The Impact of Meteorological Forcing Uncertainty on Hydrological Modeling: A Global Analysis of Cryosphere Basins, Water Resources Research, 59, e2022WR033 767, https://doi.org/10.1029/2022WR033767, 2023.

Thornthwaite, C. W.: An Approach toward a Rational Classification of Climate, Geographical Review, 38, 55, https://doi.org/10.2307/210739, 1948.

Xiang, K., Li, Y., Horton, R., and Feng, H.: Similarity and difference of potential evapotranspiration and reference crop evapotranspiration – a review, Agricultural Water Management, 232, 106 043, https://doi.org/10.1016/j.agwat.2020.106043, 2020.

Yin, Z., Lin, P., Riggs, R., Allen, G. H., Lei, X., Zheng, Z., and Cai, S.: A synthesis of Global Streamflow Characteristics, Hydrometeorology, and Catchment Attributes (GSHA) for large sample river-centric studies, Earth System Science Data, 16, 1559–1587, https://doi.org/10.5194/essd-16-1559-2024, 2024.

---

## Author Comment (AC4)

Referee #1: Clerc-Schwarzenbach, Franziska

Referee's comments are in black text

Authors's response are in blue text

1) General Comments: Thank you for giving me the opportunity to review the manuscript "Unveiling the impact of potential evapotranspiration method selection on trends in hydrological cycle components across Europe" by Vishal Thakur and co-authors. I enjoyed reading this manuscript and consider the findings to be valuable and highly relevant: Even though potential evapotranspiration is a crucial component of the hydrological cycle, factors like data availability or convenience often play a role in the selection of a potential evapotranspiration formulation. There is no uniform way of dealing with potential evapotranspiration in hydrology, and often, different concepts (such as reference evapotranspiration and potential evapotranspiration) are used interchangeably. Therefore, studies like the one by Vishal Thakur et al., analysing the effects of different formulations of potential evapotranspiration are of great value to shed light on this often-neglected topic. The manuscript presents an analysis of the influence of different potential evapotranspiration formulations on different simulated components of the hydrological cycle. To do so, the authors make use of a large-scale modelling approach and analyse the modelling results with effective methods. They come to the important conclusion that the choice of a potential evapotranspiration method has an influence on the results when studying the hydrological cycle and its components.

We would like to deeply thank you for your constructive feedback, which plays a key role in improving the quality of our manuscript. The depth and thoughtfulness of your review have greatly aided us in presenting our findings more effectively and aligning them with broader scientific discussions. We deeply appreciate your time, effort, and dedication in offering such a thorough and thoughtful review, which has significantly enhances the quality of our manuscript.

My main concern is that the majority of the chosen formulations (or methods) are temperature-based. Since temperatures were rising in the studied period between 1980 and 2019, the potential evapotranspiration methods based on temperature do show a positive trend. So far, it is not clear if temperature-based methods are still reliable under the conditions of a warming climate (see for example studies on the so-called "pan evaporation paradox": Li et al. (2013, 10.1002/wrcr.20202); Wang et al. (2017,10.1002/wat2.1207)). With eight out of twelve methods being temperature-based, a possible overestimation may strongly influence the results of this study.

We agree that our study includes more temperature-based PET methods compared to radiation-based and combination methods. This imbalance in the number of methods could potentially influence the Data Concurrence Index (DCI) analysis. To address this issue, we randomly selected four temperature-based methods, resulting in 70 combinations when paired with the other PET methods (radiation and combination methods). Upon analyzing these 70 combinations, PET consistently exhibited strong positive DCI (DCI $\geq$ 0.5) across all 553 catchments. For AET, the median number of catchments showing strong positive DCI was $535 \pm 10$, while disagreement (DCI between -0.5 and 0.5) was observed in $17 \pm 10$ catchments. For Q, $269 \pm 3$ catchments showed strong negative agreement (DCI $\leq$ -0.5), $28 \pm 3$ catchments exhibited disagreement, and $256 \pm 4$ catchments displayed strong positive agreement. Similarly, for TWS, $368 \pm 6$ catchments showed strong negative agreement, $53 \pm 7$ exhibited disagreement, and $132 \pm 4$ demonstrated strong positive agreement. The analysis of these 70 different combinations (four temperature-based methods and four complex methods) suggests that only a small number of catchments are affected in the DCI analysis. Out of 70 two combinations are shown in Figure 1 and Figure 2. However, the trend comparisons at annual and seasonal scales (Sections 3.1 and 3.2) remain unaffected by the varying number of PET method categories, as trends were evaluated for individual methods rather than aggregated categories. We will incorporate this information into the supplementary material in the revised manuscript.

[Figure]

**Figure 1.** Spatial distribution of annual scale data concurrence index (DCI) for PET, AET, Q, and TWS. Considering eight PET methods namely Blaney-Criddle, Bair-Robertson, Hamon, Hargreaves-Samani, Milly-Dunne, Priestley-Taylor, Penman-Monteith and Modified Penman-Monteith accounts CO2.

[Figure]

**Figure 2.** Spatial distribution of annual scale data concurrence index (DCI) for PET, AET, Q, and TWS. Considering eight PET methods namely Jensen-Haise, McGuinness–Bordne, Oudin, Thornthwaite, Milly-Dunne, Priestley-Taylor, Penman-Monteith and Modified Penman-Monteith accounts $CO_2$.

Specific comments and suggestions that will hopefully help to improve the different parts of the manuscript are listed below.

2) Introduction

    a) While Thornthwaite (1948) was the first to introduce the term "evapotranspiration", according to Miralles et al. (2020, 10.1029/2020WR028055), the concept itself was already used earlier. This should be specified. Furthermore, as the different concepts regarding evaporation (including "reference crop evapotranspiration") are often used interchangeably in hydrology, I would suggest to add a statement about this problem, e.g., referring to the Miralles et al. paper mentioned above as well as the already cited paper by Xiang et al. (2020). In addition, I would argue that there should be some indication if you consider evapotranspiration to include interception or not in your study. See also the commentary by Savenije (2004, 10.1002/hyp.5563).

    Thank you very much for your fruitful comments. We have elaborated development of potential evapotranspiration and added a small introduction about reference evapotranspiration. Modified sentences are as follows.

    *Potential evapotranspiration (PET) is the potential to evaporate water from the land surface to the atmosphere without Any limitation to water availability. Although the concept has been in use for centuries, Thornthwaite (1948) was the first to formally introduce the term "potential evapotranspiration" in the scientific literature. A related but distinct concept is "reference crop evapotranspiration", which is sometimes used interchangeably with PET. However, these terms differ in their conceptual basis and applications. Reference crop evapotranspiration specifically estimates the water requirements of a standardized reference crop under ideal conditions, whereas potential evapotranspiration provides a broader representation of water and energy exchange processes over diverse landscapes and large regions (Xiang et al., 2020).*

    mHM model incorporates the interception process, where a portion of actual evapotranspiration (AET) results from interception evaporation. This process is estimated as a fraction of potential

evapotranspiration (PET) using a power function derived from Deardorff (1978) and Liang et al. (1994). Furthermore, if evaporation exceeds the intercepted water, the interception storage is completely depleted.

b) In the first sentence of the paragraph starting at line 41, PET is stated to directly influence AET. In the second sentence, an alternative to this direct influence is presented. Therefore, the first sentence should be adjusted so that it becomes clear that this is just one possibility. Furthermore, depending on how interception is dealt with, this may be a necessary component to add in the list of fluxes that make up AET (in the second sentence of this paragraph).

We agree with your comment. We will update line 41 and provide a detailed description of the components contributing to AET.

3) Methods and data

a) In the beginning of section 2.2, you give the study by Hersbach et al. (2020) as a reference for ERA5-Land. I would argue that the suitable reference there is the paper by Muñoz-Sabater et al. (2021), that you cite later in the manuscript. Furthermore, in line 97, there is a reference missing for EM-Earth and SC-Earth, or it is not clear to me if the Tang et al. (2022) reference belongs to this. If the EM-Earth data are based on ERA5 data (and not on ERA5-Land data), I would suggest to include the reference to the Hersbach et al. (2020) paper there.

We will correct reference typos; the reference (Muñoz-Sabater et al.; 2021) to ERA5-Land will be corrected in the revised manuscript. The EM-Earth dataset is distinct and primarily generated by integrating multiple data sources, with SC-Earth and ERA5 being the two major contributors. To improve clarity, a reference will be added to line 97. *The EM-Earth dataset (Tang et al., 2022) is derived from observed station data SC-Earth (Tang et al., 2021) and ERA5 data (Hersbach et al., 2020).*

b) I assume the AET, Q, and TWS data in Figure 1c to be simulated values. In my opinion, this needs to be mentioned in the figure caption as it only becomes clear after reading further.

The caption for Figure 1(c) has been revised for improved clarity as follows: *Annual time series of simulated hydrological components from the mesoscale hydrological model for each representative catchment and PET estimation method. (TH: Thornthwaite, BR: Bair-Robertson, BC: Blaney-Criddle, OD: Oudin, MB: McGuinness-Borden, HM: Hamon, HS: Hargreaves-Samani, JH: Jensen-Haise, MD: Milly-Dunne, PT: Priestley-Taylor, PM: Penman-Monteith, $CO_2$: Modified Penman-Monteith accounts $CO_2$.). All units are in mm year$^{-1}$.*

c) In line 122, you state that some of the temperature-based methods also use some extraterrestrial radiation term calculated based on latitude. Potentially, this could already be mentioned earlier, when the definition of the temperature-based category is given (i.e., it could become clearer that also radiation terms can be required and a method is still considered to be temperature-based). For better clarity entire paragraph (Line 117 - 129 ) was rewritten. *We incorporate 12 PET methods at a daily scale from all three categories of estimation: temperature-based, radiation-based, and combination-based methods (Table 2). Temperature-based methods require temperature data, which can include average temperature, minimum temperature, or maximum temperature. Additionally, PET methods that incorporate extraterrestrial radiation are also considered under this category. Combination-based methods require a larger number of variables compared to temperature- and radiation-based methods to estimate various physical terms, such as wind speed and surface pressure (Table 2). Most temperature-based methods use only daily average temperature (Thornthwaite, Oudin, Hamon, Jensen-Haise, McGuinness-Bordne, and Blaney-Criddle), while Baier-Robertson employs both minimum and maximum daily temperatures. Some of these methods also include an extraterrestrial radiation term in their formulation. However, as this radiation term is calculated based on latitudinal information, only temperature data is required for PET calculation. We utilize only one radiation-based method, Milly-Dunne PET, which requires only net radiation data to estimate PET. The combination-based methods, such as Penman-Monteith and Priestley-Taylor, have a stronger*

*physical basis. In our analysis, all these physical terms are estimated following Allen (1998). Additionally, within the combination category, we employ the modified Penman-Monteith (CO2) method, which accounts for temporal variations in changing carbon dioxide concentrations. Formulation details, including mathematical equations and associated constants for each PET method, are provided in Table A1.*

d) When you list the terms that are required for the Penman-Monteith and the Priestley-Taylor method, I would suggest to list all the terms (as it's only two equations), instead of giving some examples and concluding with "etc.". Alternatively, you could give one or two examples and then place a reference to the table where you list all the input data required for each method.
We have incorporated two examples and provided a reference to the relevant table in the manuscript. The modified version of the paragraph has been addressed in subsection(c) of section 3) Methods and data of response letter.

e) For me, the description of the modelling part (paragraph starting on line 139) is hard to follow. Therefore, I ask you to give more information about this part of your study: You do one model run per catchment and PET method (553 x 12 = 6636). Is there any model calibration? If yes, please elaborate on that: What objective function(s) did you use? What was the spatial resolution? If no, where do you take the model settings from? Are these the same for all catchments? How do you make sure that the parameterization that you use matches your catchments? Did you think about equifinality and how other possible parameterizations could influence your results? How well did the model perform for the different catchments? In the very end of the discussion, the reader learns that a default parameterization was used. Please add this to the methods part and make sure that the questions listed above become clear.
We agree with your comments. We address them as follows. For each basin, we performed 12 model runs, with each run corresponding to one PET method. Therefore, for the 553 catchments, the total number of model runs is 6,636 (553 x 12). The mHM model was run at a daily time step

with a spatial resolution of 0.125° x 0.125° grid resolution. We did not perform any model calibration in our study. We used the default model's parameterization because we wanted to mimic how large-scale/global hydrologic models performed, as if they would be employed across continents or global scale. The basin-wise setup used here, enabled us to estimate corresponding river discharge, and quantify all components of the water balance equation. The default parameterization of mHM has been shown to perform well in previous studies (Kumar et al., 2013; Rakovec et al., 2016). Furthermore, it has been demonstrated as one of the best-performing configurations compared to other large-scale hydrological models (Samaniego et al., 2019). For instance, Samaniego et al. (2019) compared the performance of mHM with other hydrological models across 357 catchments. Their results showed that the median Kling–Gupta efficiency (KGE) for mHM was approximately 0.6 across these catchments. To address reviewer's concern regarding the model's performance, we conducted an evaluation of its performance against discharge across the basins, as presented in Figure 3. Overall, the model performed well, with median KGE values ranging from 0.6 to 0.75 for most PET methods. However, the Blaney-Criddle method showed a median KGE slightly higher than 0.3, which was lower compared to other methods. We will add its details in the revised manuscript.

[Figure]

**Figure 3.** Evaluation of the hydrological model (mHM) performance using simulated monthly streamflow across 553 catchments, forced with EM-Earth meteorological data. The figure presents the cumulative frequency distributions of the Kling–Gupta efficiency (KGE) and its three components: correlation (r), variability ratio (alpha), and bias ratio (beta), providing insights into the model's performance across different PET methods.

f) Related to the comment above, where does this default parameterization come from? Was there a method for potential evapotranspiration involved when this default parameterization was obtained? If so, may this have varying effects on the results based on this (or similar) methods (that can potentially profit from compensating effects) and the results based on different methods (that cannot profit from any compensating effects)? Potentially and if possible (may be limited due to data availability constraints) it would be interesting to compare the current results to the results of a calibrated model (i.e., calibrated for each of the different methods) – you also include a similar remark in the discussion. With additional calibrations that make use of the different

methods, it could become clearer if the (assumed) use of one method for the default parameterization affects the results. If additional calibrated model runs are not possible, I would suggest you to include some text on the potential effects of the default parameterization in the discussion.

We acknowledge your comments. The default parameterization came from the model developers, and it was originally established over a diverse set of German basins, in the pioneering work of Samaniego et al. (2010). Since then, mHM has become a well-established model that has been extensively evaluated across various basins and hydrological variables (Rakovec et al., 2016; Samaniego et al., 2019; Boeing et al., 2024). For example, Rakovec et al. (2016) analyzed the model's performance across 400 European catchments. Their evaluation compared mHM's discharge simulations using 36 different parameter sets and found that the model's performance was consistent regardless of parameterization. Introducing new model setups or performing additional calibration and comparative analyses is beyond the scope of this study. We agree that the calibration aspect is important and offers interesting insights. However, we prefer to explore it in our future research. Additionally, we will discuss the potential effects of model calibration in the discussion section to provide further context on this limitation in our manuscript.

g) I suggest you to clarify in lines 158 and 159 that you compare winter data to winter data, spring data to spring data, etc. so that it is immediately clear that you do not compare all four seasons with each other.

Following your comments, Lines 158–1599 were further elaborated as follows: *The slopes of each PET method were analyzed exclusively within seasons (for eaxmple winter season compared with winter season), without cross-seasonal comparisons.*

h) In section 2.3.4, you describe the modification of the DCI by including also non significant trends to be able to include all trend estimates. Are you sure that you are not including too much noise

by also counting very weak positive and negative trends? I suggest to at least show that your findings do not change if only reasonably significant trends are considered.

These are reasonable concerns, also raised by other two Reviewers, as the inclusion of weak trends can indeed introduce noise to the analysis. considering all trends allows for an assessment of the spatial consistency of directional changes, which is a key observation in our study. If these weaker trends were purely random noise, their distribution would be approximately symmetric, with equal numbers exhibiting positive and negative changes. In contrast, the fact that the weaker trends predominantly align in the same direction suggests that, while they may not meet conventional significance thresholds (e.g., $p = 0.05$), they are not statistically irrelevant. The overwhelming consistency in their direction suggests a potential underlying signal rather mere stochastic variability. Overreliance on statistical significance can lead to rejecting meaningful patterns simply because they do not meet an arbitrary threshold, due to low variability rather than the absence of a real effect.

We also respectfully disagree that in order to talk about a trend it needs to be significant. This is a common misconception in time series analysis, which has risen a lot of criticism in many scientific disciplines. We think the best example is the milestone Editorial in the American Statistican by Ronald L. Wasserstein and Lazar (2019). Among many suggestions about the correct use of statistical significance they state that"no single index should substitute for scientific reasoning" and caution against the rigid use of p-values as an absolute determinant of scientific conclusions.

Still, we recognize the importance of showing how trend filtering, i.e., statistical significance testing, affects our results. Therefore, we will maintain our original analysis, which includes all trends, while complementing it with the analysis of significant trends in the supplementary material (Figure 4 & Figure 5 shows DCI considering significant trend with p value 0.05).

Additionally, we will articulate more clearly the reasons for our decision and describe the impact of trend significance in the revised manuscript.

[Figure]

**Figure 4.** Spatial distribution of annual scale data concurrence index (DCI) for PET, AET, Q, and TWS by considering only significant trend at 95% significance level. PET represents potential evapotranspiration, AET represents actual evapotranspiration, Q represents runoff at the outlet of the catchment and TWS represents total water storage.

[Figure]

**Figure 5.** Spatial distribution of seasonal scale data concurrence index (DCI) for PET, AET, Q, and TWS by considering only significant trend at 95 % significance level. PET represents potential evapotranspiration, AET represents actual evapotranspiration, Q represents runoff at the outlet of the catchment and TWS represents total water storage.

4)  Results

a) In the first paragraph of section 3.1, you describe the different trends in PET for the three categories of catchments. As PET describes a potential, i.e., the maximum ET that could be achieved if there was no water-limitation, the trend in PET should not depend on if a catchment is waterlimited or energy limited. Thus, I suggest to formulate this paragraph differently, such that it is clear that the PET trend is influenced by other factors, but not by the availability of water. We will revise this paragraph by linking the factors influencing PET and relating them to the context of catchment types. We will update this paragraph and incorporate into the revised manuscript.

b) In the last paragraph of section 3.2, you suddenly write about December and not about the winter season (line 234). If you did monthly analyses, this needs to be stated clearly in the methods section. Otherwise, December may be the wrong term here. Yes, you are right. We made a typing mistake in this line. The paragraph was intended for seasonal comparison. Instead of "December," it should refer to the "winter season." The corrected sentence for lines 234–235 is as follows: *In contrast, for energy-limited catchments, the winter and summer seasons are the main contributors, with their impacts varying depending on the selected PET method (Figure S13)*

c) As already mentioned in the general comments: With the rising temperatures, all the temperature-based methods tend to show positive trends. Since most PET methods included are temperature-based, they have the largest influence on the DCI. It would be good to test if this unequal distribution of the different types influences your results. We agree that our study has an unequal distribution of PET methods across different categories. To address this, we conducted an analysis using an equal distribution of PET methods, selecting four temperature-based methods and four complex methods. A detailed discussion of the observed result is provided in 1) General Comments.

d) To make the part about the combinations of hydrological cycle component changes (section 3.4) easier to understand also before studying Figure 6, I suggest you to add an example of a "possible combination of hydrological cycle component changes" after you introduce this term on lines 269 and 270. Similarly, it may be good to inform the reader what you mean with "the first five hydrological cycle combinations" (lines 270 and 271).

We agree with your comments. For better clarity example of combination hydrological cycle will be added. *Here, we demonstrate the overall impact of PET methods on possible combinations of hydrological cycle component changes. For instance, one hydrological cycle combination involves an increase in all components (PRE+, AET+, Q+, TWS+). Conversely, another combination represents decreasing all components (PRE-, AET-, Q-, TWS-).*

For a better understanding of lines 270 and 271 additional text will be added to revised text. *These five combinations are as follows: (1) PRE+, AET+, Q+, TWS+; (2) PRE-, AET+, Q-, TWS-; (3) PRE+, AET+, Q-, TWS-; (4) PRE+, AET+, Q+, TWS-; and (5) PRE-, AET-, Q-, TWS-.*

e) In lines 272 and 273 you state that the temperature-based methods account for more catchments with positive trends across all hydrological cycle components than combinational methods (and you put this in comparison to the Blaney-Criddle method leading to positive trends in all components for most catchments). This statement is not clear: Do you want to say that the temperature-based methods lead to the "all positive" combination in more catchments than the combinational methods? If so, this is not true for the Baier-Robertson method, if I interpret Figure 6 correctly. Please reformulate this statement and reconsider if this should be formulated as a comparison to the first part of the sentence.

We will revise the sentence to improve its clarity.

f) The sentence regarding the combinations with AET+ and TWS- (lines 280 and 281) should come before the statement about the last five combinations to be consistent with the order in Figure 6.

We will move the sentence to the appropriate place in revised manuscript.

g) I think that Figure 6 should be improved so that it can be grasped more quickly and easily:

We acknowledge your comments regarding Figure 6. We used a specific R package to generate this figure, which has certain limitations. However, we will make every effort to implement the suggested modifications to improve its clarity and interactivity. Additionally, we will include explanatory text to help readers better interpret the figure. In the worst-case scenario, if the necessary improvements cannot be made, we may consider removing the figure from the manuscript.

- I suggest you to put the lower part (in the following called "combinations") to the top and the upper part (in the following called "catchment count") to the bottom. The reader first wants to know what combination we are looking at, and then wants to look at the results for this combination. I see that you used the "combinations" part as an "axis label", but this is not so clear when looking at the plot in the beginning as it looks more like two different parts.
We will make the changes accordingly in the revised manuscript.

- For the "combinations", consider displaying them differently. For example, a simple table-like graphic in which you could even work with additional colours would in my opinion be easier to interpret than the current way to display the combinations, see below. If you decide to do so, do not forget to change the caption.
We agree with your comment. As an alternative for better clarity, we will incorporate color in the combinations (for example, blue for a positive trend and brown for a negative trend).

- For the "catchment count", I think it would improve the readability of the figure if not all bars were the same height, i.e., if the height of the bar would decrease from left to right, proportionally to the higher number of catchments that are contained in the bars on the left than in the ones on the right. This way, it would become clearer which combinations occur how often. For that, you could consider using horizontal instead of vertical bars to gain more

space (this would also allow you to use the "combinations" part as "axis labels", but then of the vertical axis).

Thank you very much for your constructive suggestion. We will adjust the height of the bars accordingly. However, doing so may make it challenging to display the catchment numbers for combinations with a lower catchment count.

■ Please double-check the catchment count for each method. To my understanding, the number of catchments per method contained in this plot should always sum up to 553. However, for example for the Thornthwaite method, the number of catchments only sums up to 240+156+59+70+24+2=551.

We agree with your comment. In Figure 6, we presented only ten possible hydrological cycle combinations, whereas there are actually 14 possible combinations, as shown in Figure 6. The excluded combinations were not considered because they were represented by only a very small number of catchments. The remaining two catchments from the Thornthwaite method belonged to the (PRE-, AET+, Q+, TWS-) combination. Including these catchments brings the total count to 553 (240 + 156 + 59 + 70 + 24 + 2 + 2). This ensures that all catchments are accounted for in the analysis.

[Figure]

**Figure 6.** Fourteen distinct combinations of hydrological cycle components and their respective influence of PET methods on an annual scale. PRE+, AET+, Q+, and TWS+ represent an increasing trend for PRE, AET, Q, and TWS respectively. Similarly, PRE-, AET-, Q- and TWS- represent a decreasing trend. Where PRE is precipitation, AET is actual evapotranspiration, Q is runoff and TWS is total water storage. Abbreviations used for different PET methods are TH: Thornthwaite, BR: Bair-Robertson, BC: Blaney-Criddle, OD: Oudin, MB: McGuinness-Borden, HM: Hamon, HS: Hargreaves-Samani, JH: Jensen-Haise, MD: Milly-Dunne, PT: Priestley-Taylor, PM: Penman-Monteith, $CO_2$: Modified Penman-Monteith accounts $CO_2$.

■ As the Thornthwaite method and the Blaney-Criddle method are displayed in neighbouring fields for the seventh combination, please consider not using red and green for these two methods as they are hard to distinguish for colour-blind people. Alternatively, please use patterns in addition, to make it possible to distinguish the two colours.

■ We will update the colors in the revised manuscript.

5) Discussion

a) In line 302, you compare your results to the ones in the study by Hanselmann et al. (2024).Please note that while the authors of this study are affiliated in Poland, the research has been conducted for Spitsbergen.

We will update the discussion with a suitable reference.

b) In the paragraph starting on line 336, you discuss your results as well as the results of the study by Teuling et al. (2019). Based on the Teuling et al. paper as well as based on the PET methods that you considered in your study, I assume that you mean the Penman-Monteith method in this paragraph when you write about the Penman method. Please double-check and correct. The same issue occurs again in line 384.

Yes you are right. We will correct the typo penman to Penman-Monteith.

c) Later in the same paragraph, where you discuss a possible drying hydrological cycle, I think that a discussion of the study by Milly & Dunne (2016) that you refer to elsewhere in your manuscript is lacking as they studied this topic.

We agree with your comments. Our intention is to focus on a specific combination of hydrological cycle components. To enhance clarity, a minor modification will be made to lines 340–345.
*Similarly, many catchments demonstrate positive changes in AET and negative changes in Q, TWS, and PRE (drying condition of the hydrological cycle). The number of catchments demonstrating these trends varies depending on the choice of PET method. Massari et al. (2022) found that over Europe runoff deficit are more pronounced in water-limited regions due to increased AET, whereas energy-limited catchments exhibit smaller deficits. During these drying conditions, AET is further influenced by reductions in TWS (Massari et al., 2022).*

d) In line 368, you write about methods that consistently overestimate PET. It is unclear to me how you define overestimation here: Do you just consider the methods leading to the highest estimates to be overestimating PET? If so, I would not consider it to be surprising that the catchments shift to the energy-limited category (as all the higher estimates are excluded). If all the low estimates

would be excluded, the catchments would probably shift to the water-limited category. Please elaborate more on what you mean there.

In line 368, methods that consistently overestimate PET refer to those producing higher estimates compared to other PET methods. Specifically, the Jensen-Haise and Blaney-Criddle methods show higher PET values than other methods. Table S1 in the manuscript illustrates catchment shifts when different PET methods with higher estimates are excluded.

e) When you discuss the limitations of your study, you state that on the one hand, the PET methods were found to be sensitive to the input data, and on the other hand, that hydrological models are sensitive to the precipitation input. Based on this, I think it would be good to write something about the data quality of the input data that you used (as you basically state that your results are highly sensitive to these data).

We agree with the Referee's comment, which has also been highlighted by the other two Referees. PET estimation and hydrological modeling are highly dependent on input data quality. The EM-Earth dataset provides high-quality precipitation and temperature data and has been shown to perform well over Europe (Tang et al., 2022). It has undergone climatology-based bias correction and accounts for precipitation undercatch. However, since EM-Earth does not include all necessary variables for PET estimation, we utilize ERA5-Land as a complementary dataset. ERA5-Land has been demonstrated to perform better than other reanalysis datasets, including ERA5 and ERA-Interim (Muñoz-Sabater et al., 2021). Several recent global studies follow a similar strategy, combining precipitation and temperature from EM-Earth with radiation, wind speed, and other meteorological variables from ERA5-Land (Tang et al., 2023; Yin et al., 2024).

6) Summary and conclusions

a) In the very last part of the summary, you recommend an ensemble of PET formulations instead of one single formulation. While agreeing with this statement, I think that it should be supported by the study and not only occur in the end without being mentioned before. Thus, I suggest you to

use the ensemble of the different methods as a thirteenth option of the PET calculation in your manuscript.

We appreciate your suggestion. However, incorporating the 13th option would introduce additional considerations, such as creating an ensemble for each category separately or forming an ensemble across different categories. Given these complexities, we have decided to remove this point from the summary related to the ensemble option in the manuscript. This adjustment will help streamline the focus of the paper, ensuring clarity and consistency in our analysis while avoiding potential ambiguities.

7) Appendix

   a) Please add the references from where you obtained the formulations for the different methods to Table A1.

   We will add the references to Table A1 as well. Previously, references for the PET methods were included in Table 2. Adding references to Table A1 will provide a comprehensive citation for all the formulations, ensuring clarity for the readers.

8) Technical corrections

   a) The words of the title should not be capitalized.

   We will update the title in the revised manuscript.

   b) In the abstract, the third category is called "combination type", later it is called "combinational type". Please improve for consistency.

   We will homogenize it throughout the manuscript.

   c) Please note that "ERA5-Land" is written with a capital L (not "ERA5-land"). This is not consistently correct throughout the manuscript.

   We will correct it in the revised manuscript.

d) In line 75, you specify two of the components again after the abbreviations have often been used in the preceding introduction. Either, leave this away, i.e., just give the abbreviations, or give the full names of the components plus the abbreviations for all three components consistently. Furthermore, I think the sentence starting at the end of this line is redundant (stating the same as the preceding sentence in other words), please double-check and correct.

We will correct it in the revised manuscript.

e) On line 117 there is a full stop after "estimation", but then the sentence seems to continue. Please double-check and correct.

We will correct it in the revised manuscript.

f) For the McGuiness-Bordne method, first occurring on line 119, you use different ways of spelling. To my knowledge, "McGuiness-Bordne" is the correct spelling, please double-check and adjust accordingly (do not forget figures and captions). Similarly, you don't spell "BaierRobertson" and "Milly-Dunne" consistently (you sometimes write "Bair-Robertson", and in Table A1 "Milley-Dunne"). Please make sure that you use the correct spelling throughout your manuscript, including also the supplementary material (occurrence of "Milley and Dunne" in text S2). For all methods consisting of two last names, decide if you want to use a hyphen, a space, or an "and" to connect them.

We will correct typos throughout the manuscript.

g) Double-check the references given in the tables: For example, in Table 2, the reference for the Hargreaves-Samani method is in a different style than the other references.

We will update the reference.

h) It is mathematically problematic to use several letters for the same variable in an equation, e.g., "ND" could be interpreted as "N times D". Please consider reformulating.

We agree with your comment; we will correct it in the revised manuscript.

i) The first sentence of the results (line 175) is a repetition of the methods, please consider deleting.

We will delete it.

j) In the paragraph starting on line 184, it is not fully clear that the trends that are described are AET trends. I suggest you to formulate this clearer, for example by rewriting the second sentence to "…all PET methods lead to a positive AET trend in terms of median values." Similarly, in the following paragraphs describing the trends in Q and TWS (and in the results section in general), make sure that it is always clear that it isn't the PET methods that have trends but the PET methods that induce trends on the different components (if I understand your statements correctly).

We acknowledge your comment; we will elaborate this text with better clarity in the revised manuscript.

k) In line 206, you state: "…but the overall pattern of PET methods matches well with PET in all categories." This statement is unclear, did you mean a good match of PET with AET?

Our intention was to convey that PET trend patterns align with AET trend patterns. For example, if a particular PET method shows a higher trend in PET, the same method will also exhibit a higher trend in AET, though with a lower magnitude. For better clarity, we will revise this statement in manuscript.

l) In the caption of figure 3, you use mm seas-1 year-1 as a trend unit. I assume that 1 mm seas-1 year-1 in AET means that each year, 1 mm more goes to AET during the season of interest. As this seems to be an unusual unit (at least for me), I suggest you to explain this in the methods section of your manuscript already.

Thank you for your comment. For example, if the trend in the summer season is 1 mm seas$^{-1}$ year$^{-1}$. it means that each year, an additional 1 mm will be added to the summer season. Sometimes, it is also represented as mm/season per year. We will clarify this in the methods section of the manuscript.

m) In line 219, the sentence starting with "AET in the summer season" is unclearly formulated.

For better clarity in the sentence, it will be formulated as follows: *In the summer season, energy-limited and mixed catchments generally exhibit an overall positive trend in AET.*

n) The references in lines 309, 310, 313 are not formatted correctly

We will correct the reference in the revised manuscript

o) In the supplementary material (Text S1), make sure that each excerpt starts on a new line and that there is always a space between the colon after the study name and the excerpt itself. Furthermore, look for typos in the copied excerpts (e.g., in the second excerpt from the Anabalón & Sharma study, you are missing a lot of spaces). For the Shi et al. (2023) paper, it would be good to indicate which paper you mean (a or b).

We will update in the revised manuscript.

p) Please include the methods' abbreviations in the caption of Table S1.

We will add abbreviations in Table S1.

q) I assume that Figure S1 and Figure S4 show the winter season, please double-check and correct the captions.

Thank you for pointing out this typo. We will correct it in the revised manuscript.

r) In Figure S4, some of the numbers in the plot are not readable. Please see suggestions to improve Figure 6 of the manuscript (that also apply for all the other figures of this type). This should solve the problem. However, make sure that the different numbers are not written on top of each other.

We will correct it in revised manuscript.

s) For Figures S10 and S11, the correct axis labels would be "Trend in ... [...]" as the data are showing the trend and not the value of a certain component.

We will correct it in revised manuscript.

9) Individual typos

We appreciate for providing these typos; we will remove all the typos in the revised manuscript.

**References**

[revised manuscript text omitted]

---

## Author Response (AR1)

1) General Comments: Thank you for giving me the opportunity to review the manuscript "Unveiling the impact of potential evapotranspiration method selection on trends in hydrological cycle components across Europe" by Vishal Thakur and co-authors. I enjoyed reading this manuscript and consider the findings to be valuable and highly relevant: Even though potential evapotranspiration is a crucial component of the hydrological cycle, factors like data availability or convenience often play a role in the selection of a potential evapotranspiration formulation. There is no uniform way of dealing with potential evapotranspiration in hydrology, and often, different concepts (such as reference evapotranspiration and potential evapotranspiration) are used interchangeably. Therefore, studies like the one by Vishal Thakur et al., analysing the effects of different formulations of potential evapotranspiration are of great value to shed light on this often-neglected topic. The manuscript presents an analysis of the influence of different potential evapotranspiration formulations on different simulated components of the hydrological cycle. To do so, the authors make use of a large-scale modelling approach and analyse the modelling results with effective methods. They come to the important conclusion that the choice of a potential evapotranspiration method has an influence on the results when studying the hydrological cycle and its components.

We would like to thank the reviewer for their constructive feedback, which played a key role in improving the quality of our manuscript. The depth and thoughtfulness of your review have greatly aided us in presenting our findings more effectively and aligning them with broader scientific discussions. We deeply appreciate your time, effort, and dedication in offering such a thorough and thoughtful review, which has significantly enhanced the quality of our manuscript.

My main concern is that the majority of the chosen formulations (or methods) are temperature-based. Since temperatures were rising in the studied period between 1980 and 2019, the potential evapotranspiration methods based on temperature do show a positive trend. So far, it is not clear if temperature-based methods are still reliable under the conditions of a warming climate (see for example studies on the so-called "pan evaporation paradox": Li et al. (2013, 10.1002/wrcr.20202); Wang et al. (2017,10.1002/wat2.1207)). With eight out of twelve methods being temperature-based, a possible overestimation may strongly influence the results of this study.

We agree that our study includes more temperature-based PET methods compared to radiation-based and combinational methods. To strengthen our conclusions, we adopted a balanced approach by selecting four temperature-based and four complex methods (radiation + combinational). The analysis was conducted for both all trends and only statistically significant trends. The findings based on this balanced approach have been incorporated into the revised manuscript. Results based on statistically significant trends are included in the supplementary material. The DCI analysis for all trends is shown in Figure R1, while the analysis for significant trends is presented in Figure R2.

[Figure]

**Figure R1.** Spatial distribution of annual scale data concurrence index (DCI) for PET, AET, Q, and TWS. Considering eight PET methods namely Blaney-Criddle, Bair-Robertson, Hamon, Hargreaves-Samani, Milly-Dunne, Priestley-Taylor, Penman-Monteith and Modified Penman-Monteith accounts CO2.

[Figure]

**Figure R2.** Spatial distribution of annual scale data concurrence index (DCI) for PET, AET, Q, and TWS. The DCI value at each shown catchment corresponds to eight significant trends (one radiation-based, three combinational, and four from temperature-based methods). Considering eight PET methods namely Blaney-Criddle, Bair-Robertson, Hamon, Hargreaves-Samani, Milly-Dunne, Priestley-Taylor, Penman-Monteith, and Modified Penman-Monteith accounts $CO_2$. Where DCI represents data concurrence index, PET represents potential evapotranspiration, AET represents actual evapotranspiration, Q represents runoff at the outlet of the catchment, and TWS represents total water storage.

Specific comments and suggestions that will hopefully help to improve the different parts of the manuscript are listed below.

2) Introduction

 a) While Thornthwaite (1948) was the first to introduce the term "evapotranspiration", according to Miralles et al. (2020, 10.1029/2020WR028055), the concept itself was already used earlier. This should be specified. Furthermore, as the different concepts regarding evaporation (including "reference crop evapotranspiration") are often used interchangeably in hydrology, I would suggest to add a statement about this problem, e.g., referring to the Miralles et al. paper mentioned above as well as the already cited paper by Xiang et al. (2020). In addition, I would argue that there should be some indication if you consider evapotranspiration to include interception or not in your study. See also the commentary by Savenije (2004, 10.1002/hyp.5563).

Thank you very much for your helpful comments. We have elaborated development of potential evapotranspiration and added a brief introduction about reference evapotranspiration. Additionally, interception details are included in another paragraph of the introduction. Modified sentences read as follows.

*Although the concept has been in use for centuries, Thornthwaite (1948) was the first to formally introduce the term "potential evapotranspiration" in the scientific literature. A related but distinct concept is "reference crop evapotranspiration", which is sometimes used interchangeably with PET. However, these terms differ in their conceptual basis and applications. Reference crop evapotranspiration specifically estimates the water requirements of a standardized reference crop under ideal conditions, whereas potential evapotranspiration provides a broader representation of water and energy exchange processes over diverse landscapes and large regions (Xiang et al., 2020).*

*The mesoscale Hydrological Model (mHM) explicitly represents interception, where a portion of AET is derived from interception evaporation. This process is estimated as a fraction of PET*

*using a power function derived from Deardorff (1978) and Liang et al. (1994). When the evaporative demand exceeds the intercepted water, the interception storage is fully depleted. AET in mHM is mainly contributed by canpoy evaporation, soil evaporation and open water evaporation. Interception storage in the model is estimated as a function of leaf area index (LAI) that varies depending on vegetation types and season.*

b) In the first sentence of the paragraph starting at line 41, PET is stated to directly influence AET. In the second sentence, an alternative to this direct influence is presented. Therefore, the first sentence should be adjusted so that it becomes clear that this is just one possibility. Furthermore, depending on how interception is dealt with, this may be a necessary component to add in the list of fluxes that make up AET (in the second sentence of this paragraph).

We revised the text with more clarity. Revised text are as follows:

*PET influences AET and consequently impacts the estimation of infiltration, Q, and TWS in hydrological models. PET can have direct as well as indirect influence on AET. In hydrological models, AET is estimated by either separately determining water surface evaporation, soil evaporation, and vegetation transpiration and then combining these based on land use patterns or by first assessing potential evapotranspiration and subsequently adjusting it to actual evapotranspiration using the soil moisture extraction function (Zhao et al., 2013).*

3) Methods and data

a) In the beginning of section 2.2, you give the study by Hersbach et al. (2020) as a reference for ERA5-Land. I would argue that the suitable reference there is the paper by Muñoz-Sabater et al. (2021), that you cite later in the manuscript. Furthermore, in line 97, there is a reference missing for EM-Earth and SC-Earth, or it is not clear to me if the Tang et al. (2022) reference belongs to this. If the EM-Earth data are based on ERA5 data (and not on ERA5-Land data), I would suggest to include the reference to the Hersbach et al. (2020) paper there.

We corrected reference in revised manuscript (Muñoz-Sabater et al.; 2021). The EM-Earth dataset is distinct and primarily generated by integrating multiple data sources, with SC-Earth and ERA5 being the two major contributors. To improve clarity, we added references as follows.

*The EM-Earth dataset (Tang et al., 2022) is derived from observed station data SC-Earth (Tang et al., 2021) and ERA5 data (Hersbach et al., 2020).*

b) I assume the AET, Q, and TWS data in Figure 1c to be simulated values. In my opinion, this needs to be mentioned in the figure caption as it only becomes clear after reading further.

The caption for Figure 1(c) has been revised for improved clarity as follows:

*Annual time series of simulated hydrological components from the mesoscale hydrological model for each representative catchment and PET estimation method. (TH: Thornthwaite, BR: Bair-Robertson, BC: Blaney-Criddle, OD: Oudin, MB: McGuinness-Borden, HM: Hamon, HS: Hargreaves-Samani, JH: Jensen-Haise, MD: Milly-Dunne, PT: Priestley-Taylor, PM: Penman-Monteith, $CO_2$: Modified Penman-Monteith accounts $CO_2$.). All units are in mm year$^{-1}$.*

c) In line 122, you state that some of the temperature-based methods also use some extraterrestrial radiation term calculated based on latitude. Potentially, this could already be mentioned earlier, when the definition of the temperature-based category is given (i.e., it could become clearer that also radiation terms can be required and a method is still considered to be temperature-based).

For better clarity entire paragraph (Line 117 - 129 ) was rewritten. *We incorporate 12 PET methods at a daily scale from all three categories of estimation: temperature, radiation, and combinational methods (Table 2). Temperature-based methods require temperature data, which can include average temperature, minimum temperature, or maximum temperature. Additionally, PET methods that incorporate extraterrestrial radiation are also considered under this category. Combinational-based methods require a larger number of variables compared to temperature- and radiation-based methods to estimate various physical terms, such as wind speed and surface pressure (Table 2). Most temperature-based methods use only daily average temperature*

*(Thornthwaite, Oudin, Hamon, Jensen-Haise, McGuinness-Bordne, and Blaney-Criddle), while Baier-Robertson employs both minimum and maximum daily temperatures. Some of these methods also include an extraterrestrial radiation term in their formulation. However, since this radiation term is calculated based on latitude and follows a consistent annual cycle varying only with the calendar date, only temperature data is needed for PET calculation. We utilize only one radiation-based method, Milly-Dunne PET, which requires only net radiation data to estimate PET. The combinational methods, such as Penman-Monteith and Priestley-Taylor, have a stronger physical basis. In our analysis, all these physical terms are estimated following Allen (1998). Additionally, within the combinational category, we employ the modified Penman-Monteith (CO2) method, which accounts for temporal variations in changing carbon dioxide concentrations. Formulation details, including mathematical equations and associated constants for each PET method, are provided in Table A1.*

d) When you list the terms that are required for the Penman-Monteith and the Priestley-Taylor method, I would suggest to list all the terms (as it's only two equations), instead of giving some examples and concluding with "etc.". Alternatively, you could give one or two examples and then place a reference to the table where you list all the input data required for each method.

We have incorporated two examples and provided a reference to the relevant table in the manuscript. The revised version of the paragraph has been addressed in the previous section, and the content is outlined below.

*We incorporate 12 PET methods at a daily scale from all three categories of estimation: temperature, radiation, and combinational methods (Table 2). Temperature-based methods require temperature data, which can include average temperature, minimum temperature, or maximum temperature. Additionally, PET methods that incorporate extraterrestrial radiation are also considered under this category. Combinational-based methods require a larger number of variables compared to temperature- and radiation-based methods to estimate various physical*

*terms, such as wind speed and surface pressure (Table 2). Most temperature-based methods use only daily average temperature (Thornthwaite, Oudin, Hamon, Jensen-Haise, McGuinness-Bordne, and Blaney-Criddle), while Baier-Robertson employs both minimum and maximum daily temperatures. Some of these methods also include an extraterrestrial radiation term in their formulation. However, since this radiation term is calculated based on latitude and follows a consistent annual cycle varying only with the calendar date, only temperature data is needed for PET calculation. We utilize only one radiation-based method, Milly-Dunne PET, which requires only net radiation data to estimate PET. The combinational methods, such as Penman-Monteith and Priestley-Taylor, have a stronger physical basis. In our analysis, all these physical terms are estimated following Allen (1998). Additionally, within the combinational category, we employ the modified Penman-Monteith ($CO_2$) method, which accounts for temporal variations in changing carbon dioxide concentrations. Formulation details, including mathematical equations and associated constants for each PET method, are provided in Table A1.*

e) For me, the description of the modelling part (paragraph starting on line 139) is hard to follow. Therefore, I ask you to give more information about this part of your study: You do one model run per catchment and PET method (553 x 12 = 6636). Is there any model calibration? If yes, please elaborate on that: What objective function(s) did you use? What was the spatial resolution? If no, where do you take the model settings from? Are these the same for all catchments? How do you make sure that the parameterization that you use matches your catchments? Did you think about equifinality and how other possible parameterizations could influence your results? How well did the model perform for the different catchments? In the very end of the discussion, the reader learns that a default parameterization was used. Please add this to the methods part and make sure that the questions listed above become clear.

We have included model performance and associated details in the modelling part. The newly added details are as follows:

*In this research default parametrization is used for model setup. The default parameterization of mHM has been shown to perform well in previous studies (Kumar et al., 2013; Rakovec et al., 2016). Furthermore, it has been demonstrated as one of the best-performing configurations compared to other large-scale hydrological models (Samaniego et al., 2019). Additionally, our assessment of model performance is consistent with the findings of Samaniego et al. (2019). Our model evaluation against discharge shows that the median Kling–Gupta Efficiency (KGE) ranges from 0.60 to 0.75 across most PET methods (Figure R3).*

[Figure]

**Figure R3.** Evaluation of the hydrological model (mHM) performance using simulated monthly streamflow across 553 catchments, forced with EM-Earth meteorological data. The figure presents the cumulative frequency distributions of the Kling–Gupta efficiency (KGE) and its three components: correlation (r), variability ratio (alpha), and bias ratio (beta), providing insights into the model's performance across different PET methods.

f) Related to the comment above, where does this default parameterization come from? Was there a method for potential evapotranspiration involved when this default parameterization was obtained? If so, may this have varying effects on the results based on this (or similar) methods (that can potentially profit from compensating effects) and the results based on different methods (that cannot profit from any compensating effects)? Potentially and if possible (may be limited due to data availability constraints) it would be interesting to compare the current results to the results of a calibrated model (i.e., calibrated for each of the different methods) – you also include a similar remark in the discussion. With additional calibrations that make use of the different methods, it could become clearer if the (assumed) use of one method for the default parameterization affects the results. If additional calibrated model runs are not possible, I would suggest you to include some text on the potential effects of the default parameterization in the discussion.

We decided not to compare calibrated and uncalibrated model results in this manuscript. This step was taken to avoid unnecessarily lengthening the manuscript and due to time constraints, as it is the first my PhD paper. However, we propose this comparison as a direction for future research and intend to explore it further. In the discussion, we address the impact of default parameterization and reference studies that have employed this approach. Additionally, we discuss the model performance of the PET method, which served as the basis for these parameterizations. The revised text is as follows:

*Large-scale hydrological models, including mHM, typically rely on default parameterization. In mHM, these parameters were initially developed using German basins, as outlined by Samaniego et al. (2010); Kumar et al. (2013). Since then, mHM has been extensively tested across various basins and hydrological variables (Rakovec et al., 2016; Samaniego et al., 2019; Boeing et al., 2024). For instance, Rakovec et al. (2016) evaluated discharge simulations across 400 European catchments using 36 parameter sets, demonstrating consistent model performance regardless of*

*parameterization, thereby reinforcing confidence in the model's reliability. Similar approaches have been applied in global water models; Beck et al. (2017) employed ensemble parameters derived from ten catchments, while Kumar et al. (2013) tested default parameters from European basins across 80 American catchments with varying climatic conditions. While these studies demonstrate the robustness of default parameterization, investigating how PET-specific calibration affects hydrological trends could provide valuable insights for future research. Notably, the Hargreaves-Samani PET method, used in developing these parameters in mHM, demonstrated best model performance in this study but did not consistently stand out compared to other PET methods in trend analysis across hydrological components. Moreover, the study is confined to temperate European catchments, leaving a gap in assessing arid and tropical climates, where distinct patterns may emerge. While the Modified Penman-Monteith method accounts for $CO_2$, it did not exhibit substantial differences compared to the Penman-Monteith method, indicating the need for further exploration of this method. It would be interesting to assess their impact under changing climate conditions and their implications for trend assessments.*

g) I suggest you to clarify in lines 158 and 159 that you compare winter data to winter data, spring data to spring data, etc. so that it is immediately clear that you do not compare all four seasons with each other.

Following your comments, Lines 158–159 were further elaborated as follows:
*The slopes of each PET method were analyzed exclusively within seasons (for eaxmple winter season compared with winter season), without cross-seasonal comparisons.*

h) In section 2.3.4, you describe the modification of the DCI by including also non significant trends to be able to include all trend estimates. Are you sure that you are not including too much noise by also counting very weak positive and negative trends? I suggest to at least show that your findings do not change if only reasonably significant trends are considered.

These are reasonable concerns, also raised by the other two Referees. We agree that including non-significant trends can potentially introduce noise; however, in our case, this choice was deliberate and based on the following rationale:

Including all trends enables the evaluation of spatial coherence in directional changes, which is central to our study. If weak trends were merely random noise, we would expect their directions to be evenly distributed across space. However, we observe a strong spatial alignment, even among weaker trends, which suggests an underlying signal beyond random variability. Moreover, the reliance solely on statistical significance (e.g., $p < 0.05$) can be problematic, as noted in the editorial by Ronald L. Wasserstein and Lazar (2019) in The American Statistician. They emphasize that "no single index should substitute for scientific reasoning", and caution against rigid thresholding, which may obscure meaningful but subtle patterns.

Importantly, the scope of our study is not to isolate only the strongest or most significant trends, but rather to examine how the choice of PET method influences the direction and magnitude of all trend estimates. This broader perspective allows us to assess the systematic impact of PET selection across the full spectrum of changes (both strong and weak) which is essential for evaluating methodological sensitivity and spatial consistency. Limiting the analysis only to significant trends would risk missing meaningful shifts in regions where variability is high or records are short, even though such areas may still exhibit consistent directional behavior under different PET formulations.

That said, to directly address your concern, we conducted a comparative analysis using the Mann–Kendall trend test, computing the DCI: (1) Using all trend estimates (main manuscript), and (2) Using only statistically significant trends ($p < 0.05$; shown in Supplementary Figures R4 and R5). This comparison confirms that the main patterns and conclusions are robust, regardless of whether all trends or only significant ones are included. We have clarified this approach and justification in the revised Methods, Results, and Conclusions sections.

[Figure]

**Figure R4.** Spatial distribution of annual scale data concurrence index (DCI) for PET, AET, Q, and TWS by considering only significant trend at 95% significance level. PET represents potential evapotranspiration, AET represents actual evapotranspiration, Q represents runoff at the outlet of the catchment and TWS represents total water storage.

[Figure]

**Figure R5.** Spatial distribution of seasonal scale data concurrence index (DCI) for PET, AET, Q, and TWS by considering only significant trend at 95 % significance level. PET represents potential evapotranspiration, AET represents actual evapotranspiration, Q represents runoff at the outlet of the catchment and TWS represents total water storage.

4) Results

a) In the first paragraph of section 3.1, you describe the different trends in PET for the three categories of catchments. As PET describes a potential, i.e., the maximum ET that could be achieved if there was no water-limitation, the trend in PET should not depend on if a catchment is waterlimited or energy limited. Thus, I suggest to formulate this paragraph differently, such that it is clear that the PET trend is influenced by other factors, but not by the availability of water.

We revised this paragraph by linking the factors influencing PET and relating them to the context of catchment types. The updated text are as follows:

*By applying the Theil-Sen slope method, we observe that changes in PET depend on the choice of PET estimation formulation (Figure 2). Considerable variability is observed among the PET methods, with median slopes ranging from slightly positive to 6 mm year$^{-1}$ during the 1980–2019 period. The Jensen-Haise method shows the highest change among all methods across different catchment categories. Generally, changes in PET are higher in water-limited than in energy-limited catchments. This difference arises since temperature-based methods depend on temperature changes. Conversely, combinational methods are influenced by more than one meteorological variable (temperature, wind speed, radiation, etc.). When we consider only statistically significant trends ($p < 0.05$), we observe that temperature-based PET methods consistently demonstrate statistically significant positive trends for most of the catchments (Table S2). Radiation-based and combinational methods account for fewer catchments than temperature-based PET methods (Table S2), implying that they generate weaker trends compared to the temperature-based PET methods. In addition, overall trend variability among the PET methods decreases from energy-limited to water-limited catchments, irrespectively of trend significance.*

b) In the last paragraph of section 3.2, you suddenly write about December and not about the winter season (line 234). If you did monthly analyses, this needs to be stated clearly in the methods section. Otherwise, December may be the wrong term here.

*We corrected typing mistake in this line. The paragraph was intended for seasonal comparison. Instead of "December," it should refer to the "winter season." The corrected sentence for lines 234–235 is as follows:*

*In contrast, for energy-limited catchments, the winter and summer seasons are the main contributors, with their impacts varying depending on the selected PET method (Figure S20)*

c) As already mentioned in the general comments: With the rising temperatures, all the temperature-based methods tend to show positive trends. Since most PET methods included are temperature-based, they have the largest influence on the DCI. It would be good to test if this unequal distribution of the different types influences your results.

We conducted an analysis using an equal distribution of PET methods, selecting four temperature-based methods, one radiation and three combinational methods. We have included it in the results section of the manuscript. A detailed discussion of the observed result is provided in 1) General Comments.

d) To make the part about the combinations of hydrological cycle component changes (section 3.4) easier to understand also before studying Figure 6, I suggest you to add an example of a "possible combination of hydrological cycle component changes" after you introduce this term on lines 269 and 270. Similarly, it may be good to inform the reader what you mean with "the first five hydrological cycle combinations" (lines 270 and 271).

Along with the updates in this figure, we have also refined the text in this section. A dedicated paragraph now provides specific details regarding the pattern of hydrological cycle components and the criteria for selecting specific patterns. Newly added text are as follows:

*In the previous section, we compared PET methods and their influence on individual hydrological components (P, AET, Q, and TWS), including agreement among PET methods. Here, we assess the influence of PET methods on patterns of key hydrological components across European catchments, identifying the most prevalent trend patterns where components (P, AET, Q, and*

*TWS) concurrently increase or decrease. For instance, one pattern involves all components showing positive trends. Figure 6 summarizes these patterns, presenting the average and total number of catchments associated with each PET method. The analysis includes only the five most frequent patterns, covering the majority of catchments, excluding those patterns with minimal representation.*

e) In lines 272 and 273 you state that the temperature-based methods account for more catchments with positive trends across all hydrological cycle components than combinational methods (and you put this in comparison to the Blaney-Criddle method leading to positive trends in all components for most catchments). This statement is not clear: Do you want to say that the temperature-based methods lead to the "all positive" combination in more catchments than the combinational methods? If so, this is not true for the Baier-Robertson method, if I interpret Figure 6 correctly. Please reformulate this statement and reconsider if this should be formulated as a comparison to the first part of the sentence.
We wrote the entire section with better clarity.

f) The sentence regarding the combinations with AET+ and TWS- (lines 280 and 281) should come before the statement about the last five combinations to be consistent with the order in Figure 6.
We wrote the entire section with better clarity.

g) I think that Figure 6 should be improved so that it can be grasped more quickly and easily:
The figure has been replotted to enhance its visual clarity and convey information more effectively. Additionally, we have rewritten the entire section in the revised manuscript, aiming to improve its structure and readability for the audience.

■ I suggest you to put the lower part (in the following called "combinations") to the top and the upper part (in the following called "catchment count") to the bottom. The reader first wants to know what combination we are looking at, and then wants to look at the results for this

combination. I see that you used the "combinations" part as an "axis label", but this is not so clear when looking at the plot in the beginning as it looks more like two different parts.

We have systematically updated all figures to include both all trends and statistically significant trends. Figure R7 is provided as a demonstration of this approach.

■ For the "combinations", consider displaying them differently. For example, a simple table-like graphic in which you could even work with additional colours would in my opinion be easier to interpret than the current way to display the combinations, see below. If you decide to do so, do not forget to change the caption.

We have incorporated a tabular format to present the combinations, as illustrated in the first panel of Figure R7.

■ For the "catchment count", I think it would improve the readability of the figure if not all bars were the same height, i.e., if the height of the bar would decrease from left to right, proportionally to the higher number of catchments that are contained in the bars on the left than in the ones on the right. This way, it would become clearer which combinations occur how often. For that, you could consider using horizontal instead of vertical bars to gain more space (this would also allow you to use the "combinations" part as "axis labels", but then of the vertical axis).

We have incorporated a bar plot that shows average catchment count, as illustrated in the second panel of Figure R7.

■ Please double-check the catchment count for each method. To my understanding, the number of catchments per method contained in this plot should always sum up to 553. However, for example for the Thornthwaite method, the number of catchments only sums up to 240+156+59+70+24+2=551.

There are 14 possible patterns (Figure R6). The excluded patterns were not considered because they were represented by only a very small number of catchments. The remaining

two catchments from the Thornthwaite method belonged to the (PRE-, AET+, Q+, TWS-) pattern. Including these catchments brings the total count to 553 (240 + 156 + 59 + 70 + 24 + 2 + 2). This approach ensures that all catchments are considered in the analysis. However, in the revised manuscript, we focus only on the top five patterns, excluding those with fewer catchments (Figure R7).

[Figure]

**Figure R6.** Pattern of different hydrological cycle components and the corresponding influence of PET methods on an annual scale. The first panel represents different patterns of hydrological cycle components. Each vertical column in this table corresponds to one pattern of hydrological cycle. For example, the first column is filled with '+' signs, indicating that all hydrological components (PRE, AET, Q, and TWS) exhibit positive changes. The '+' and '-' signs denote positive and negative changes in the respective components. The second panel shows the number of catchments associated with each PET method for the corresponding hydrological cycle patterns. The color of each cell represents a specific PET method, and each column aligns with the hydrological cycle pattern represented in the corresponding column of Panel One. Where PRE is precipitation, AET is actual evapotranspiration, Q is runoff and TWS is total water storage. Abbreviations used for different PET methods are TH: Thornthwaite, BR: Bair-Robertson, BC: Blaney-Criddle, OD: Oudin, MB: McGuinness-Borden, HM: Hamon, HS: Hargreaves-Samani, JH: Jensen-Haise, MD: Milly-Dunne, PT: Priestley-Taylor, PM: Penman-Monteith, $CO_2$: Modified Penman-Monteith accounts $CO_2$.

[Figure]

**Figure R7.** Pattern of different hydrological cycle components and the corresponding influence of PET methods on an annual scale. The first panel represents different patterns of hydrological cycle components. Each vertical column in this table corresponds to one pattern of hydrological cycle. For example, the first column is filled with '+' signs, indicating that all hydrological components (PRE, AET, Q, and TWS) exhibit positive changes. The '+' and '-' signs denote positive and negative changes in the respective components. In the second panel, each bar represents the average number of catchments for each hydrological cycle pattern. The third panel shows the number of catchments associated with each PET method for the corresponding hydrological cycle patterns. The color of each cell represents a specific PET method, and each column aligns with the hydrological cycle pattern represented in the corresponding column of Panel One. Where PRE is precipitation, AET is actual evapotranspiration, Q is runoff and TWS is total water storage. Abbreviations used for different PET methods are TH: Thornthwaite, BR: Bair-Robertson, BC: Blaney-Criddle, OD: Oudin, MB: McGuinness-Borden, HM: Hamon, HS: Hargreaves-Samani, JH: Jensen-Haise, MD: Milly-Dunne, PT: Priestley-Taylor, PM: Penman-Monteith, $CO_2$: Modified Penman-Monteith accounts $CO_2$.

- As the Thornthwaite method and the Blaney-Criddle method are displayed in neighbouring fields for the seventh combination, please consider not using red and green for these two methods as they are hard to distinguish for colour-blind people. Alternatively, please use patterns in addition, to make it possible to distinguish the two colours.

  *We updated the colors in the revised manuscript.*

5) Discussion

   a) In line 302, you compare your results to the ones in the study by Hanselmann et al. (2024). Please note that while the authors of this study are affiliated in Poland, the research has been conducted for Spitsbergen.

      *We revised the sentence and updated it with more references. revised text in the manuscript is as follows:*

      *Similar findings have been reported for various regions by other researchers (Kingston et al., 2009; Hanselmann et al., 2024; Seiller and Anctil, 2016).*

   b) In the paragraph starting on line 336, you discuss your results as well as the results of the study by Teuling et al. (2019). Based on the Teuling et al. paper as well as based on the PET methods that you considered in your study, I assume that you mean the Penman-Monteith method in this paragraph when you write about the Penman method. Please double-check and correct. The same issue occurs again in line 384.

      *We corrected the typo penman to Penman-Monteith.*

   c) Later in the same paragraph, where you discuss a possible drying hydrological cycle, I think that a discussion of the study by Milly & Dunne (2016) that you refer to elsewhere in your manuscript is lacking as they studied this topic.

      *Our intention was to focus on a specific combination or pattern of hydrological cycle components. To enhance clarity, a minor modification was made to lines 340–345. Updated text reads as*

*...Similarly, many catchments demonstrate positive changes in AET and negative changes in Q, TWS, and PRE (drying condition of the hydrological cycle). The number of catchments demonstrating these trends varies depending on the choice of PET method. Massari et al. (2022) found that over Europe runoff deficit are more pronounced in water-limited regions due to increased AET, whereas energy-limited catchments exhibit smaller deficits. During these drying conditions, AET is further influenced by reductions in TWS (Massari et al., 2022)....*

d) In line 368, you write about methods that consistently overestimate PET. It is unclear to me how you define overestimation here: Do you just consider the methods leading to the highest estimates to be overestimating PET? If so, I would not consider it to be surprising that the catchments shift to the energy-limited category (as all the higher estimates are excluded). If all the low estimates would be excluded, the catchments would probably shift to the water-limited category. Please elaborate more on what you mean there.

Our objective was to demonstrate that the choice of PET method, along with the grouping of methods, has a substantial impact on catchment classification. We simplify the line 368 text in the revised manuscript. Table S1 in the manuscript illustrates catchment shifts when different PET methods as well as group of PET methods.

e) When you discuss the limitations of your study, you state that on the one hand, the PET methods were found to be sensitive to the input data, and on the other hand, that hydrological models are sensitive to the precipitation input. Based on this, I think it would be good to write something about the data quality of the input data that you used (as you basically state that your results are highly sensitive to these data).

We have incorporated the details and quality of the input data. The additional text is as follow:

*The EM-Earth dataset provides high-quality precipitation and temperature data and has been shown to perform well over Europe (Tang et al., 2022). It has undergone climatology-based bias correction and accounts for precipitation undercatch. However, since EM-Earth does not include*

*all necessary variables for PET estimation, we utilize ERA5-Land as a complementary dataset. ERA5-Land has been demonstrated to perform better than other reanalysis datasets, including ERA5 and ERA-Interim (Muñoz-Sabater et al., 2021). Nonetheless, its limitations in hydrological modeling have been acknowledged by Clerc-Schwarzenbach et al. (2024); Tarek et al. (2020). Several recent global studies follow a similar strategy, combining precipitation and temperature from EM-Earth with radiation, wind speed, and other meteorological variables from ERA5-Land (Tang et al., 2023; Yin et al., 2024; Rakovec et al., 2023). These meteorological data combinations, along with the simulated hydrological components derived from them, demonstrate lower uncertainty across Europe (Tang et al., 2023).*

6) Summary and conclusions

a) In the very last part of the summary, you recommend an ensemble of PET formulations instead of one single formulation. While agreeing with this statement, I think that it should be supported by the study and not only occur in the end without being mentioned before. Thus, I suggest you to use the ensemble of the different methods as a thirteenth option of the PET calculation in your manuscript.

We removed this point from the summary related to the ensemble option in the manuscript. This adjustment helped streamline the focus of the paper, ensuring clarity and consistency in our analysis while avoiding potential ambiguities. The updated text reads as:

*Our research demonstrates the critical role of PET method selection and its implications for quantifying fluctuations in the hydrological cycle. Our findings reveal that two methods notably deviate from the others. Specifically, the Jensen-Haise method shows higher trend values, while the Milly-Dunne method exhibits lower trends in water-limited catchments. Consequently, we recommend exercising caution when applying these methods as they appear to be outliers. Despite these variations, the PET methods generally agree that atmospheric moisture demand is increasing across Europe, reflecting recent shifts in temperature and radiation. The observed*

*variability in trend magnitudes emphasizes the importance of careful PET method selection to ensure robust and representative assessments of hydrological trends.*

7) Appendix

   a) Please add the references from where you obtained the formulations for the different methods to Table A1.

   We added the references to Table A1 as well.

8) Technical corrections

   a) The words of the title should not be capitalized.

   We updated the title in the revised manuscript.

   *Unveiling the impact of potential evapotranspiration method selection on trends in hydrological cycle components across Europe*

   b) In the abstract, the third category is called "combination type", later it is called "combinational type". Please improve for consistency.

   We have homogenized it throughout the manuscript.

   c) Please note that "ERA5-Land" is written with a capital L (not "ERA5-land"). This is not consistently correct throughout the manuscript.

   We have corrected it in the revised manuscript.

   d) In line 75, you specify two of the components again after the abbreviations have often been used in the preceding introduction. Either, leave this away, i.e., just give the abbreviations, or give the full names of the components plus the abbreviations for all three components consistently. Furthermore, I think the sentence starting at the end of this line is redundant (stating the same as the preceding sentence in other words), please double-check and correct.

   We updated it in the revised manuscript.

   *(....AET, Q, and TWS....)*

e) On line 117 there is a full stop after "estimation", but then the sentence seems to continue. Please double-check and correct.

We have corrected it in the revised manuscript.

f) For the McGuiness-Bordne method, first occurring on line 119, you use different ways of spelling. To my knowledge, "McGuiness-Bordne" is the correct spelling, please double-check and adjust accordingly (do not forget figures and captions). Similarly, you don't spell "BaierRobertson" and "Milly-Dunne" consistently (you sometimes write "Bair-Robertson", and in Table A1 "Milley-Dunne"). Please make sure that you use the correct spelling throughout your manuscript, including also the supplementary material (occurrence of "Milley and Dunne" in text S2). For all methods consisting of two last names, decide if you want to use a hyphen, a space, or an "and" to connect them.

We corrected typos throughout the manuscript.

*Correct spelling is McGuinness-Bordne.*

g) Double-check the references given in the tables: For example, in Table 2, the reference for the Hargreaves-Samani method is in a different style than the other references.

We updated the correct reference in revised manuscript.

h) It is mathematically problematic to use several letters for the same variable in an equation, e.g., "ND" could be interpreted as "N times D". Please consider reformulating.

We corrected it in the revised manuscript.

i) The first sentence of the results (line 175) is a repetition of the methods, please consider deleting.

It is corrected in the revised manuscript.

j) In the paragraph starting on line 184, it is not fully clear that the trends that are described are AET trends. I suggest you to formulate this clearer, for example by rewriting the second sentence to "...all PET methods lead to a positive AET trend in terms of median values." Similarly, in the following paragraphs describing the trends in Q and TWS (and in the results section in general),

make sure that it is always clear that it isn't the PET methods that have trends but the PET methods that induce trends on the different components (if I understand your statements correctly).

*We have incorporated the suggestions. Also, it is to note that we dedicate a single paragraph to each component.*

k) In line 206, you state: "…but the overall pattern of PET methods matches well with PET in all categories." This statement is unclear, did you mean a good match of PET with AET?

*We revised it in the updated text.*

*Overall, the AET trend patterns (high and low trends) for energy-limited and mixed catchments are similar to the trends in PET for these catchments, regardless of trend magnitude, with a few exceptions such as Blaney-Criddle and Jensen-Haise (Figure 2).*

l) In the caption of figure 3, you use mm seas-1 year-1 as a trend unit. I assume that 1 mm seas-1 year-1 in AET means that each year, 1 mm more goes to AET during the season of interest. As this seems to be an unusual unit (at least for me), I suggest you to explain this in the methods section of your manuscript already.

*This suggestion is incorporated in the method section of the revised manuscript.*

*The units of trend at the annual scale are expressed as mm $year^{-1}$, while at the seasonal scale, they are represented as mm $seas^{-1}$ $year^{-1}$. For instance, a summer season trend of 1 mm $seas^{-1}$ $year^{-1}$ indicates that each year, an additional 1 mm is added to the summer season.*

m) In line 219, the sentence starting with "AET in the summer season" is unclearly formulated.

*We wrote the entire section 3.2 in the revised manuscript.*

n) The references in lines 309, 310, 313 are not formatted correctly

*We updated references in the revised manuscript*

o) In the supplementary material (Text S1), make sure that each excerpt starts on a new line and that there is always a space between the colon after the study name and the excerpt itself. Furthermore,

look for typos in the copied excerpts (e.g., in the second excerpt from the Anabalón & Sharma study, you are missing a lot of spaces). For the Shi et al. (2023) paper, it would be good to indicate which paper you mean (a or b).

We corrected all the typos and spacing in the revised manuscript.

p) Please include the methods abbreviations in the caption of Table S1.

We added the abbreviation in the Table S1.

q) I assume that Figure S1 and Figure S4 show the winter season, please double-check and correct the captions.

We corrected this typo in the revised manuscript.

r) In Figure S4, some of the numbers in the plot are not readable. Please see suggestions to improve Figure 6 of the manuscript (that also apply for all the other figures of this type). This should solve the problem. However, make sure that the different numbers are not written on top of each other.

We have corrected this issue in all similar figures throughout the manuscript.

s) For Figures S10 and S11, the correct axis labels would be "Trend in ... [...]" as the data are showing the trend and not the value of a certain component.

In response to Reviewer three suggestions, we have merged and moved these figures from the supplementary section to Section 3.5 in the main manuscript. Additionally, the figures were revised along with the spatial variation in precipitation.

9) Individual typos

We have addressed all the typos as noted by the reviewer.

Referee #2

Referee's comments are in black text

Authors's response are in blue text

1) General Comments: In this work, "Unveiling the Impact of Potential Evapotranspiration Method Selection on Trends in Hydrological Cycle Components Across Europe," the authors assess 12 potential evapotranspiration (PET) formulations across the European continent using regional hydrological modelling to quantify their impact on PET trends and their implications for the main hydrological cycle components: actual evapotranspiration (AET), total water storage (TWS), and runoff (Q). They conclude that the PET model selection conditions the simulated trends and influences the analyzed hydrological component. The paper reads well; I enjoyed it while reading it. Moreover, I think the study is relevant for the catchment hydrological community; the impact of PET formulations has usually been overcome in calibration frameworks, and assessing its actual impact at a continental scale is a valuable result.

We thank the Referee #2 for their constructive feedback on this manuscript. The valuable comments provided contributed significantly to enhancing the quality of the manuscript. We were grateful for your time and effort invested in improving our study.

However, I have some concerns, especially regarding the methodological approach. On the one hand, I missed information about the model framework, for instance, what baseline calibration you used and how the main analyzed hydrological cycle components are linked to the model. This is key to understanding the impact of your analysis. On the other hand, the authors talked about trends, but no proper statistical trend analysis has been performed.

Details regarding the model setup and its performance are thoroughly addressed in Referee #1 comments under Section 3) Methods and data, specifically in subsections (e) and (f). Additionally, we conducted an analysis considering significant trends, as discussed in Referee #1 comment on

subsection (h) of Section 3) Methods and data. The revised text related to modelling part and model performance reads as:

*......In this research default parametrization is used for model setup. The default parameterization of mHM has been shown to perform well in previous studies (Kumar et al., 2013; Rakovec et al., 2016). Furthermore, it has been demonstrated as one of the best-performing configurations compared to other large-scale hydrological models (Samaniego et al., 2019). Additionally, our assessment of model performance is consistent with the findings of Samaniego et al. (2019). Our model evaluation against discharge shows that the median Kling–Gupta Efficiency (KGE) ranges from 0.60 to 0.75 across most PET methods (Figure R3)....*

*Large-scale hydrological models, including mHM, typically rely on default parameterization. In mHM, these parameters were initially developed using German basins, as outlined by Samaniego et al. (2010); Kumar et al. (2013). Since then, mHM has been extensively tested across various basins and hydrological variables (Rakovec et al., 2016; Samaniego et al., 2019; Boeing et al., 2024). For instance, Rakovec et al. (2016) evaluated discharge simulations across 400 European catchments using 36 parameter sets, demonstrating consistent model performance regardless of parameterization, thereby reinforcing confidence in the model's reliability.....*

Specific comments

2) Introduction

a) In paragraph one (lines 24-33), it would be nice to briefly mention that other concepts like reference evapotranspiration are widely used when computing AET.

We have addressed the discussion on reference evapotranspiration in Referee #1's comment under subsection (a) of the Introduction section. The text that we included in the revised manuscript is as follows: *Potential evapotranspiration (PET) is the potential to evaporate water from the land surface to the atmosphere without Any limitation to water availability. Although the concept has been in use for centuries, Thornthwaite (1948) was the first to formally introduce the term*

*"potential evapotranspiration" in the scientific literature. A related but distinct concept is "reference crop evapotranspiration", which is sometimes used interchangeably with PET. However, these terms differ in their conceptual basis and applications. Reference crop evapotranspiration specifically estimates the water requirements of a standardized reference crop under ideal conditions, whereas potential evapotranspiration provides a broader representation of water and energy exchange processes over diverse landscapes and large regions (Xiang et al., 2020).*

b) In paragraph two (lines 24-33), I would also include some sentences explaining why there have been more than 50 models for computing PET. Are they physically based formulations? Are they empirical and therefore linked to where they were initially formulated?

We added further details about PET formulation in the introduction section of the revised manuscript. The updated text is as follows:

*Out of these 100+ methods, the majority are temperature-based methods (40+), followed by radiation-based methods (30+) and combination-based methods (10+). Many of these empirical methods were initially developed and tested for particular regional scales or climatic conditions. For instance, the Thornthwaite method is most suitable for humid climates, while the Hargreaves-Samani method is particularly effective in arid and semi-arid regions. Similarly, the Hamon method is suitable for all climates. All methods in these three categories incorporate several assumptions (climatic conditions and data availability), resulting in significant differences in their estimates (Lu et al., 2005).*

c) In the third paragraph (lines 41-47). Since your study assesses the impact of PET selection in other water cycle components, I would include more context and references about the connection between different components.

We extended this introduction paragraph further in relation with other hydrological cycle components along with AET related modification as suggested by Referee #1. The updated text in this paragraph is as follows:

*PET influences AET and consequently impacts the estimation of infiltration, Q, and TWS in hydrological models. PET can have direct as well as indirect influence on AET. In hydrological models, AET is estimated by either separately determining water surface evaporation, soil evaporation, and vegetation transpiration and then combining these based on land use patterns or by first assessing potential evapotranspiration and subsequently adjusting it to actual evapotranspiration using the soil moisture extraction function (Zhao et al., 2013). The mesoscale Hydrological Model (mHM) explicitly represents interception, where a portion of AET is derived from interception evaporation. This process is estimated as a fraction of PET using a power function derived from Deardorff (1978) and Liang et al. (1994). When the evaporative demand exceeds the intercepted water, the interception storage is fully depleted. Interception storage in the mHM is estimated as a function of leaf area index (LAI) that varies depending on vegetation types and season. AET in mHM is mainly contributed by canpoy evaporation, soil evaporation and open water evaporation. AET, being a key component of the water balance, affects the estimation of other water balance components (Q and TWS). While Q remains relatively insensitive, AET and TWS are more responsive to the choice of PET method (Bai et al., 2016). Hence, uncertainty in PET estimation influences the quantification of change in water cycle components.*

3) Methods and data

a) Could you elaborate a bit about the quality of your data? Later in the discussion, you mentioned that their uncertainties were important to your results.

We have incorporated the details and quality of the input data. The additional text is as follows:

*The EM-Earth dataset provides high-quality precipitation and temperature data and has been shown to perform well over Europe (Tang et al., 2022). It has undergone climatology-based bias*

*correction and accounts for precipitation undercatch. However, since EM-Earth does not include all necessary variables for PET estimation, we utilize ERA5-Land as a complementary dataset. ERA5-Land has been demonstrated to perform better than other reanalysis datasets, including ERA5 and ERA-Interim (Muñoz-Sabater et al., 2021). Nonetheless, its limitations in hydrological modeling have been acknowledged by Clerc-Schwarzenbach et al. (2024); Tarek et al. (2020). Several recent global studies follow a similar strategy, combining precipitation and temperature from EM-Earth with radiation, wind speed, and other meteorological variables from ERA5-Land (Tang et al., 2023; Yin et al., 2024; Rakovec et al., 2023). These meteorological data combinations, along with the simulated hydrological components derived from them, demonstrate lower uncertainty across Europe (Tang et al., 2023).*

b) How were the criteria used for choosing the 533 catchments? You mentioned that you try to cover all European climates, but is that the only reason? Why is there no catchment in Italy or Greece? Why are there these big differences between catchment sizes?

We addressed the conditions to select to different catchments across Europe. The revised text is as follows:

*This study includes 553 European catchments ranging in size from 500 km$^2$ to 252 000 km$^2$. Catchments were selected based on the following criteria: first, a minimum area of 500 km$^2$; second, at least 10 years of observed discharge data from the GRDC database; and third, a closed water balance condition $((P - Q)/P < 1)$. The selected catchments are divided into three categories based on the aridity index: energy-limited, mixed, and water-limited.*

c) Why do the authors select these 12 specific PET models? Please add in Appendix A1 reference to each one of the chosen formulations.

We selected widely used PET methods from different categories to ensure broad applicability across hydrology, agriculture, and climate science. The temperature-based methods are widely used in many research studies due to their minimal data requirement. Radiation-based methods

are more complex than temperature-based methods. Additionally, we included a PET formulation that accounts for stomatal responses, which has gained increasing attention due to its ability to capture the impact of stomatal regulation on evapotranspiration. We added the references in Appendix A1 in the revised manuscript, as suggested by the reviewer.

d) The authors mention that "the basins were not calibrated for each PET method to access their response in hydrological cycle components." I understand that this is a hypothesis of your study, but in any case, I assume they must be a baseline calibration. How does the model perform in this baseline calibration? Which PET is considered in this baseline calibration? Which parameter set was used in this reference calibration? Which the target variable that the model was calibrated for? I think that, in general, a deeper description of the model might help the reader to understand the implications of selecting one or other PET. Especially how the parameterization of PET-AET-soil water balance interaction is solved.

We have addressed the modelling details, including the model setup and its performance, in Referee #1 comments, Section 3) Methods and Data, specifically in subsections (e) and (f). The revised text is follows:

*We run mHM(v5.12.0) over 553 European catchments, using the meteorological data from EM-Earth and the 12 different PET estimation methods. Overall, 6 636 (12 × 553) mHM simulations are performed for all the study basins. The model was set up for each catchment at a daily temporal resolution and 0.125° × 0.125° spatial scale. All meteorological forcings were kept constant with only varying PET estimates. To calculate TWS, we aggregate soil moisture at different layers, canopy interception storage, snowpack, groundwater levels, sealed area reservoirs, and unsaturated zone reservoirs at each grid cell and time step. The hydrological components (AET, Q, and TWS) and PET are averaged over the catchment area and monthly time steps. In this research default parametrization is used for model setup. The default parameterization of mHM has been shown to perform well in previous studies (Kumar et al.,*

*2013; Rakovec et al., 2016). Furthermore, it has been demonstrated as one of the best-performing configurations compared to other large-scale hydrological models (Samaniego et al., 2019). Additionally, our assessment of model performance is consistent with the findings of Samaniego et al. (2019). Our model evaluation against discharge shows that the median Kling–Gupta Efficiency (KGE) ranges from 0.60 to 0.75 across most PET methods (Figure R3).*

e) Regarding the trend analysis, to my knowledge, Sen's slope method is a non-parametric test to compute the magnitude and direction of linear change on a time series, as the authors state in lines 146-150. To be able to talk about a trend, it needs to be significant. To assess that, other statistical tests should be performed. The Mann-Kendall test and its variations are the most commonly used for these purposes. For instance, one paper you cited, Anabalón and Sharma, 2017, used a test of the Kendall family (Seasonal Kendall trend test) to determine the trends and the Sen's slope test to determine their magnitudes. If only Sen's slope is computed, the authors might talk about the evolution of the slope but not about trends.

These are reasonable concerns, also raised by the other two Referees regarding statistical significant trends (or slopes). We already had a big discussion among the co-author group wether we should use the term "slope" or "trend" during the preparation of the original manuscript. We decided to use "trends" as it is more common in the literature, although we agree that it can lead to confusion if no statistical testing is provided, which has been done in the revised analysis. We also agree that including non-significant trends can potentially introduce noise; however, in our case, this choice was deliberate and based on the following rationale:

Including all trends enables the evaluation of spatial coherence in directional changes, which is central to our study. If weak trends were merely random noise, we would expect their directions to be evenly distributed across space. However, we observe a strong spatial alignment, even among weaker trends, which suggests an underlying signal beyond random variability. Moreover, the reliance solely on statistical significance (e.g., $p < 0.05$) can be problematic, as noted in the

editorial by Ronald L. Wasserstein and Lazar (2019) in The American Statistician. They emphasize that "no single index should substitute for scientific reasoning", and caution against rigid thresholding, which may obscure meaningful but subtle patterns.

Importantly, the scope of our study is not to isolate only the strongest or most significant trends, but rather to examine how the choice of PET method influences the direction and magnitude of all trend estimates. This broader perspective allows us to assess the systematic impact of PET selection across the full spectrum of changes (both strong and weak) which is essential for evaluating methodological sensitivity and spatial consistency. Limiting the analysis only to significant trends would risk missing meaningful shifts in regions where variability is high or records are short, even though such areas may still exhibit consistent directional behavior under different PET formulations.

That said, we conducted a comparative analysis using the Mann–Kendall trend test, computing the DCI: (1) Using all trend estimates (main manuscript), and (2) Using only statistically significant trends ($p < 0.05$; shown in Supplementary Figures R4 and R5). This comparison confirms that the main patterns and conclusions are robust, regardless of whether all trends or only significant ones are included. We have clarified this approach and justification in the revised Methods, Results, and Conclusions sections.

4) Results

a) As I stated before, the authors should not talk about trends but rather the evolution of the slope. So, sections 3.1, 3.2, and 3.3 might be rewritten in this sense or carried out the trend analysis and talk about trends.

We revised the entire Results section to reflect two cases: (1) considering all trends and (2) considering only statistically significant trends. The all-trend case includes all slope values without filtering, whereas the significant trend case includes only statistically significant slopes ($p < 0.05$).

b) Besides talking about trends, I think it would be nice for the reader to have some information about the actual values, at least for PET. I would include 12 maps, one per PET model selected with actual PET values. It would help the reader to spot differences.

We understand the reviewer intention and therefore we have included the recommended plot (Figure R8) in the supplementary material of the revised manuscript.

[Figure]

**Figure R8.** Average annual PET across European catchments for each PET method. Abbreviations used for different PET methods are TH: Thornthwaite, BR: Bair-Robertson, BC: Blaney-Criddle, OD: Oudin, MB: McGuinness-Borden, HM: Hamon, HS: Hargreaves-Samani, JH: Jensen-Haise, MD: Milly-Dunne, PT: Priestley-Taylor, PM: Penman-Monteith, $CO_2$: Modified Penman-Monteith accounts $CO_2$.

5) Discussion

   a) There are topics I would highlight in the discussion that it did not. For instance, are the different sizes of the catchments conditioning your results? Another issue I would appreciate including is a discussion about why specific models produce different results when they are within the same category. That is, why are there differences between temperature models? Is it linked to their specific formulation?

   Thank you for bringing these points. In response, we have included the suggested points in the discussion, especially regarding the variability among PET methods and the influence of catchment size. The revised text reads as follows:

   *Despite belonging to the same category, PET methods exhibit notable variation within each PET category. This is largely due to the structural and empirical differences in their formulations. For example, while both the Jensen–Haise and Hargreaves–Samani methods account for extraterrestrial radiation and air temperature, Hargreaves–Samani includes a diurnal temperature range term not present in Jensen–Haise. Thornthwaite relies instead on a heat index derived from monthly temperature. In the case of combinational methods, Penman–Monteith and its modified version differ slightly due to the addition of a $CO_2$ concentration term. Structurally similar methods like McGuinness–Bordne, Oudin, and Jensen–Haise differ primarily due to the empirical constants, which were derived based on region-specific or climate-specific conditions (Proutsos et al., 2023). In addition to PET formulation, catchment size does not influence the overall findings related to PET and other hydrological components (Figure S20). This aligns with Tang et al. (2023), who reported that spatial averaging over larger catchments results in lower uncertainty and more reliable outcomes.*

6) Summary and conclusions

   a) Conclusion 7 is obvious. I would remove it.

Thank you for this suggestion. Based on our analysis, we identified patterns between precipitation and potential evapotranspiration across different catchment categories and hydrological components. Accordingly, we have modified Conclusion 7 in the manuscript as follows:

*Precipitation primarily governs trends in all hydrological components and catchment types, except for AET in energy-limited catchments, which is largely influenced by PET variations.*

7) Technical corrections

a) Please homogenize "combination type" vs. "combinational type."

We corrected it in the revised manuscript.

b) Figure 1b: it is unclear what each dot represents. I assume they are catchments, but it seems to me less than 533.

Figure 1b represents only the test catchments, which are specifically highlighted with thick black borders in Figure 1a. There are a total of 36 data points, consisting of 12 points (one for each PET method) for each analyzed basins. To improve its clarity, the figure (Figure R9) has been further revised to partly also address the comments of Referee #3 along the same point.

[Figure]

**Figure R9.** Catchment classification to energy-limited, mixed, and water-limited categories. a) Catchment locations; black borders indicate a representative catchment of each category. b) Classification example within the Budyko space for the representative catchments. c) Annual time series of simulated hydrological components from the mesoscale hydrological model for each representative catchment and PET estimation method (TH: Thornthwaite, BR: Bair-Robertson, BC: Blaney-Criddle, OD: Oudin, MB: McGuinness-Borden, HM: Hamon, HS: Hargreaves-Samani, JH: Jensen-Haise, MD: Milly-Dunne, PT: Priestley-Taylor, PM: Penman-Monteith, $CO_2$: Modified Penman-Monteith accounts $CO_2$.). All units are in mm year$^{-1}$.

c) Line 141: "For the each" should be "for each."

We have updated it in the revised manuscript.

d) In Figures 2 and 3, I recommend using the same range on the y-axis for each variable to see the differences. In addition, I would add the number of catchments in each category, that is, 189, 330, and 34 for energy-limited, mixed, and water-limited, respectively.

To facilitate comparison within each category, we used separate scales for each catchment category and hydrological component. Using the same y-axis range for all catchment categories could make it easier to compare them directly. However, our main goal is to compare PET methods within each category separately. Adjusting the y-axis for each category helps us clearly see the trends for PET methods. If we used the same y-axis range, the higher trends in energy-limited catchments would dominate the scale, making it harder to notice the smaller changes in water-limited catchments. This would reduce the clarity and usefulness of the comparisons in water-limited catchments.

e) Figure 6, not all categories sum 553. Please revise. In addition, I think the bar representation with different numbers of catchments and models in each is not intuitive. Maybe the same structure but in a table format for the upper part, in which models are rows, would be more intuitive.

We thank the Referee for their comment. A similar point was also raised by Referee #1. We have explained the reasons for the lower catchment count in the text below and have updated the revised manuscript with the necessary details for improved clarity.

There are 14 possible patterns (Figure R6). The excluded patterns were not considered because they were represented by only a very small number of catchments. The remaining two catchments from the Thornthwaite method belonged to the (PRE-, AET+, Q+, TWS-) pattern. Including these catchments brings the total count to 553 (240 + 156 + 59 + 70 + 24 + 2 + 2). This approach ensures that all catchments are considered in the analysis. However, in the revised manuscript, we focus only on the top five patterns, excluding those with fewer catchments (Figure R7).

f) Please check the name of the PET model throughout the manuscript; there are some inconsistencies.

Thank you for your comment. We corrected such inconsistencies in the revised manuscript.

1) Overall evaluation

The authors presented a study on the impact of 12 PET formulations on the trend of a set of components of the hydrological cycle in 553 catchments across Europe. They used a large-scale rainfall-runoff model to simulate actual evapotranspiration (AET), total water storage (TWS) and runoff (Q) multiple times by varying the PET forcing according to the 12 selected methods. Then, they analysed the annual and seasonal trend of PET, AET, TWS and Q obtained thought the different PET methods. They concluded that the choice of PET formulation influences the components of the hydrological cycle.

The work has a strong potential and the issue is of great interest in the field of catchment hydrology. In addition, this experiment could help fill a gap in the literature, which currently lacks a clear understanding of the effects of different PET formulations on rainfall-runoff modelling. However, I have few major concerns, especially about the methodological approach, which I think should be addressed in order to enhance the reliability of the results, facilitate and improve their interpretation, and meet the standards required for publication in HESS.

We sincerely thank Referee #3 for their constructive feedback, which helps to further improve the quality of our manuscript. We sincerely appreciate the time and effort invested in providing such a thorough and insightful review.

Most of my concerns were already highlighted in detail by the other two referees. Therefore, I would focus exclusively on the most critical issues, which need significant improvements.

2) General comments

  a) Modelling framework and model accuracy

A more detailed description of the modelling framework is certainly needed in order to better understand the experiment and its results. Please provide information about model spatial and temporal resolution, model calibration (or previously calibrated model settings) including objective function(s), calibration/validation period, input data used, etc. If a default parameterisation is used, as stated in the very last part of the manuscript, I believe the authors should elaborate about it and its impact on the outcomes of the analysis (i.e. can it be reliable?). In general, I suggest providing a brief overview about model performances against observed streamflow (which I suppose were used somehow for model parameterisation and/or to evaluate the default parameterisation) across the study catchments. I am aware that's definitely not the focus of the study but, since the entire analysis is based on a set of model outputs (streamflow included), I believe it is important to verify (and show) model accuracy in order to consolidate the interpretation of the results and draw solid conclusions. In fact, even if on one hand good model accuracy in reproducing streamflow does not guarantee a faithful reproduction of other hydrological components, on the other hand I would tend not to rely on the state variables of a poorly performing model. Maybe you can mention about model performance in the text and report the details in the Supplement.

Finally, I agree with referee Franziska Clerc-Schwarzenbach that, if a method for potential evapotranspiration was involved in the model parameterisation, authors should provide details about it and comment about the potential effect it could have on the outcomes of the experiment.

We thank the referee for providing constructive comments. The detailed model setup and and its performance is discussed Referee #1 comment, in Section 3) Methods and Data, in subsections (e) and (f). The revised text related to model performance is as follows:

*We run mHM(v5.12.0) over 553 European catchments, using the meteorological data from EM-Earth and the 12 different PET estimation methods. Overall, 6 636 (12 × 553) mHM simulations are performed for all the study basins. The model was set up for each catchment at a*

*daily temporal resolution and 0.125° × 0.125° spatial scale. All meteorological forcings were kept constant with only varying PET estimates. To calculate TWS, we aggregate soil moisture at different layers, canopy interception storage, snowpack, groundwater levels, sealed area reservoirs, and unsaturated zone reservoirs at each grid cell and time step. The hydrological components (AET, Q, and TWS) and PET are averaged over the catchment area and monthly time steps. In this research default parametrization is used for model setup. The default parameterization of mHM has been shown to perform well in previous studies (Kumar et al., 2013; Rakovec et al., 2016). Furthermore, it has been demonstrated as one of the best-performing configurations compared to other large-scale hydrological models (Samaniego et al., 2019). Additionally, our assessment of model performance is consistent with the findings of Samaniego et al. (2019). Our model evaluation against discharge shows that the median Kling–Gupta Efficiency (KGE) ranges from 0.60 to 0.75 across most PET methods (Figure R3).*

Updated text related to model default parametrization as follows:

*Large-scale hydrological models, including mHM, typically rely on default parameterization. In mHM, these parameters were initially developed using German basins, as outlined by Samaniego et al. (2010); Kumar et al. (2013). Since then, mHM has been extensively tested across various basins and hydrological variables (Rakovec et al., 2016; Samaniego et al., 2019; Boeing et al., 2024). For instance, Rakovec et al. (2016) evaluated discharge simulations across 400 European catchments using 36 parameter sets, demonstrating consistent model performance regardless of parameterization, thereby reinforcing confidence in the model's reliability. Similar approaches have been applied in global water models; Beck et al. (2017) employed ensemble parameters derived from ten catchments, while Kumar et al. (2013) tested default parameters from European basins across 80 American catchments with varying climatic conditions. While these studies demonstrate the robustness of default parameterization, investigating how PET-specific calibration affects hydrological trends could provide valuable insights for future research.*

*Notably, the Hargreaves-Samani PET method, used in developing these parameters in mHM, demonstrated best model performance in this study but did not consistently stand out compared to other PET methods in trend analysis across hydrological components. Moreover, the study is confined to temperate European catchments, leaving a gap in assessing arid and tropical climates, where distinct patterns may emerge. While the Modified Penman-Monteith method accounts for $CO_2$, it did not exhibit substantial differences compared to the Penman-Monteith method, indicating the need for further exploration of this method. It would be interesting to assess their impact under changing climate conditions and their implications for trend assessments.*

b) Trend analysis

First of all, I am sorry to say that the trend analysis is lacking. In particular, authors computed and took into account exclusively the non-parametric Sen's slope test, which estimates the magnitude of the trend of a time series but does not ensure its statistical significance. To affirm that a signal has a trend, it must be statistically significant. Therefore, I ask to the authors to complete the trend analysis by associating a significance test (e.g. Mann-Kendall) to each trend magnitude (Sen's slope) and, consequently, change all the results and their interpretation accordingly.

In addition, I suggest excluding (maybe adopting a threshold) very week positive/negative trends when computing DCI, which may include a lot of noise and mask some aspects of your results.

These are reasonable concerns, also raised by the other two Referees. We agree that including non-significant trends can potentially introduce noise; however, in our case, this choice was deliberate and based on the following rationale:

Including all trends enables the evaluation of spatial coherence in directional changes, which is central to our study. If weak trends were merely random noise, we would expect their directions to be evenly distributed across space. However, we observe a strong spatial alignment, even among weaker trends, which suggests an underlying signal beyond random variability. Moreover, the reliance solely on statistical significance (e.g., $p < 0.05$) can be problematic, as noted in the

editorial by Ronald L. Wasserstein and Lazar (2019) in The American Statistician. They emphasize that "no single index should substitute for scientific reasoning", and caution against rigid thresholding, which may obscure meaningful but subtle patterns.

Importantly, the scope of our study is not to isolate only the strongest or most significant trends, but rather to examine how the choice of PET method influences the direction and magnitude of all trend estimates. This broader perspective allows us to assess the systematic impact of PET selection across the full spectrum of changes (both strong and weak) which is essential for evaluating methodological sensitivity and spatial consistency. Limiting the analysis only to significant trends would risk missing meaningful shifts in regions where variability is high or records are short, even though such areas may still exhibit consistent directional behavior under different PET formulations.

That said, to directly address your concern, we conducted a comparative analysis using the Mann–Kendall trend test, computing the DCI: (1) Using all trend estimates (main manuscript), and (2) Using only statistically significant trends ($p < 0.05$; shown in Supplementary Figures R4 and R5). This comparison confirms that the main patterns and conclusions are robust, regardless of whether all trends or only significant ones are included. We have clarified this approach and justification in the revised Methods, Results, and Conclusions sections.

c) Results and discussion

I personally find some parts of the results section very hard to follow. In particular, please consider reviewing the text on seasonal trends (Section 3.2) and on combination of hydrological cycle components (Section 3.4).

Thank you. We have thoroughly revised the Results and Discussion sections in the manuscript.

In addition, when commenting DCI outcomes in Figure 4 and 5, authors refer to Northern/central/Southern Europe to develop the description. It would be useful to be more specific, because sometimes the text is misleading. For instance, at line 256 you state "…Q shows

a strong decreasing trend for all PET methods in most central European catchments" but if I look at figure 5, central-Eastern DCI for Q are mostly negative. Is eastern Europe not included in "central"? If so, comment also about Eastern Europe. Again, Great Britain is considered Northern or central Europe.

We have revised these sections to include a clearer classification of regions including the geographical specification of northern, eastern, western, southern, and southeastern Europe. The revised text has been incorporated into the manuscript, with these revised sections in Lines 274–310 and 354–379 of the revised manuscript.

Figure 6 is not intuitive and difficult to interpret (and must be revised since some of the PET methods don't sum 553?). I strongly agree with the suggestions of Franziska Clerc-Schwarzenbach and Anonmymous referee #2. Also, the figure format and meaning should be explained in detail in the text before commenting it. Moreover, I suggest adding maps of the catchments coloured accordingly to the obtained combinations (or at least some of them), in order to be able to locate basins in space.

All figures have been systematically updated to present both general and statistically significant trends. Figure R7 represent as an example of this approach. Spatial variations of hydrological cycle component combinations are included in the supplementary figures (Figure R10).

The trends of AET, TWS and Q are strongly influenced not only by the PET method but also by PRE trends. Even if it is obvious, I would report PRE trends (and their significance) in the results (and not only in the Supplement) and use it to justify the trend direction of the other components.

We have added the dedicated section 3.5 to demonstrate precipitation and PET relation with other hydrological components. Revised text is as follows :

*Precipitation is an important component of the hydrological cycle. Figure R11 shows the changes in precipitation (P) without considering statistical significance. Annually, positive P trends dominate northern, western, southern, and southeastern catchments, while central Europe shows*

*mixed positive and negative trends. In western Europe, a few catchments exhibit decreasing P trends (Figure R11A). Seasonally, southern catchments experience increased P in winter, spring, and autumn but declines in summer. Southeastern Europe shows consistent P increases across all seasons, while eastern Europe exhibits negative P trends in summer and autumn and positive trends in winter and spring (Figure R11A). P demonstrates a higher correlation with Q and TWS across all catchment categories (Figure R11C). This suggests that P has a greater influence on Q and TWS than PET. In energy-limited catchments, AET is mainly driven by PET. In mixed catchments, both P and PET contribute to AET. In water-limited catchments, AET is mainly influenced by P. When we consider statistically significant P trends, only 129 catchments show significant trends at the annual scale. Across seasons, the number of statistically significant catchments varies from 20 to 61 (Figure S18). Despite the limited number of statistically significant catchments, our findings regarding the influence of P and PET on AET, Q, and TWS remain consistent with all trends.*

Finally, the discussion about the obtained combinations of hydrological cycle components is poor. I believe it should be extended.

We have extended the discussion about the combination of hydrological cycle components in the revised manuscript. Updated texts read as follows:

*Across European catchments, two dominant patterns of changes in hydrological cycle are observed when considering all trends: simultaneous increases in all components and declines in all except AET. These two patterns collectively account for over 60% of European catchments. However, when focusing only on significant trends, the pattern of increases in all components becomes more dominant, with only approximately 15% of catchments exhibiting significant pattern. This first pattern, characterized by increases in all hydrological components, aligns with findings by Teuling et al. (2019), who reported rising P, AET, and Q in central-western Europe and declines in these components in the Mediterranean. Their analysis, based on the*

*Penman–Monteith method, also indicated fewer catchments classified under this pattern compared to temperature-based methods, which show stronger responses. This contrast highlights the role of PET method selection in interpreting hydrological cycle intensification. In contrast to the first pattern, the second prominent pattern is characterized by increased AET alongside declines in P, Q, and TWS. This pattern suggests that water reserves are being depleted to sustain evapotranspiration, a mechanism particularly evident in water-limited regions. Bruno and Duethmann (2024) noted that rising atmospheric demand, without sufficient water supply, results in reduced Q and TWS. Massari et al. (2022) similarly reported that increasing AET contributes to Q reductions in water-limited regions. Even with decreasing P and Q, continued declines in TWS appear to support increases in AET (Massari et al., 2022). The contrasting nature of these two patterns underscores the critical role of PET method selection in shaping hydrological responses across different catchment categories. The implications of hydrological cycle patterns are critical for water resource management. In catchments where P, Q, TWS, and AET increase simultaneously, water availability rises, benefiting agriculture and storage. However, this intensification can also elevate flood risks, as higher P drives increases in Q and TWS while sustaining AET through enhanced soil moisture and vegetation greening, amplifying river flows and storage levels. For the hydrological cycle pattern P, Q, and TWS decrease while AET increases, elevated temperatures and persistent vegetation productivity can exacerbate soil moisture depletion, intensifying drought conditions. This scenario draws more water from storage, and vegetation tapping into deeper moisture reserves may further elevate AET, compounding stress on surface water supplies, reducing Q and TWS, and threatening agricultural and ecosystem stability.*

[Figure]

**Figure R10.** Spatial distribution of catchments corresponding to two hydrological cycle patterns for each PET method. Yellow catchment denotes an increase in all hydrological cycle components. Blue catchments represent a decrease in PRE, Q, and TWS and an increase in AET. Abbreviations used for different PET methods are TH: Thornthwaite, BR: Bair-Robertson, BC: Blaney-Criddle, OD: Oudin, MB: McGuinness-Borden, HM: Hamon, HS: Hargreaves-Samani, JH: Jensen-Haise, MD: Milly-Dunne, PT: Priestley-Taylor, PM: Penman-Monteith, $CO_2$: Modified Penman-Monteith accounts $CO_2$.

[Figure]

**Figure R11.** Spatial variation of precipitation (P) trends and their relationship with other hydrological components across catchment categories. Panel A shows the spatial distribution of increasing and decreasing annual P trends. Panel B illustrates the seasonal variation in increasing and decreasing P trends. Panel C represents the median correlation between P and AET, Q, and TWS, as well as between PET and AET, Q, and TWS, for each catchment category across all PET methods.

3) Additional minor comments

   a) Figure 1b: I would specify in the text (not only in the caption) that the example refers to the catchments with bolder black contours in panel a. In addition, I would avoid interpolating the points: please use just dots of different colour.

The interpolation lines in Figure 1b have been removed and are now represented as colored points for each catchment category (Figure R9). We revised the text as suggested by the Referee. The revised text are as follows:

*Three representative catchments from each category are indicated by dark black lines in (Figure R9a) and are plotted in (Figure R9b)*

b) line 95: Please give some information about time coverage of the datasets, which I guess can justify your following choice regarding the simulation period.

We have included the temporal and spatial information for each dataset in Table 1 of the revised manuscript.

c) lines 103-104: I perfectly understand this choice, since ERA5-Land precipitation and temperature are known to be often not accurate, leading to a degradation of model performances. However, since one may wonder why not all variables from ERA5-Land are used, I would refer to recent studies highlighting such issues (e.g Clerc-Schwarzenbach et al. 2024, Tarek et al. 2020)

Thank you for your suggestions. We have incorporated the details and quality of the input data. The additional text is as follows:

*The EM-Earth dataset provides high-quality precipitation and temperature data and has been shown to perform well over Europe (Tang et al., 2022). It has undergone climatology-based bias correction and accounts for precipitation undercatch. However, since EM-Earth does not include all necessary variables for PET estimation, we utilize ERA5-Land as a complementary dataset. ERA5-Land has been demonstrated to perform better than other reanalysis datasets, including ERA5 and ERA-Interim (Muñoz-Sabater et al., 2021). Nonetheless, its limitations in hydrological modeling have been acknowledged by Clerc-Schwarzenbach et al. (2024); Tarek et al. (2020). Several recent global studies follow a similar strategy, combining precipitation and temperature from EM-Earth with radiation, wind speed, and other meteorological variables from ERA5-Land (Tang et al., 2023; Yin et al., 2024; Rakovec et al., 2023). These meteorological data*

*combinations, along with the simulated hydrological components derived from them, demonstrate lower uncertainty across Europe (Tang et al., 2023).*

Community #1 Miyuru Gunathilake

1) General Comment: The manuscript by Thakur et al. 2024 is well written. The methodology is clear and robust. The authors used the mesoscale Hydrological Model (mHM) to simulate water balance components of 550+ catchments across Europe under diverse climatic conditions. The outputs offer valuable insights to the scientific community.

We express our deep thank to Miyuru Gunathilake for constructive feedback, which help in improving the quality of our manuscript. We sincerely appreciate the time and effort invested in providing a thorough review.

2) Minor Comments: There are some minor comments which the authors could incorporate to further enhance the readability.

a) To carry out statistical tests (Mann-Kendall etc.) the data distribution should follow certain criteria(s). (For instance, normality etc.). Have you checked for this?

We estimated the magnitude of the slope using the Sen's slope method. Due to its non-parametric nature, it does not require normality assumptions.

b) The description under "2.3.2 mesoscale Hydrological Model (mHM)" could be moved to the Appendix.

We also received valuable feedback from the other three referees regarding this section, which includes providing crucial information relevant to the modelling aspect. Therefore, we have kept this section in the main manuscript.

c) In the Abstract it is mentioned that "The findings reveal that the Jensen-Haise method produces the highest trends for PET on both annual and seasonal scales (summer, spring, and autumn)". What did you mean by "highest"? "Magnitude" wise or in terms of the "Significance" of the trend? Please be clear.

We have updated the concerned texts in the revised manuscript.

"....highest trend magnitude...."

d) Please check the manuscript for spacing. In some instance you have double spaces after the full stop.

Thank you very much for your comment. We have updated the identified double spaces.

**References**

[revised manuscript text omitted]

---

## Referee Report (RR1)

**Review of the revised manuscript "Unveiling the impact of potential evapotranspiration method selection on trends in hydrological cycle components across Europe", submitted to HESS by Vishal Thakur et al.**

*Dear Vishal Thakur and co-authors, thank you for re-submitting the revised version of your manuscript. Please find below my open comments and questions. All the best for the finalization of this study! Best wishes, Franziska Clerc-Schwarzenbach*

**General and specific comments**

In my opinion, the revised manuscript is a great improvement compared to the originally submitted version. I especially appreciate that you added additional analyses to assess the influence of the large number of temperature-based methods as well as if there is a change in the results when only significant trends are considered. Thank you for answering my concerns in such a detailed way.

Here are a few minor points that I would still like to raise:

- I think that the introduction strongly benefits from the thoughts that you added on different terms and concepts that are relevant for an evapotranspiration study. However, on lines 28-30 you state that PET is used in agriculture. I would argue (and this is also how I understand Xiang et al.) that it is actually *not* PET which is used in agriculture, but mainly $ET_0$.
- In lines 42-46, you added interesting information on the number of temperature-based, radiation-based and combinational methods, as well as on the suitability of different methods for specific climates. If all of this comes from Lu et al. (2005), I think it would be good to make this clearer, and also to state that in the last 20 years, even more formulae were developed (see for example Valiantzas (2013), 10.1016/j.jhydrol.2013.09.005). If you used other references for these new points, can you please add them?
- Thank you for not shying any effort to change Fig. 6. I really like the result and together with the additional text about it, it is now much more easily understandable for me. Potentially, you could also add in the caption (in addition to the text) why the rows do not always add up to the number of catchments, but I also understand that the caption is already quite long, so consider this just an idea.

**Technical corrections and individual typos**

Please find below some comments and suggestions for small corrections that should be implemented to improve the quality of the text. In general, I think that the manuscript will benefit from a thorough proof-reading to smooth out any remaining errors.

Please carefully go again through the list of technical corrections that I had provided with my review. Some of these have not been corrected in the revised manuscript, I noted down some points, but this may not be complete:

- (technical correction d) I think that the sentence starting at the end of line 89 (formerly line 75) is redundant (stating the same as the preceding sentence in other words) and incomplete, please double-check and correct.
- (technical correction f) "Baier-Robertson" and "McGuinness-Bordne" are still not consistently spelled correctly (and maybe this also applies to other names, I did not check them all). Please make sure that they are all used in a correct and consistent manner, including in the tables. Similarly, also make sure that the Penmann-Monteith[$CO_2$] method is named consistently.

- (technical correction g) In Table 1, the reference that should be "Tucker et al. (2004) is in a different format than the other references. Please double-check all references, especially those that were changed manually.

Note that the first paragraph of chapter 2.1 is there twice. The same applies for the paragraph starting on line 421 (very slight differences in the two occurrences).

Note that the figures in the supplementary material should be re-ordered so that they are referred to in increasing order in the manuscript (e.g., Figure S21 is mentioned already on line 224).

Please also double-check the correction of all the individual typos that I had listed in the original review, some are still there in the revised manuscript. There were also some new typos (for example: line 58: "canopy", line 202: "example", lines 480/481: brackets around references) which should be corrected.

---

## Author Response (AR2)

1) General and specific comments: In my opinion, the revised manuscript is a great improvement compared to the originally submitted version. I especially appreciate that you added additional analyses to assess the influence of the large number of temperature-based methods as well as if there is a change in the results when only significant trends are considered. Thank you for answering my concerns in such a detailed way. Here are a few minor points that I would still like to raise:

We would like to thank the Referee once again for their continued engagement and constructive feedback. We appreciate the time and effort devoted to reviewing our revised manuscript and believe the minor suggestions have helped us further refine and improve its quality

(a) I think that the introduction strongly benefits from the thoughts that you added on different terms and concepts that are relevant for an evapotranspiration study. However, on lines 28-30 you state that PET is used in agriculture. I would argue (and this is also how I understand (Xiang et al.) that it is actually not PET which is used in agriculture, but mainly ET0.

We agree with the reviewer that in agricultural applications, reference evapotranspiration ($ET_0$) is commonly used. Therefore, we have removed the sentence from the text to avoid any confusion.

*Removed text: In agriculture, it is employed for irrigation scheduling and modeling crop water requirements (Xiang et al., 2020).*

(b) In lines 42-46, you added interesting information on the number of temperature-based, radiation-based and combinational methods, as well as on the suitability of different methods for specific climates. If all of this comes from Lu et al. (2005), I think it would be good to make this clearer, and also to state that in the last 20 years, even more formulae were developed (see for

example Valiantzas (2013), 10.1016/j.jhydrol.2013.09.005). If you used other references for these new points, can you please add them?

*Thank you for your comment. We agree with your observation. We had already referred to the recent study by (Proutsos et al., 2023) and cited it at the beginning of the paragraph. However, to improve clarity, we have now added the citation again in the revised manuscript.*

*Old text: Out of these 100+ methods, the majority are temperature-based methods (40+), followed by radiation-based methods (30+) and combination-based methods (10+).*

*Revised text: Out of these 100+ methods, the majority are temperature-based methods (40+), followed by radiation-based methods (30+) and combination-based methods (10+) (Proutsos et al., 2023).*

(c) Thank you for not shying any effort to change Fig. 6. I really like the result and together with the additional text about it, it is now much more easily understandable for me. Potentially, you could also add in the caption (in addition to the text) why the rows do not always add up to the number of catchments, but I also understand that the caption is already quite long, so consider this just an idea.

This is a very helpful suggestion. We have included a dedicated paragraph (the first paragraph of Section 3.4) that discusses these key details related to the figures. Therefore, we believe it is not necessary to further extend the caption.

2) Please find below some comments and suggestions for small corrections that should be implemented to improve the quality of the text. In general, I think that the manuscript will benefit from a thorough proof-reading to smooth out any remaining errors. Please carefully go through the list of technical corrections that I had provided with my review. Some of these have not been corrected in the revised manuscript. I noted down some points, but this may not be complete:

a) (technical correction d) I think that the sentence starting at the end of line 89 (formerly line 75) is redundant (stating the same as the preceding sentence in other words) and incomplete, please double-check and correct.

*We agree with your comment; however, our intended meaning for this sentence was different. We have revised it accordingly in the updated manuscript.*

*Old text: To assess the agreement between changes in different PET methods and corresponding hydrological components.*

*Revised text: We further evaluate the agreement among PET methods by applying the Data Concurrence Index (DCI) to the trends of each corresponding hydrological cycle component (AET, Q, and TWS).*

b) (technical correction f) "Baier-Robertson" and "McGuinness-Bordne" are still not consistently spelled correctly (and maybe this also applies to other names, I did not check them all). Please make sure that they are all used in a correct and consistent manner, including in the tables. Similarly, also make sure that the Penman-Monteith[CO2] method is named consistently.

*We have revised the spellings of 'Bair' to 'Baier', 'Borden' to 'Bordne', and updated 'Penman-Monteith which account $CO_2$' to 'Penman-Monteith[$CO_2$]'. We have also corrected several previously unnoticed typographical errors throughout the manuscript.*

c) (technical correction g) In Table 1, the reference that should be "Tucker et al. (2004) is in a different format than the other references. Please double-check all references, especially those that were changed manually.

*We have corrected the format of the mentioned as well as other references.*

d) Note that the first paragraph of chapter 2.1 is there twice. The same applies for the paragraph starting on line 421 (very slight differences in the two occurrences).

*We agree with your comment. The repetitive text has been removed from both of the mentioned locations in the revised manuscript.*

e) Note that the figures in the supplementary material should be re-ordered so that they are referred to in increasing order in the manuscript (e.g., Figure S21 is mentioned already on line 224).

Thank you very much for your comment. We have arranged the supplementary figures in ascending numerical order.

f) Please also double-check the correction of all the individual typos that I had listed in the original review, some are still there in the revised manuscript. There were also some new typos (for example: line 58: "canopy", line 202: "example", lines 480/481: brackets around references) which should be corrected.

We have corrected the typographical errors in line 58 and line 202, and also revised the references in lines 480/481. Additionally, we carefully reviewed the individual typos mentioned in the previous review. Most of them had already been addressed, and the remaining ones have been incorporated in the revised manuscript.

**References**

Proutsos, N., Tigkas, D., Tsevreni, I., Alexandris, S. G., Solomou, A. D., Bourletsikas, A., Stefanidis, S., and Nwokolo, S. C.: A Thorough Evaluation of 127 Potential Evapotranspiration Models in Two Mediterranean Urban Green Sites, Remote Sensing, 15, 3680, https://doi.org/10.3390/rs15143680, 2023.

Xiang, K., Li, Y., Horton, R., and Feng, H.: Similarity and difference of potential evapotranspiration and reference crop evapotranspiration – a review, Agricultural Water Management, 232, 106 043, https://doi.org/10.1016/j.agwat.2020.106043, 2020.